# The Impact of Anisotropic Covariance Structure on the Training Dynamics and Generalization Error of Linear Networks

## Abstract

The success of deep neural networks largely depends on the statistical structure of the training data. While learning dynamics and generalization on isotropic data are well-established, the impact of pronounced anisotropy on these crucial aspects is not yet fully understood. We examine the impact of data anisotropy, represented by a spiked covariance structure, a canonical yet tractable model, on the learning dynamics and generalization error of a two-layer linear network in a linear regression setting. Our analysis reveals that the learning dynamics proceed in two distinct phases, governed initially by the input-output correlation and subsequently by other principal directions of the data structure. Furthermore, we derive an analytical expression for the generalization error, quantifying how the alignment of the spike structure of the data with the learning task improves performance. Our findings offer deep theoretical insights into how data anisotropy shapes the learning trajectory and final performance, providing a foundation for understanding complex interactions in more advanced network architectures.

## 1 Introduction

Deep learning has achieved remarkable success in diverse fields, ranging from image recognition to natural language processing, over the past decade. A key factor contributing to the high performance of deep learning is the availability of vast amounts of training data. While deep networks perform well at learning complex patterns from large data volumes, their performance is inherently sensitive to the underlying statistical structure of the data. Therefore, understanding how the data structure influences learning dynamics and generalization performance remains a fundamental problem in deep learning research.

The inherent non-linearity of deep networks makes a theoretical analysis of their learning process difficult. Consequently, simplified models, such as linear networks, are often employed. Although linear networks are less expressive than non-linear networks, they provide a tractable framework for analyzing important aspects of deep learning, including convergence (Arora et al., 2018; Du & Hu, 2019), learning dynamics (Saxe et al., 2013; 2019; Arora et al., 2018; Tarmoun et al., 2021; Min et al., 2021; Atanasov et al., 2022; Braun et al., 2022; Dominé et al., 2024; Kunin et al., 2024), and implicit bias of gradient descent (Arora et al., 2019; Woodworth et al., 2020; Pesme et al., 2021; Kunin et al., 2023; Varre et al., 2023).

The learning dynamics of linear networks are well-understood when the input data is isotropic. Under this condition, we can derive exact expressions for learning trajectories under certain initial conditions. However, real-world data is rarely isotropic, as many datasets exhibit complex anisotropic structures, where the variance is heterogeneously distributed, often being predominantly captured by a subset of principal components. Extending these theories to anisotropic data settings is notably challenging, even for these simple linear models (Tarmoun et al., 2021). Thus, a deeper theoretical understanding of how data anisotropy shapes learning trajectories and affects generalization performance is crucial.

In this study, we investigate the impact of data anisotropy on the learning dynamics and generalization performance of linear networks. We consider a linear regression problem where the input data is generated from

a spiked covariance model. This model introduces controlled anisotropy by featuring one large eigenvalue (the spike) and small remaining eigenvalues. Specifically, we examine how spike magnitude and spike-target alignment affect the learning dynamics and generalization.

Our approach yields several important results. First, we identify distinct learning phases determined by the anisotropic data structure. To analyze the weight dynamics in the learning phases, we introduce a basis that captures the anisotropic structure of the input data. By effectively reducing the dimensionality of the dynamics, this framework allows us to simplify the description of the dynamics and consistently explain the overall weight evolution. Second, we derive an analytical expression for the generalization error of linear models trained on spiked covariance data. These findings provide a deeper theoretical understanding of learning under anisotropic data, establishing a crucial perspective for analyzing the behavior of more complex neural network architectures.

Our contributions are as follows:

- We demonstrate that two-layer linear networks trained on spiked covariance data exhibit distinct two-phase learning dynamics. We show that data anisotropy is essential for the existence of the later phase, as in its absence the dynamics reduce to the behavior in isotropic settings analyzed in previous studies (Section 3).

- We develop an analytical framework that reduces the high-dimensional dynamics to a tractable two-variable system of differential equations. This approach clearly identifies the mechanisms governing weight evolution and the existence of the two learning phases (Section 3.2).

- We derive an analytical expression for the generalization error, explicitly quantifying its dependence on data anisotropy, namely the magnitude and alignment of the spike with the target vector (Section 4).

## 1.1 Related Work

While a linear network is representationally equivalent to a single-layer model, its multi-layer structure introduces non-linear learning dynamics. Consequently, linear networks have been widely used to understand the learning dynamics of deep neural networks. Numerous important insights have been obtained for linear networks, including their loss landscape (Baldi & Hornik, 1989; Fukumizu, 1998), convergence properties (Arora et al., 2018; Du & Hu, 2019), generalization capabilities (Lampinen & Ganguli, 2018; Poggio et al., 2018; Advani et al., 2020; Huh, 2020; Gunasekar et al., 2018), and the implicit bias of gradient descent (Arora et al., 2019; Woodworth et al., 2020; Pesme et al., 2021; Kunin et al., 2023; Varre et al., 2023).

The analytical tractability of linear network dynamics is significantly enhanced when the input data is whitened. Under this assumption, exact solutions for the learning dynamics have been derived in several cases (Fukumizu, 1998; Saxe et al., 2013; 2019; Arora et al., 2018; Tarmoun et al., 2021; Min et al., 2021; Atanasov et al., 2022; Braun et al., 2022; Dominé et al., 2024; Kunin et al., 2024). For instance, Saxe et al. (2013; 2019) demonstrated that for deep linear networks with small or specific spectral initialization, the learning process decouples into independent modes governed by the singular values of the weight matrices. In another analysis with specific balanced initialization, the dynamics can be described by matrix Riccati equations, which have closed-form solutions (Fukumizu, 1998; Braun et al., 2022; Tarmoun et al., 2021; Dominé et al., 2024).

In anisotropic settings, the symmetries that simplify analysis in the isotropic case, such as the rotational invariance of the input data distribution, are absent. This absence generally makes the derivation of exact closed-form solutions for weight evolution intractable, even for linear networks. For example, Tarmoun et al. (2021) suggest that obtaining exact dynamics for general anisotropic data is broadly unachievable. Consequently, research often turns to alternative approaches, such as perturbation analysis (Gidel et al., 2019) and fixed-point analysis (Zhang et al., 2024).

To describe the learning dynamics of linear networks under general data in more detail, Advani et al. (2020) provided a set of dynamical equations for multi-layer linear networks under small initialization, linking

their behavior to that of linear networks in which the weight matrix has a rank-one structure. While this observation has been used to analyze various phenomena in linear networks (Shan & Bordelon, 2021; Ji & Telgarsky, 2018; Marion & Chizat, 2024), their contribution centered on deriving these reduced equations rather than providing explicit solutions for general anisotropic data. More recently, Atanasov et al. (2022) identified two-phase learning dynamics in linear networks under small initialization. They provide exact solutions for isotropic data, but their analytical approach for anisotropic data relies on an initial linearization approximation. Consequently, while identifying the early-phase alignment with the input-output correlation direction, they cannot capture its subsequent nonlinear weight saturation. For the later phase, they deduce that weights must realign to converge, but leave its nonlinear trajectory analytically unresolved. Our work directly addresses these limitations by using a spiked covariance model to reduce the high-dimensional dynamics into a tractable two-variable system. This allows us to characterize the mechanisms and derive explicit timescales for both early-phase saturation and later-phase realignment.

Beyond training dynamics, understanding the generalization performance of neural networks, especially in overparameterized regimes, is a crucial challenge. Gradient-based optimization in such models often exhibits an implicit bias towards solutions with specific properties, such as minimum-norm solutions in linear models with small initialization (Gunasekar et al., 2017; 2018; Arora et al., 2019; Min et al., 2021; Yun et al., 2021). The generalization error of these solutions, particularly in high-dimensional settings, has been intensely studied, with random matrix theory proving to be a powerful tool for precise characterization (Bartlett et al., 2020; Belkin et al., 2020; Mei & Montanari, 2022; Atanasov et al., 2024). Random matrix theory has yielded key insights, enabling exact calculations of generalization error for models like ridgeless and kernel regression, and elucidating dependencies on dimensionality, sample size, and noise, including the "double descent" phenomenon (Adlam & Pennington, 2020; Hastie et al., 2022; Canatar et al., 2021). These advancements allow for a more detailed understanding of how data anisotropies influence generalization (Wu & Xu, 2020; Richards et al., 2021; Mel & Ganguli, 2021; Mel & Pennington, 2021; Wei et al., 2022). While some studies examine the generalization error dynamics (e.g., Pezeshki et al. (2022)), they typically assume only one layer is optimized. In contrast, we analyze the final generalization error for two-layer linear networks with joint updates. By applying the theoretical foundations of random matrix theory, our work provides a precise analytical expression for the spiked covariance model that explicitly quantifies the error's dependence on two key aspects of the anisotropy, namely, the spike's magnitude and its alignment with the target vector.

## 2 Preliminaries

In this study, we investigate a standard high-dimensional linear regression problem with structured data, using linear networks trained via gradient descent.

### 2.1 Generative model

We consider a dataset of $n$ input-output pairs $\{(\boldsymbol{x}_i, y_i)\}_{i=1}^{n}$, where $\boldsymbol{x}_i \in \mathbb{R}^d$ and $y_i \in \mathbb{R}$. The input data $\boldsymbol{x}_i$ is drawn from a multivariate Gaussian distribution with zero mean and a covariance matrix $\boldsymbol{\Sigma} \in \mathbb{R}^{d \times d}$ given by

$$\boldsymbol{\Sigma} = \sigma^2(\boldsymbol{I} + \rho \boldsymbol{\mu} \boldsymbol{\mu}^\top), \tag{1}$$

where $\|\boldsymbol{\mu}\| = 1$. The covariance matrix $\boldsymbol{\Sigma}$ has one eigenvalue $\sigma^2(1 + \rho)$ corresponding to the eigenvector $\boldsymbol{\mu}$ (the spike direction), and $d - 1$ eigenvalues equal to $\sigma^2$ in the subspace orthogonal to $\boldsymbol{\mu}$. This model is known as the spiked covariance model (Johnstone, 2001). The output $y_i \in \mathbb{R}$ is given by

$$y_i = \boldsymbol{\beta}^\top \boldsymbol{x}_i, \tag{2}$$

where $\boldsymbol{\beta} \in \mathbb{R}^d$ is the target vector with $\|\boldsymbol{\beta}\| = 1$. We define the spike-target alignment as $A = \boldsymbol{\mu}^\top \boldsymbol{\beta}$. Without loss of generality, we can assume $A \geq 0$, as the sign of $\boldsymbol{\mu}$ can be flipped without changing the covariance matrix $\boldsymbol{\Sigma}$. Using this target vector, the input-output correlation is given by

$$\boldsymbol{\Sigma}_{xy} = \mathbb{E}[y\boldsymbol{x}] = \boldsymbol{\Sigma}\boldsymbol{\beta}. \tag{3}$$

In this paper, we investigate how the dynamics and generalization error of a linear network depend on the spike magnitude $\rho$ and the spike-target alignment $A$.

### 2.2 Two-layer linear network

A two-layer linear network of width $m$ is defined as

$$f(\boldsymbol{x}) = \boldsymbol{a}^\top \boldsymbol{W} \boldsymbol{x}. \tag{4}$$

Here $\boldsymbol{x} \in \mathbb{R}^d$ is the input, $\boldsymbol{W} \in \mathbb{R}^{m \times d}$ and $\boldsymbol{a} \in \mathbb{R}^m$ are the weight parameters for the first and second layers, respectively. In particular, we consider the case where $m \geq d$.

For analytical tractability, the network is initialized with small random weights whose variances are scaled by $s^2$ (where $s \ll 1$ is the initialization scale), and is trained using full-batch gradient descent with learning rate $\eta$ (or time constant $\tau = 1/2\eta$) to minimize the squared error on the training dataset:

$$\mathcal{L}(\boldsymbol{W}, \boldsymbol{a}) = \frac{1}{n} \sum_{i=1}^{n} \left( f(\boldsymbol{x}_i) - y_i \right)^2. \tag{5}$$

In the limit of an infinitesimal learning rate, the learning dynamics of gradient descent are well approximated by the following gradient flow:

$$\tau \frac{d}{dt} \boldsymbol{W} = \boldsymbol{a} \left( \hat{\boldsymbol{\Sigma}}_{xy}^\top - \boldsymbol{a}^\top \boldsymbol{W} \hat{\boldsymbol{\Sigma}} \right), \tag{6}$$

$$\tau \frac{d}{dt} \boldsymbol{a}^\top = \left( \hat{\boldsymbol{\Sigma}}_{xy}^\top - \boldsymbol{a}^\top \boldsymbol{W} \hat{\boldsymbol{\Sigma}} \right) \boldsymbol{W}^\top, \tag{7}$$

where $\hat{\boldsymbol{\Sigma}}_{xy}$ denotes the empirical input-output correlation, and $\hat{\boldsymbol{\Sigma}}$ denotes the empirical input covariance:

$$\hat{\boldsymbol{\Sigma}}_{xy} = \frac{1}{n} \sum_{i=1}^{n} y_i \boldsymbol{x}_i, \quad \hat{\boldsymbol{\Sigma}} = \frac{1}{n} \sum_{i=1}^{n} \boldsymbol{x}_i \boldsymbol{x}_i^\top. \tag{8}$$

## 3 Learning Dynamics in Two-Layer Linear Network under Spiked Covariance Model

### 3.1 Empirical Observations

We begin by empirically examining the learning dynamics of a two-layer linear network using numerical simulations. To capture the essential behavior of the weight parameters, we show the learning trajectory in a two-dimensional phase plane defined by the coordinates $\hat{\boldsymbol{r}}_1^\top \boldsymbol{W} \boldsymbol{\mu}$ and $\hat{\boldsymbol{r}}_1^\top \boldsymbol{W} \boldsymbol{\mu}_\perp$, where $\hat{\boldsymbol{r}}_1$ is a fixed vector and $\boldsymbol{\mu}_\perp$ denotes a vector orthogonal to $\boldsymbol{\mu}$. Figure 1 shows this trajectory, together with orthogonal lines whose directions are defined by the vectors $\boldsymbol{v}_1$ and $\boldsymbol{v}_2$ (precise definitions of $\hat{\boldsymbol{r}}_1$, $\boldsymbol{v}_1$, and $\boldsymbol{v}_2$ are provided in the subsequent subsection). The learning trajectory clearly exhibits two distinct phases. It initially evolves primarily along the direction of $\boldsymbol{v}_1$, then makes a sharp turn and develops along $\boldsymbol{v}_2$ until it converges to the final solution.

These two-phase learning dynamics are also evident in the time evolution of the training loss. In Figure 2, we plot the loss together with the temporal evolution of the $\boldsymbol{v}_1$ and $\boldsymbol{v}_2$ components of $\hat{\boldsymbol{r}}_1^\top \boldsymbol{W}$, namely, $\hat{\boldsymbol{r}}_1^\top \boldsymbol{W} \boldsymbol{v}_1$ and $\hat{\boldsymbol{r}}_1^\top \boldsymbol{W} \boldsymbol{v}_2$. Initially, the training loss remains on a plateau before dropping sharply. This drop corresponds to the rapid growth of the $\boldsymbol{v}_1$ component of $\hat{\boldsymbol{r}}_1^\top \boldsymbol{W}$, while the $\boldsymbol{v}_2$ component remains negligible. Subsequently, the loss decreases smoothly, driven by the growth of the $\boldsymbol{v}_2$ component of $\hat{\boldsymbol{r}}_1^\top \boldsymbol{W}$ and adjustments to the $\boldsymbol{v}_1$ component.

Based on these empirical observations, we define two distinct phases of learning: an *early phase*, characterized by the rapid growth of the $\boldsymbol{v}_1$ component of $\hat{\boldsymbol{r}}_1^\top \boldsymbol{W}$, and a *later phase*, characterized by the subsequent growth of its $\boldsymbol{v}_2$ component. In the following, we provide a theoretical analysis of these observed dynamics.

### 3.2 Theoretical Analysis

To analyze the two-phase behavior, we consider the limit as $n \to \infty$ with fixed input dimension $d$, such that the empirical covariance matrices $\hat{\boldsymbol{\Sigma}}$ and the empirical input-output correlation vector $\hat{\boldsymbol{\Sigma}}_{xy}$ converge to their population counterparts, $\hat{\boldsymbol{\Sigma}} \to \boldsymbol{\Sigma}$ and $\hat{\boldsymbol{\Sigma}}_{xy} \to \boldsymbol{\Sigma}_{xy} = \boldsymbol{\Sigma} \boldsymbol{\beta}$, respectively.

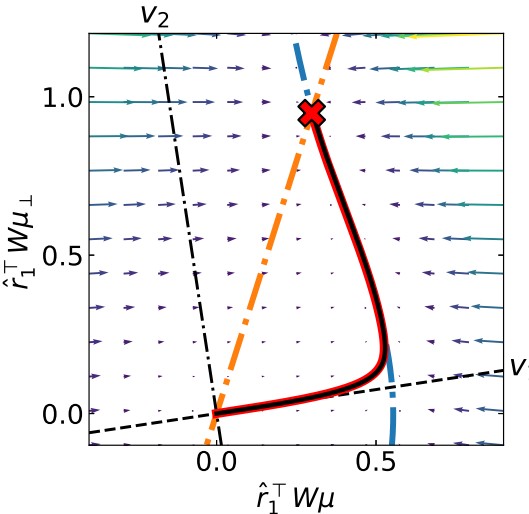

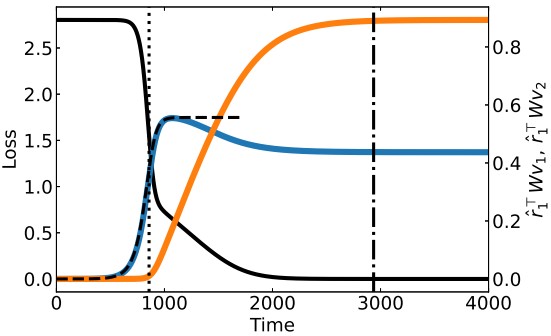

Figure 1: Learning trajectories in a phase plane with the spike direction $\boldsymbol{\mu}$ as the horizontal axis and the perpendicular direction $\boldsymbol{\mu}_\perp = (\boldsymbol{I} - \boldsymbol{\mu}\boldsymbol{\mu}^\top)\boldsymbol{\beta}/\|(\boldsymbol{I} - \boldsymbol{\mu}\boldsymbol{\mu}^\top)\boldsymbol{\beta}\|$ as the vertical axis. The vector field and nullclines (blue: $dw_1/dt = 0$, orange: $dw_2/dt = 0$) are from the reduced two-variable dynamics (Equations (16) and (17)). The red trajectory is projected from a simulation, while the black trajectory is from the reduced two-variable dynamics (Equations (16) and (17)). The basis directions, $\boldsymbol{v}_1$ (dashed) and $\boldsymbol{v}_2$ (dash-dotted), are also shown. The close match between trajectories validates the dimensionality reduction analysis. The parameters are set to $A = 0.3$, $\rho = 20$, and $\sigma^2 = 1$, with the number of training samples $n = 10000$. The network parameters are $d = 30$ and $m = 50$ and the initialization is performed as $W_{ij} \sim \mathcal{N}(0, s^2/d)$ and $a_i \sim \mathcal{N}(0, s^2/m)$ with $s = 10^{-5}$.

Figure 2: Time evolution of the training loss and magnitudes of weight components. The plotted values are the numerically obtained training loss (black) and the projections of the weight vectors $\boldsymbol{w}_1$ (blue) and $\boldsymbol{w}_2$ (orange) onto the initial growth direction $\hat{\boldsymbol{r}}_1$, namely, $\hat{\boldsymbol{r}}_1^\top \boldsymbol{W} \boldsymbol{v}_1$ and $\hat{\boldsymbol{r}}_1^\top \boldsymbol{W} \boldsymbol{v}_2$, respectively. The black dashed line is the analytical approximation for the early phase given by Equation (22), with its initial value determined by Equation (18). Vertical dotted lines and vertical dash-dotted lines indicate characteristic timescales for the early and later phases, respectively, with the latter defined in Inequality (23) using $\delta = 3$. The parameters are the same as those in Figure 1.

### 3.2.1 Dimensionality Reduction to a Two-Variable System

To reduce the dimensionality of the dynamics, we first introduce a basis adapted to the data structure, designed to capture the principal axes of learning and thus allowing for a clear characterization of each phase. We construct this basis, denoted by the matrix $\boldsymbol{V} = (\boldsymbol{v}_1, \boldsymbol{v}_2, \ldots, \boldsymbol{v}_d) \in \mathbb{R}^{d \times d}$, as follows:

- The first basis vector is defined as $\boldsymbol{v}_1 = \boldsymbol{\Sigma}_{xy}/\|\boldsymbol{\Sigma}_{xy}\|$, to capture the weight growth along the input-output correlation direction in the early phase.

- The second basis vector, $\boldsymbol{v}_2$, is defined based on the alignment of $\boldsymbol{\Sigma}_{xy}$ and $\boldsymbol{\beta}$. If they are not parallel, $\boldsymbol{v}_2$ is defined as the normalized vector in the plane spanned by $\{\boldsymbol{\Sigma}_{xy}, \boldsymbol{\beta}\}$ that is orthogonal to $\boldsymbol{v}_1$. In this case, it captures the adjustment of the weights toward $\boldsymbol{\beta}$. The sign of $\boldsymbol{v}_2$ will be specified just below for analytical convenience. In the parallel case, where no such adjustment occurs, $\boldsymbol{v}_2$ is simply chosen as any normalized vector orthogonal to $\boldsymbol{v}_1$ to complete the basis.

- The remaining vectors, $\{\boldsymbol{v}_3, \ldots, \boldsymbol{v}_d\}$, form an orthonormal basis for the subspace orthogonal to the span of $\{\boldsymbol{\Sigma}_{xy}, \boldsymbol{\beta}\}$.

Using the basis $\boldsymbol{V}$, we can represent the weight matrix as $\boldsymbol{W} = (\boldsymbol{w}_1, \boldsymbol{w}_2, \ldots, \boldsymbol{w}_d)\boldsymbol{V}^\top$. By substituting this expression into the gradient flow equation (Equation (6)), we obtain the following dynamics for each component vector:

$$\tau \frac{d}{dt}\boldsymbol{w}_1 = \|\boldsymbol{\Sigma}_{xy}\|\boldsymbol{a} - (\lambda_1 \boldsymbol{a}^\top \boldsymbol{w}_1 - \nu \boldsymbol{a}^\top \boldsymbol{w}_2)\boldsymbol{a}, \tag{9}$$

$$\tau \frac{d}{dt}\boldsymbol{w}_2 = -(-\nu \boldsymbol{a}^\top \boldsymbol{w}_1 + \lambda_2 \boldsymbol{a}^\top \boldsymbol{w}_2)\boldsymbol{a}, \tag{10}$$

$$\tau \frac{d}{dt}\boldsymbol{w}_i = -\sigma^2 \boldsymbol{a}^\top \boldsymbol{w}_i \boldsymbol{a}, \quad (i = 3, \ldots, d). \tag{11}$$

Similarly, from Equation (7), the dynamics for the second-layer weight $\boldsymbol{a}$ are given by:

$$\tau \frac{d}{dt}\boldsymbol{a} = \|\boldsymbol{\Sigma}_{xy}\|\boldsymbol{w}_1 - (\lambda_1 \boldsymbol{a}^\top \boldsymbol{w}_1 - \nu \boldsymbol{a}^\top \boldsymbol{w}_2)\boldsymbol{w}_1 - (-\nu \boldsymbol{a}^\top \boldsymbol{w}_1 + \lambda_2 \boldsymbol{a}^\top \boldsymbol{w}_2)\boldsymbol{w}_2 - \sum_{i=3}^{d} \sigma^2 \boldsymbol{a}^\top \boldsymbol{w}_i \boldsymbol{w}_i, \tag{12}$$

where $\lambda_1 = \boldsymbol{v}_1^\top \boldsymbol{\Sigma} \boldsymbol{v}_1$, $\lambda_2 = \boldsymbol{v}_2^\top \boldsymbol{\Sigma} \boldsymbol{v}_2$, and $\nu = -\boldsymbol{v}_1^\top \boldsymbol{\Sigma} \boldsymbol{v}_2$. Here, we choose the direction of $\boldsymbol{v}_2$ so that $\nu$ is positive. The derivation of Equations (9) to (12), together with explicit forms of $\|\boldsymbol{\Sigma}_{xy}\|$, $\lambda_1$, $\lambda_2$, and $\nu$, is given in Appendix A.

With the dynamics expressed in this adapted basis, we can further reduce the high-dimensional system by isolating the active subspace. From Equation (11), it follows that $\tau \frac{d}{dt}\|\boldsymbol{w}_i\|^2 = -2\sigma^2 (a^\top w_i)^2 \leq 0$ for $i \geq 3$. This guarantees that the magnitude of these components is monotonically non-increasing. Since the initial values $\boldsymbol{w}_i(0)$ are of scale $O(s)$, these components remain bounded by $O(s)$ throughout the entire training. Thus, the overall dynamics are effectively confined to the subspace spanned by $\boldsymbol{w}_1$, $\boldsymbol{w}_2$, and $\boldsymbol{a}$.

Focusing on this restricted subspace, we can now examine the beginning of learning. For a two-layer linear network initialized at the small scale $s$, its dynamics can be approximated by linear differential equations by neglecting higher-order terms:

$$\tau \frac{d}{dt}\boldsymbol{w}_1 = \|\boldsymbol{\Sigma}_{xy}\|\boldsymbol{a} + O(s^3), \quad \tau \frac{d}{dt}\boldsymbol{a} = \|\boldsymbol{\Sigma}_{xy}\|\boldsymbol{w}_1 + O(s^3) \tag{13}$$

while $\tau d\boldsymbol{w}_2/dt = O(s^3)$. These equations decouple into an exponentially growing mode $(\boldsymbol{a} + \boldsymbol{w}_1)/2$ and a decaying mode $(\boldsymbol{a} - \boldsymbol{w}_1)/2$. Consequently, $\boldsymbol{w}_1$ and $\boldsymbol{a}$ rapidly align and grow symmetrically along the input-output correlation direction $\boldsymbol{v}_1 = \boldsymbol{\Sigma}_{xy}/\|\boldsymbol{\Sigma}_{xy}\|$. In contrast, $\boldsymbol{w}_2$ lacks this initial driving force and remains at the initialization scale $O(s)$. The following lemma formalizes the exponential growth of these dominant components, $\boldsymbol{w}_1$ and $\boldsymbol{a}$, at the beginning of learning:

**Lemma 3.1.** *There exists a constant $C > 0$ such that for all sufficiently small $s > 0$ and for all $0 \leq t < \frac{2\tau}{3(\|\boldsymbol{\Sigma}_{xy}\| + 2\operatorname{Tr}\boldsymbol{\Sigma})} \log \frac{1}{s}$, the solutions $\boldsymbol{w}_1$ and $\boldsymbol{a}$ to Equation (13) satisfy*

$$\left\|\boldsymbol{w}_1(t) - \boldsymbol{r}_1 e^{\|\boldsymbol{\Sigma}_{xy}\|t/\tau}\right\| \leq Cs, \quad \left\|\boldsymbol{a}(t) - \boldsymbol{r}_1 e^{\|\boldsymbol{\Sigma}_{xy}\|t/\tau}\right\| \leq Cs \tag{14}$$

*where $\boldsymbol{r}_1 = (\boldsymbol{a}(0) + \boldsymbol{w}_1(0))/2$.*

The proof of Lemma 3.1 is in Appendix B.1. This result indicates that the weights initially grow along the direction $\hat{\boldsymbol{r}}_1 = \boldsymbol{r}_1/\|\boldsymbol{r}_1\|$. The following lemma establishes that these vectors remain confined to this initial growth direction throughout the entire learning phase, with the orthogonal components strictly bounded by $O(s)$:

**Lemma 3.2.** *Let $\boldsymbol{P}_\perp = \boldsymbol{I} - \hat{\boldsymbol{r}}_1 \hat{\boldsymbol{r}}_1^\top$ be the projection operator. There exists a constant $C_\perp > 0$ such that for all sufficiently small $s > 0$ and for all $t \geq 0$, the orthogonal components of the trajectories satisfy:*

$$\|\boldsymbol{P}_\perp \boldsymbol{a}(t)\| + \|\boldsymbol{P}_\perp \boldsymbol{w}_1(t)\| + \|\boldsymbol{P}_\perp \boldsymbol{w}_2(t)\| \leq C_\perp s. \tag{15}$$

The proof of Lemma 3.2 is deferred to Appendix B.3.

This rank-one condensation enables us to describe the evolution of these vectors using only their scalar magnitudes: $\boldsymbol{a}(t) \approx a(t)\hat{\boldsymbol{r}}_1$, $\boldsymbol{w}_1(t) \approx w_1(t)\hat{\boldsymbol{r}}_1$, and $\boldsymbol{w}_2(t) \approx w_2(t)\hat{\boldsymbol{r}}_1$. This effectively reduces the dimensionality of the problem and implies that the first-layer weight matrix can be approximated as $\boldsymbol{W} \approx \hat{\boldsymbol{r}}_1 \boldsymbol{w}(t)^\top \boldsymbol{V}^\top$, where $\boldsymbol{w}(t) = (w_1(t), w_2(t), 0, \ldots, 0)^\top$. This rank-one structure of the weight matrix is consistent with findings in previous research (Advani et al., 2020; Atanasov et al., 2022; Shan & Bordelon, 2021; Ji & Telgarsky, 2018; Marion & Chizat, 2024).

To further decouple the system, we employ a conservation law $\frac{d}{dt}(\boldsymbol{a}\boldsymbol{a}^\top - \sum_{i=1}^{d} \boldsymbol{w}_i \boldsymbol{w}_i^\top) = 0$, which can be derived from the fact that the time derivatives of $\boldsymbol{a}\boldsymbol{a}^\top$ and $\boldsymbol{W}\boldsymbol{W}^\top$ are identical under the gradient flow dynamics (Equations (6) and (7)). Considering the initialization condition, this implies $\boldsymbol{a}\boldsymbol{a}^\top - \sum_{i=1}^{d} \boldsymbol{w}_i \boldsymbol{w}_i^\top = O(s^2)$ for all time. Given that $\boldsymbol{w}_i$ for $i \geq 3$ remain at $O(s)$, taking the trace simplifies this relation to $a(t)^2 \approx w_1(t)^2 + w_2(t)^2$. This allows us to eliminate $a(t)$ and exactly reduce the high-dimensional dynamics into a closed two-variable system, formally established in the following theorem:

**Theorem 3.3.** *Suppose the original two-layer linear network is initialized at a sufficiently small scale $s > 0$. Let $(w_1, w_2)$ be the unique solution to the initial value problem*

$$\tau \frac{d}{dt} w_1 = \|\boldsymbol{\Sigma}_{xy}\| \sqrt{w_1^2 + w_2^2} - \lambda_1 w_1(w_1^2 + w_2^2) + \nu w_2(w_1^2 + w_2^2), \tag{16}$$

$$\tau \frac{d}{dt} w_2 = \nu w_1(w_1^2 + w_2^2) - \lambda_2 w_2(w_1^2 + w_2^2), \tag{17}$$

*with initial conditions defined by the projections of the network's initial weights*

$$w_1(0) = \frac{1}{2} \left[ \hat{\boldsymbol{r}}_1^\top (\boldsymbol{a}(0) + \boldsymbol{w}_1(0)) - \frac{(\hat{\boldsymbol{r}}_1^\top \boldsymbol{w}_2(0))^2}{\hat{\boldsymbol{r}}_1^\top (\boldsymbol{a}(0) + \boldsymbol{w}_1(0))} \right], \qquad w_2(0) = \hat{\boldsymbol{r}}_1^\top \boldsymbol{w}_2(0). \tag{18}$$

*Then, there exists a constant $C > 0$, independent of $s$, such that for all $t \geq 0$, the trajectories $\boldsymbol{a}(t)$, $\boldsymbol{w}_1(t)$, and $\boldsymbol{w}_2(t)$ of the exact gradient flow defined in Equations (9), (10) and (12) are well-approximated by the solution $(w_1, w_2)$, with the approximation error satisfying*

$$\left\| \boldsymbol{a}(t) - \hat{\boldsymbol{r}}_1 \sqrt{w_1(t)^2 + w_2(t)^2} \right\| \leq Cs, \quad \|\boldsymbol{w}_1(t) - \hat{\boldsymbol{r}}_1 w_1(t)\| \leq Cs, \quad \|\boldsymbol{w}_2(t) - \hat{\boldsymbol{r}}_1 w_2(t)\| \leq Cs. \tag{19}$$

The proof of Theorem 3.3 is provided in detail in Appendix B.4. While random initialization generally introduces an $O(s^2)$ discrepancy between $a(t)^2$ and $w_1(t)^2 + w_2(t)^2$, Equations (16) and (17) seem to describe the dynamics under the exact relationship $a(t)^2 = w_1(t)^2 + w_2(t)^2$. As rigorously shown in Appendix B.4, the validity of this description relies on the initialization given in Equation (18). These initial conditions are constructed so that the initial discrepancy is confined entirely to the decaying mode $(\boldsymbol{a} - \boldsymbol{w}_1)/2$, while the exponentially growing mode $(\boldsymbol{a} + \boldsymbol{w}_1)/2$ is preserved exactly. As a result, the approximation error does not amplify during the early phase.

### 3.2.2 Two-Phase Learning Dynamics

We now analyze the learning dynamics of the network using the reduced dynamics established in Theorem 3.3. Since these reduced dynamics track the high-dimensional trajectory up to an $O(s)$ error, analyzing them is sufficient to understand the actual network behavior.

**Early Phase.** Starting from the small initialization, $w_1$ grows exponentially as shown in Lemma 3.1. While Atanasov et al. (2022) identified this initial alignment along the direction of $\boldsymbol{v}_1$ using a linearized multivariate system, the subsequent temporal evolution of the weight magnitude under anisotropic data was not fully explored.

To characterize this evolution, we must examine the behavior of the component $w_2$. As established in the following proposition, the growth of $w_2$ is strictly bounded by $w_1^3$. This guarantees that $w_1$ remains effectively decoupled from $w_2$, dominating the early phase dynamics.

Formally, we define a constant $R$ satisfying $\|\boldsymbol{\Sigma}_{xy}\| - \lambda_1 R^2 > 0$ and an exit time $T_R = \inf\{t > 0 : w_1^2 + w_2^2 \geq R^2\}$. For the subsequent claims, we restrict our analysis to the interval $t \in [0, T_R)$.

**Proposition 3.4.** *There exists a constant $C > 0$ such that for all sufficiently small $s > 0$ and all $t \in [0, T_R)$, the following inequality holds:*

$$|w_2(t)| \leq \frac{2\nu\sqrt{1 + (\nu/\lambda_2)^2}}{3(\|\mathbf{\Sigma}_{xy}\| - \lambda_1 R^2)}|w_1(t)|^3 + Cs. \tag{20}$$

The proof of Proposition 3.4 is in Appendix C.1.1. Building on the strong suppression of $w_2$, we can approximate the evolution of $w_1$ with a single scalar differential equation:

**Proposition 3.5.** *For all sufficiently small $s > 0$ and all $t \in [0, T_R)$, the evolution of $w_1(t)$ is governed by*

$$\tau \frac{d}{dt} w_1(t) = \|\mathbf{\Sigma}_{xy}\| w_1(t) - \lambda_1 w_1(t)^3 + \varepsilon_R(t) \tag{21}$$

*where the remainder $\varepsilon_R(t)$ satisfies $|\varepsilon_R(t)| \leq C_1|w_1(t)|^5 + C_2 s$ for constants $C_1, C_2 > 0$ independent of $s$.*

The proof of Proposition 3.5 is provided in Appendix C.1.2. As long as $w_1$ has not reached a significant magnitude, the residual $\varepsilon_R(t)$ remains suppressed, effectively decoupling the dynamics of $w_1$ from the other variables. Solving this truncated equation yields the following sigmoidal growth:

$$w_1(t) \approx \sqrt{\frac{\|\mathbf{\Sigma}_{xy}\|}{\left(1 - e^{-2\|\mathbf{\Sigma}_{xy}\|t/\tau} + \frac{\|\mathbf{\Sigma}_{xy}\|}{\lambda_1}w_1(0)^{-2}e^{-2\|\mathbf{\Sigma}_{xy}\|t/\tau}\right)\lambda_1}}. \tag{22}$$

The term $\frac{\|\mathbf{\Sigma}_{xy}\|}{\lambda_1}w_1(0)^{-2}e^{-2\|\mathbf{\Sigma}_{xy}\|t/\tau}$ in the denominator initially dominates the expression due to the small initial value $w_1(0) = O(s)$. As the exponential factor decays, this term diminishes rapidly, causing $w_1(t)$ to transition sharply from its initial $O(s)$ scale to its $O(1)$ saturation value, approximately $\sqrt{\|\mathbf{\Sigma}_{xy}\|/\lambda_1}$.

Equation (22) reveals that, ignoring the residual $\varepsilon_R(t)$, the magnitude evolves according to the same sigmoidal law governing isotropic data. This observation suggests that the anisotropy primarily determines the alignment direction, while the growth dynamics of the magnitude in this phase effectively follow the isotropic case along that direction.

This sigmoidal shape and its characteristic timescale, $\frac{\tau}{2\|\mathbf{\Sigma}_{xy}\|}\log(\|\mathbf{\Sigma}_{xy}\|/(\lambda_1 s^2))$, are visually confirmed in Figure 2. This implies that learning in this phase becomes faster as $\|\mathbf{\Sigma}_{xy}\| = \sigma^2\sqrt{1 + ((1 + \rho)^2 - 1)A^2}$ increases, which occurs with a larger spike magnitude $\rho$ and greater alignment $A$ between the spike direction and the target vector.

**Later Phase.** Next, we examine the phase that occurs once $w_1$ approaches saturation. As $w_1$ nears saturation, the upper bound for $w_2$ established in Proposition 3.4 becomes less restrictive. In general anisotropic settings where the target vector $\boldsymbol{\beta}$ is neither perfectly aligned with nor orthogonal to the spike direction $\boldsymbol{\mu}$ ($0 < A < 1$), the network must adjust its weights to capture the target using a component other than $w_1$. The growth of the second component $w_2$ is driven by the coupling term $\nu$, which is non-zero only under this alignment condition and when the data is anisotropic ($\rho > 0$).

Conversely, if these conditions are not met (i.e., if $A = 0$, $A = 1$, or $\rho = 0$), the term $\nu$ vanishes. As seen from Equation (17), $w_2$ does not grow beyond its initial $O(s)$ magnitude in these cases. In such simpler cases, the learning dynamics are fully described by the early phase alone, a scenario that has been extensively studied (Saxe et al., 2013; Fukumizu, 1998; Atanasov et al., 2022).

While the coupled differential equations for $w_1$ and $w_2$ are analytically intractable in general, the following proposition provides a conditional upper bound on the time required for $w_2$ to approach its final value.

**Proposition 3.6.** *Let $w_{1\infty} = \lim_{t \to \infty} w_1(t)$ and $w_{2\infty} = \lim_{t \to \infty} w_2(t) > 0$. Choose $0 < \varepsilon < w_{2\infty}$, and fix $\delta > 0$ such that $\varepsilon < (1 - e^{-\delta})w_{2\infty}$. Let $t_1 < t_2$ be the first times at which $w_2(t_1) = \varepsilon$ and $w_2(t_2) = (1 - e^{-\delta})w_{2\infty}$, respectively. Suppose that these times exist and that, for all $t \in [t_1, t_2]$, the trajectory satisfies $w_1(t) \geq w_{1\infty}$ and $w_2(t) < w_{2\infty}$. Then the duration of this later phase $t_2 - t_1$ is bounded above by*

$$t_2 - t_1 \leq \frac{\tau}{\lambda_2}\left[\frac{\nu}{\lambda_2}\arctan\left(\frac{\nu}{\lambda_2}(1 - e^{-\delta})\right) + \frac{1}{2}\log\left(1 + \left(\frac{\nu}{\lambda_2}(1 - e^{-\delta})\right)^2\right) + \delta\right]. \tag{23}$$

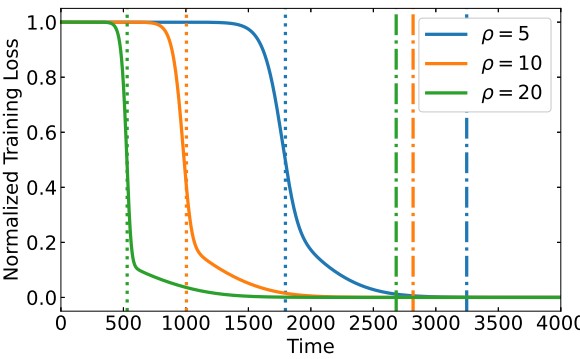
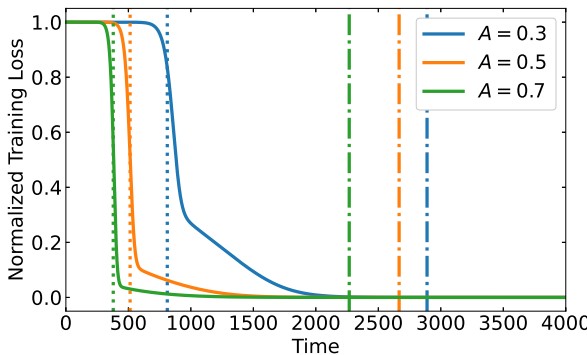

(a) Varying $\rho$ while keeping $A = 0.5$ fixed.     (b) Varying $A$ while keeping $\rho = 20$ fixed.

Figure 3: Evolution of the normalized training loss for different settings of (a) spike magnitude $\rho$ and (b) spike-target alignment $A$. The loss $\mathcal{L}$ is normalized by the empirical second moment of the labels, $\frac{1}{n} \sum_{i=1}^{n} y_i^2$. Vertical dotted and dash-dotted lines indicate the characteristic timescales for the early and later learning phases, respectively, with the latter defined in Inequality (23) using $\delta = 3$. Other parameters are the same as in Figure 1.

This bound is tight if $w_1$ changes little during the growth of $w_2$. The analysis of the timescale highlights that the speed of the later phase is governed by $1/\lambda_2$, representing the inverse of data variance along the $\boldsymbol{v}_2$ direction. The proof of Proposition 3.6 can be found in Appendix C.2.

Figure 2 also shows the later-phase dynamics, as the projection $\hat{\boldsymbol{r}}_1^\top \boldsymbol{W} \boldsymbol{v}_2$ (corresponding to $w_2$ in the reduced dynamics) begins to grow after $\hat{\boldsymbol{r}}_1^\top \boldsymbol{W} \boldsymbol{v}_1$ (corresponding to $w_1$) has nearly saturated. Furthermore, the characteristic timescale $t_2$ derived in Inequality (23), marked by a vertical dash-dotted line, coincides well with the saturation of the $w_2$ component, validating the theoretical analysis.

The effects of spike magnitude $\rho$ and spike-target alignment $A$ on the training loss are summarized in Figure 3. As shown in the figure, the learning process indeed consists of two phases, with the acceleration in the early phase being the dominant factor shaping the overall training trajectory. This dominance of the early phase, which is more pronounced for a larger spike magnitude $\rho$ and stronger spike-target alignment $A$, leads to a substantial initial loss reduction and keeps the training loss consistently lower throughout the process. Conversely, when these parameters are small, the loss decreases more slowly, and it becomes difficult to distinguish between the two phases. The theory developed here explains these observations by showing how anisotropic data structures, namely, spike magnitude $\rho$ and spike-target alignment $A$, influence the two learning phases. A larger spike magnitude $\rho$ primarily accelerates the early phase by increasing the magnitude of the input-output correlation $\|\boldsymbol{\Sigma}_{xy}\|$, while a stronger spike-target alignment $A$ achieves this by slowing the convergence of the later phase through a reduction of $\lambda_2$, i.e., the data variance along the direction of $\boldsymbol{v}_2$.

## 4 Generalization Error under Spiked Covariance Model

We now investigate how the spiked covariance structure affects the generalization performance. While the behavior of generalization error is well-established for isotropic data, its precise dependence on the key parameters of an anisotropic structure, in particular the spike magnitude $\rho$ and spike-target alignment $A$, has not been fully characterized. This section provides a quantitative analysis of this relationship by deriving a precise analytical expression for the generalization error. Our result explicitly demonstrates this connection, offering a clear theoretical explanation for how these structural properties of data influence generalization.

We focus on the high-dimensional regime where the number of samples $n$ and the data dimension $d$ are large, specifically when $n, d \to \infty$ such that their ratio $\gamma = d/n > 1$. Under this condition, the regression problem admits infinitely many solutions that achieve the optimal training loss. It is known that for high-dimensional regression, linear networks trained with gradient flow from a small initialization converge to a solution close

to the minimum $\ell_2$-norm solution given by:

$$\hat{\boldsymbol{\beta}}_0 = \boldsymbol{X}^\top (\boldsymbol{X}\boldsymbol{X}^\top)^+ \boldsymbol{y}, \tag{24}$$

where $\boldsymbol{X} \in \mathbb{R}^{n \times d}$ is the design matrix whose rows are the input data $\boldsymbol{x}_i$, and $\boldsymbol{y} \in \mathbb{R}^n$ is the vector of corresponding outputs $y_i$ (Min et al., 2021; Yun et al., 2021).

This solution is also identical to that obtained by ridgeless regression, which corresponds to taking the limit $\lambda \to 0^+$ in ridge regression:

$$\begin{aligned}
\hat{\boldsymbol{\beta}}_\lambda &= \arg\min_{\boldsymbol{\beta}} \frac{1}{n} \sum_{i=1}^n (y_i - \boldsymbol{\beta}^\top \boldsymbol{x}_i)^2 + \lambda \|\boldsymbol{\beta}\|^2 \\
&= (\boldsymbol{X}^\top \boldsymbol{X} + \lambda n \boldsymbol{I})^{-1} \boldsymbol{X}^\top \boldsymbol{y}.
\end{aligned} \tag{25}$$

Therefore, the generalization error of the solution $R_0 = \mathbb{E}[(\hat{\boldsymbol{\beta}}_0^\top \boldsymbol{x} - \boldsymbol{\beta}^\top \boldsymbol{x})^2]$ can be evaluated by taking the ridgeless limit of the ridge regression error

$$R_\lambda = \mathbb{E}[(\hat{\boldsymbol{\beta}}_\lambda^\top \boldsymbol{x} - \boldsymbol{\beta}^\top \boldsymbol{x})^2]. \tag{26}$$

In the high-dimensional limit, recent advances in random matrix theory (Wei et al., 2022; Hastie et al., 2022; Canatar et al., 2021; Wu & Xu, 2020; Richards et al., 2021; Mel & Ganguli, 2021; Atanasov et al., 2024) establish that $R_\lambda$ converges to:

$$\hat{R}_\lambda = \frac{\partial \kappa(\lambda)}{\partial \lambda} \kappa(\lambda)^2 \sum_{i=1}^d \frac{\lambda_i}{(\kappa(\lambda) + \lambda_i)^2} (\boldsymbol{\beta}^\top \boldsymbol{u}_i)^2, \tag{27}$$

where $\boldsymbol{u}_i$ and $\lambda_i$ are the eigenvectors and eigenvalues of the covariance matrix of the input data, $\boldsymbol{\Sigma} = \sum_{i=1}^d \lambda_i \boldsymbol{u}_i \boldsymbol{u}_i^\top$, and $\kappa(\lambda)$ is the unique positive solution of:

$$1 = \frac{\lambda}{\kappa(\lambda)} + \frac{1}{n} \sum_{i=1}^d \frac{\lambda_i}{\kappa(\lambda) + \lambda_i}. \tag{28}$$

The final generalization error for the minimum $\ell_2$-norm solution is then determined by the limit $\hat{R}_0 = \lim_{\lambda \to 0^+} \hat{R}_\lambda$. By specializing this framework to the spiked covariance model, we can explicitly quantify the impact of data anisotropy on the generalization error:

**Theorem 4.1.** *Assume the high-dimensional limit $d, n \to \infty$ with $d/n \to \gamma > 1$, and let $\boldsymbol{\Sigma} = \sigma^2(\boldsymbol{I} + \rho\boldsymbol{\mu}\boldsymbol{\mu}^\top)$ and $A = \boldsymbol{\mu}^\top \boldsymbol{\beta}$. Then, for a two-layer linear network trained via gradient flow (Equations (6) and (7)) under the initialization scale $s \to 0$, the generalization error converges almost surely to:*

$$\hat{R}_0 = \sigma^2 \left( 1 - \frac{1}{\gamma} \right) \left[ 1 - A^2 \left( 1 - \frac{1+\rho}{(1+\rho/\gamma)^2} \right) \right]. \tag{29}$$

*Proof sketch.* The spiked covariance matrix $\boldsymbol{\Sigma}$ has a pricipal eigenvalue $\lambda_1 = \sigma^2(1+\rho)$ corresponding to the spike direction $\boldsymbol{u}_1 = \boldsymbol{\mu}$, while the remaining $d-1$ eigenvalues are $\lambda_i = \sigma^2$. Thus, the spectral distribution of $\boldsymbol{\Sigma}$ is $\frac{1}{d}\delta_{\sigma^2(1+\rho)} + \frac{d-1}{d}\delta_{\sigma^2}$.

In the high-dimensional limit where $n, d \to \infty$ with $d/n \to \gamma$, the contribution of the single spike to the sum $\frac{1}{n} \sum_{i=1}^d \frac{\lambda_i}{\kappa(\lambda) + \lambda_i} = \frac{\gamma}{d} \sum_{i=1}^d \frac{\lambda_i}{\kappa(\lambda) + \lambda_i}$ vanishes, as its weight $1/d$ becomes negligible. Consequently, the sum is dominated by the $d-1$ isotropic eigenvalues $\sigma^2$. The implicit equation for $\kappa(\lambda)$ then asymptotically reduces to:

$$1 = \frac{\lambda}{\kappa(\lambda)} + \gamma \frac{\sigma^2}{\kappa(\lambda) + \sigma^2}. \tag{30}$$

Taking the limit $\lambda \to 0^+$ and differentiating the implicit equation with respect to $\lambda$, we obtain

$$\lim_{\lambda \to 0^+} \kappa(\lambda) = \sigma^2(\gamma - 1) \quad \text{and} \quad \lim_{\lambda \to 0^+} \frac{\partial \kappa(\lambda)}{\partial \lambda} = \frac{\gamma}{\gamma - 1}. \tag{31}$$

Now, since $A = \boldsymbol{\mu}^\top \boldsymbol{\beta}$, we can decompose the target vector as $\boldsymbol{\beta} = A\boldsymbol{\mu} + \sqrt{1 - A^2}\boldsymbol{u}_2$, where we choose $\boldsymbol{u}_2 = (\boldsymbol{I} - \boldsymbol{\mu}\boldsymbol{\mu}^\top)\boldsymbol{\beta}/\|(\boldsymbol{I} - \boldsymbol{\mu}\boldsymbol{\mu}^\top)\boldsymbol{\beta}\|$ as one of the eigenvectors of $\boldsymbol{\Sigma}$ orthogonal to $\boldsymbol{\mu}$. Substituting this decomposition, along with Equation (31), into Equation (27) gives the generalization error. $\square$

The full proof of Theorem 4.1 can be found in Appendix D. Note that this expression with $\rho = 0$ recovers previous results for isotropic data (Mei & Montanari, 2022; Belkin et al., 2020). To better assess the performance of the model relative to the signal strength, we evaluate the normalized generalization error, which is the error $\hat{R}_0$ divided by the variance of the true signal, $\mathbb{E}[y^2] = \boldsymbol{\beta}^\top \boldsymbol{\Sigma} \boldsymbol{\beta} = \sigma^2(1 + \rho A^2)$. Our analysis predicts that this normalized error $\hat{R}_0/\mathbb{E}[y^2]$ is minimized when the spike magnitude $\rho$ is large and spike-target alignment $A$ is strong. This indicates that a pronounced anisotropy of the data structure significantly improves generalization if the anisotropy, represented by the spike, is suitably aligned with the direction of the target vector of the task. Conversely, when the target signal is orthogonal to the spike ($A = 0$), the spike provides no benefit to generalization performance. This overall theoretical dependence of the normalized error on both spike magnitude and spike-target alignment is shown in Figure 4.

To validate these theoretical predictions, we conducted numerical simulations of a two-layer linear network with small initialization, illustrated in Figure 5. The results show a close match between the empirical normalized generalization errors and the theoretical predictions across a range of parameters.

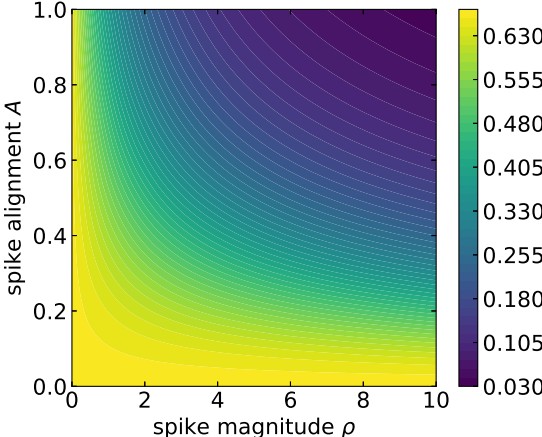

Figure 4: Normalized generalization error $\hat{R}_0/\boldsymbol{\beta}^\top \boldsymbol{\Sigma} \boldsymbol{\beta}$ as a function of spike-target alignment $A$ and spike magnitude $\rho$ for a fixed ratio $\gamma = 3$.

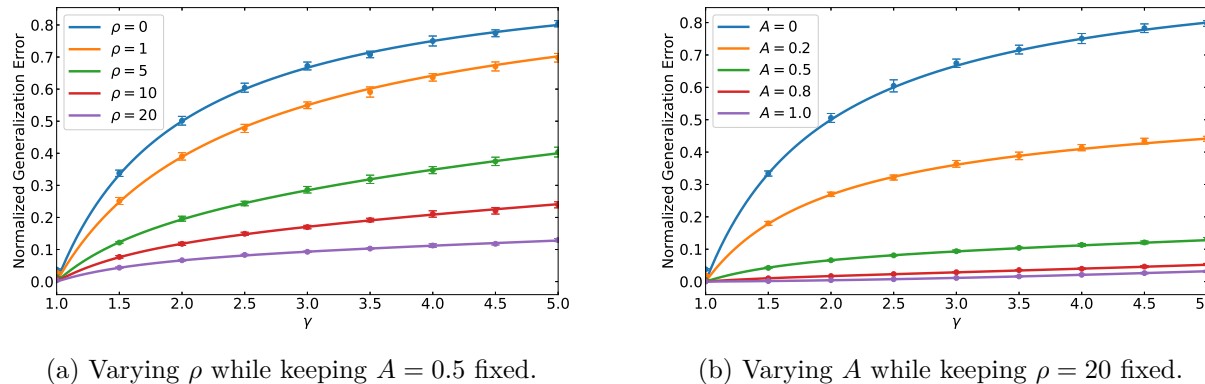

(a) Varying $\rho$ while keeping $A = 0.5$ fixed.  (b) Varying $A$ while keeping $\rho = 20$ fixed.

Figure 5: Normalized generalization error $\hat{R}_0/\boldsymbol{\beta}^\top \boldsymbol{\Sigma} \boldsymbol{\beta}$ as a function of the ratio $\gamma = d/n$. Solid lines represent the theoretical prediction (Equation (29)) and points are results of numerical simulations. The empirical results are averaged over 10 independent trials, with error bars indicating the standard deviation. All simulations were performed with a two-layer linear network of input dimension $d = 3000$ and width $m = 3000$, with $\sigma^2 = 1$. The initialization procedure is the same as in Figure 1.

# 5 Discussion and Conclusion

In this work, we investigated the learning dynamics of two-layer linear networks trained on data with a spiked covariance structure and analyzed the resulting generalization error. Theoretical analysis revealed several key insights into how data anisotropy shapes the learning process and generalization.

First, we characterized the learning dynamics, revealing two distinct phases. The early phase involves rapid weight growth in the direction of the input-output correlation $\boldsymbol{v}_1$. The speed of this phase, proportional to $\|\boldsymbol{\Sigma}_{xy}\|$, increases with both a larger spike magnitude and a stronger spike-target alignment. Once this component saturates, the later phase begins. During this phase, learning along the direction $\boldsymbol{v}_2$, which is defined within the span of $\{\boldsymbol{\Sigma}_{xy}, \boldsymbol{\beta}\}$ and orthogonal to $\boldsymbol{v}_1$, becomes significant. Its growth rate is governed by $\lambda_2 = \boldsymbol{v}_2^\top \boldsymbol{\Sigma} \boldsymbol{v}_2$ under the condition that the component in the $\boldsymbol{v}_1$ direction decreases during the later phase. The analysis of the dynamics in this data-adapted basis reveals a mechanism behind these two phases. By decomposing weight evolution along these axes, we obtain a reduced-dimensionality model that clearly explains how the network learns features in phases.

Second, we derived an analytical expression for the generalization error in the high-dimensional regime where $d, n \to \infty$ with $d/n = \gamma > 1$. The expression (Equation (29)) quantifies the dependence of the error on the spike magnitude and the spike-target alignment. Our analysis shows that a larger, well-aligned spike leads to lower generalization error, confirming the benefit of prominent, task-relevant data features. This result extends previous work on isotropic data by quantifying the impact of the spiked covariance structure, a fundamental model of data anisotropy.

Our findings connect to and extend several lines of prior research. While the dynamics of linear networks on isotropic data are well-understood (Fukumizu, 1998; Saxe et al., 2013; 2019; Braun et al., 2022; Dominé et al., 2024), and the initial alignment phase under small initialization has been previously identified (Atanasov et al., 2022), our work investigates the later phases of learning under a specific anisotropic model. This provides a more complete picture of how different components of the solution are learned over time. The generalization error analysis complements previous works (Hastie et al., 2022; Wu & Xu, 2020; Richards et al., 2021; Mel & Ganguli, 2021) by providing concrete expressions for the spiked model, making the effects of anisotropy explicit.

However, our study has several limitations. The analysis is restricted to two-layer linear networks, and the spiked covariance model represents only a specific form of anisotropy. Additionally, the theory developed here employs small initialization, leaving the effects of varying initialization scales between layers unexplored. Furthermore, the dynamics analysis assumes a finite dimension $d$ with an infinite number of samples ($n \to \infty$), whereas the generalization analysis is performed in the high-dimensional limit where $d, n \to \infty$ with their ratio held constant.

These limitations suggest several directions for future research. A direct extension of our work would involve generalizing the analysis to non-linear activations and deeper architectures. Based on our findings, another valuable direction would be to investigate learning dynamics under more complex anisotropic structures, such as multiple spikes or power-law spectral densities that are ubiquitous in natural datasets. Moreover, reconciling our dynamics and generalization analyses within a unified theoretical framework remains a significant problem. Finally, a crucial next step is to explore practical strategies, such as initialization schemes, learning rate schedules, and optimization algorithms, that explicitly exploit data anisotropy to achieve faster convergence and lower generalization error.

In conclusion, this work provides a theoretical perspective on the relationship between data anisotropy, learning dynamics, and generalization in linear networks. By analyzing how a simple anisotropic structure influences learning phases and the final performance, we contribute to a deeper understanding of the mechanisms that shape learning in structured data environments.

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

# A  Derivation of Decomposed Dynamics and Covariance Components

## A.1  Derivation of the Decomposed Dynamics

In this subsection, we provide the derivation of the decomposed dynamics (Equations (9) to (12)) used in the main text.

Let $\widetilde{\boldsymbol{\Sigma}}_{xy} = \boldsymbol{V}^\top \boldsymbol{\Sigma}_{xy}$. Then, by construction of the basis $\boldsymbol{V}$, we have $\boldsymbol{v}_1 = \boldsymbol{\Sigma}_{xy}/\|\boldsymbol{\Sigma}_{xy}\|$ and hence $\boldsymbol{\Sigma}_{xy}^\top \boldsymbol{v}_i = 0$ for $i = 2, \ldots, d$. Therefore,

$$\boldsymbol{\Sigma}_{xy}^\top \boldsymbol{V} = (\boldsymbol{\Sigma}_{xy}^\top \boldsymbol{v}_1, \boldsymbol{\Sigma}_{xy}^\top \boldsymbol{v}_2, \ldots, \boldsymbol{\Sigma}_{xy}^\top \boldsymbol{v}_d) = (\|\boldsymbol{\Sigma}_{xy}\|, 0, \ldots, 0). \tag{A.1}$$

Let $\widetilde{\boldsymbol{\Sigma}} := \boldsymbol{V}^\top \boldsymbol{\Sigma} \boldsymbol{V}$. Since the vectors $\{\boldsymbol{v}_3, \ldots, \boldsymbol{v}_d\}$ span the subspace orthogonal to $\text{span}\{\boldsymbol{\Sigma}_{xy}, \boldsymbol{\beta}\}$, and this subspace is an eigenspace of the spiked covariance matrix with eigenvalue $\sigma^2$, we have

$$\widetilde{\boldsymbol{\Sigma}} = \begin{pmatrix} \lambda_1 & -\nu & 0 & \cdots & 0 \\ -\nu & \lambda_2 & 0 & \cdots & 0 \\ 0 & 0 & \sigma^2 & & 0 \\ \vdots & \vdots & & \ddots & \vdots \\ 0 & 0 & 0 & \cdots & \sigma^2 \end{pmatrix}, \tag{A.2}$$

where $\lambda_1 = \boldsymbol{v}_1^\top \boldsymbol{\Sigma} \boldsymbol{v}_1$, $\lambda_2 = \boldsymbol{v}_2^\top \boldsymbol{\Sigma} \boldsymbol{v}_2$, and $\nu = -\boldsymbol{v}_1^\top \boldsymbol{\Sigma} \boldsymbol{v}_2$.

We also define

$$\widetilde{\boldsymbol{W}} := \boldsymbol{W} \boldsymbol{V} = (\boldsymbol{w}_1, \boldsymbol{w}_2, \ldots, \boldsymbol{w}_d). \tag{A.3}$$

Multiplying Equation (6) on the right by $\boldsymbol{V}$ and using $\boldsymbol{W} = \widetilde{\boldsymbol{W}} \boldsymbol{V}^\top$, $\boldsymbol{\Sigma}_{xy} = \boldsymbol{V} \widetilde{\boldsymbol{\Sigma}}_{xy}$, and $\boldsymbol{\Sigma} = \boldsymbol{V} \widetilde{\boldsymbol{\Sigma}} \boldsymbol{V}^\top$ yields

$$\begin{aligned} \tau \frac{d}{dt} \widetilde{\boldsymbol{W}} &= \tau \frac{d}{dt} (\boldsymbol{W} \boldsymbol{V}) \\ &= \boldsymbol{a} \left( \boldsymbol{\Sigma}_{xy}^\top - \boldsymbol{a}^\top \boldsymbol{W} \boldsymbol{\Sigma} \right) \boldsymbol{V} \\ &= \boldsymbol{a} \left( \boldsymbol{\Sigma}_{xy}^\top \boldsymbol{V} - \boldsymbol{a}^\top \widetilde{\boldsymbol{W}} \boldsymbol{V}^\top \boldsymbol{\Sigma} \boldsymbol{V} \right) \\ &= \boldsymbol{a} \left( \widetilde{\boldsymbol{\Sigma}}_{xy}^\top - \boldsymbol{a}^\top \widetilde{\boldsymbol{W}} \widetilde{\boldsymbol{\Sigma}} \right). \end{aligned} \tag{A.4}$$

Since $\widetilde{\boldsymbol{W}} = (\boldsymbol{w}_1, \boldsymbol{w}_2, \ldots, \boldsymbol{w}_d)$, taking the $i$-th column gives

$$\tau \frac{d}{dt} \boldsymbol{w}_i = \boldsymbol{a} \left( (\widetilde{\boldsymbol{\Sigma}}_{xy})_i - (\widetilde{\boldsymbol{\Sigma}} \widetilde{\boldsymbol{W}}^\top \boldsymbol{a})_i \right), \tag{A.5}$$

where $(\widetilde{\boldsymbol{\Sigma}}_{xy})_i$ and $(\widetilde{\boldsymbol{\Sigma}} \widetilde{\boldsymbol{W}}^\top \boldsymbol{a})_i$ denote the $i$-th entries of $\widetilde{\boldsymbol{\Sigma}}_{xy}$ and $\widetilde{\boldsymbol{\Sigma}} \widetilde{\boldsymbol{W}}^\top \boldsymbol{a}$, respectively.

Using $\widetilde{\boldsymbol{\Sigma}}_{xy} = (\|\boldsymbol{\Sigma}_{xy}\|, 0, \ldots, 0)^\top$ and $\widetilde{\boldsymbol{W}}^\top \boldsymbol{a} = (\boldsymbol{a}^\top \boldsymbol{w}_1, \ldots, \boldsymbol{a}^\top \boldsymbol{w}_d)^\top$, together with the block structure of $\widetilde{\boldsymbol{\Sigma}}$, we obtain Equations (9) to (11):

$$\tau \frac{d}{dt} \boldsymbol{w}_1 = \|\boldsymbol{\Sigma}_{xy}\| \boldsymbol{a} - (\lambda_1 \boldsymbol{a}^\top \boldsymbol{w}_1 - \nu \boldsymbol{a}^\top \boldsymbol{w}_2) \boldsymbol{a}, \tag{A.6}$$

$$\tau \frac{d}{dt} \boldsymbol{w}_2 = -(-\nu \boldsymbol{a}^\top \boldsymbol{w}_1 + \lambda_2 \boldsymbol{a}^\top \boldsymbol{w}_2) \boldsymbol{a}, \tag{A.7}$$

$$\tau \frac{d}{dt} \boldsymbol{w}_i = -\sigma^2 \boldsymbol{a}^\top \boldsymbol{w}_i \boldsymbol{a}, \qquad (i = 3, \ldots, d). \tag{A.8}$$

Similarly, using Equation (7), we obtain

$$\begin{aligned} \tau \frac{d}{dt} \boldsymbol{a}^\top &= \left( \boldsymbol{\Sigma}_{xy}^\top - \boldsymbol{a}^\top \boldsymbol{W} \boldsymbol{\Sigma} \right) \boldsymbol{W}^\top \\ &= \left( \widetilde{\boldsymbol{\Sigma}}_{xy}^\top \boldsymbol{V}^\top - \boldsymbol{a}^\top \widetilde{\boldsymbol{W}} \boldsymbol{V}^\top \boldsymbol{V} \widetilde{\boldsymbol{\Sigma}} \boldsymbol{V}^\top \right) \boldsymbol{V} \widetilde{\boldsymbol{W}}^\top \\ &= \left( \widetilde{\boldsymbol{\Sigma}}_{xy}^\top - \boldsymbol{a}^\top \widetilde{\boldsymbol{W}} \widetilde{\boldsymbol{\Sigma}} \right) \widetilde{\boldsymbol{W}}^\top \end{aligned} \tag{A.9}$$

Taking the transpose and expanding in terms of $\widetilde{\boldsymbol{W}} = (\boldsymbol{w}_1, \ldots, \boldsymbol{w}_d)$ gives

$$\tau \frac{d}{dt} \boldsymbol{a} = \widetilde{\boldsymbol{W}} \left( \widetilde{\boldsymbol{\Sigma}}_{xy} - \widetilde{\boldsymbol{\Sigma}} \widetilde{\boldsymbol{W}}^\top \boldsymbol{a} \right) = \sum_{i=1}^{d} \boldsymbol{w}_i \left( (\widetilde{\boldsymbol{\Sigma}}_{xy})_i - (\widetilde{\boldsymbol{\Sigma}} \widetilde{\boldsymbol{W}}^\top \boldsymbol{a})_i \right). \tag{A.10}$$

Substituting as before, we obtain Equation (12):

$$\tau \frac{d}{dt} \boldsymbol{a} = \|\boldsymbol{\Sigma}_{xy}\| \boldsymbol{w}_1 - (\lambda_1 \boldsymbol{a}^\top \boldsymbol{w}_1 - \nu \boldsymbol{a}^\top \boldsymbol{w}_2) \boldsymbol{w}_1 - (-\nu \boldsymbol{a}^\top \boldsymbol{w}_1 + \lambda_2 \boldsymbol{a}^\top \boldsymbol{w}_2) \boldsymbol{w}_2 - \sum_{i=3}^{d} \sigma^2 \boldsymbol{a}^\top \boldsymbol{w}_i \boldsymbol{w}_i. \tag{A.11}$$

## A.2 Derivation of Input-Output Correlation and Input Covariance Components

In this subsection, we provide detailed derivations for the quantities related to the correlation between the spike and the target signal, discussed in the main text. We use the notation $A = \boldsymbol{\mu}^\top \boldsymbol{\beta}$.

First, we calculate the norm of the input-output correlation $\boldsymbol{\Sigma}_{xy}$. Using Equation (1) and Equation (3), we can derive it as follows:

$$\begin{aligned} \|\boldsymbol{\Sigma}_{xy}\| &= \|\sigma^2 (\boldsymbol{I} + \rho \boldsymbol{\mu} \boldsymbol{\mu}^\top) \boldsymbol{\beta}\| \\ &= \sigma^2 \sqrt{1 + 2\rho A^2 + \rho^2 A^2} \\ &= \sigma^2 \sqrt{1 + ((1+\rho)^2 - 1) A^2}. \end{aligned} \tag{A.12}$$

Next, we calculate the components of the input covariance matrix $\boldsymbol{\Sigma}$ in the basis $\{\boldsymbol{v}_1, \boldsymbol{v}_2\}$. These components are the diagonal terms $\lambda_1 = \boldsymbol{v}_1^\top \boldsymbol{\Sigma} \boldsymbol{v}_1$ and $\lambda_2 = \boldsymbol{v}_2^\top \boldsymbol{\Sigma} \boldsymbol{v}_2$, and the off-diagonal term $\nu = -\boldsymbol{v}_1^\top \boldsymbol{\Sigma} \boldsymbol{v}_2$. To compute them, we first need the inner products of the spike direction $\boldsymbol{\mu}$ with the basis vectors.

The inner product between $\boldsymbol{\mu}$ and $\boldsymbol{v}_1$ is computed as follows:

$$\begin{aligned} \boldsymbol{\mu}^\top \boldsymbol{v}_1 &= \frac{1}{\|\boldsymbol{\Sigma}_{xy}\|} \sigma^2 \boldsymbol{\mu}^\top (\boldsymbol{I} + \rho \boldsymbol{\mu} \boldsymbol{\mu}^\top) \boldsymbol{\beta} \\ &= \frac{(1+\rho) A}{\sqrt{1 + ((1+\rho)^2 - 1) A^2}}. \end{aligned} \tag{A.13}$$

Since $\|\boldsymbol{\mu}\| = 1$ and $\boldsymbol{v}_1$ and $\boldsymbol{v}_2$ form an orthonormal basis for the relevant subspace, we have $(\boldsymbol{\mu}^\top \boldsymbol{v}_1)^2 + (\boldsymbol{\mu}^\top \boldsymbol{v}_2)^2 = 1$. The inner product of $\boldsymbol{\mu}$ with $\boldsymbol{v}_2$ is then calculated as:

$$
\begin{aligned}
\boldsymbol{\mu}^\top \boldsymbol{v}_2 &= -\sqrt{1 - (\boldsymbol{\mu}^\top \boldsymbol{v}_1)^2} \\
&= -\sqrt{1 - \frac{(1+\rho)^2 A^2}{1 + ((1+\rho)^2 - 1)A^2}}.
\end{aligned}
\tag{A.14}
$$

Using these inner products, we can now calculate the components of the covariance matrix. The first diagonal component, $\lambda_1$, is:

$$
\begin{aligned}
\lambda_1 &= \boldsymbol{v}_1^\top \boldsymbol{\Sigma} \boldsymbol{v}_1 \\
&= \sigma^2 \boldsymbol{v}_1^\top (\boldsymbol{I} + \rho \boldsymbol{\mu} \boldsymbol{\mu}^\top) \boldsymbol{v}_1 \\
&= \sigma^2 (1 + \rho(\boldsymbol{\mu}^\top \boldsymbol{v}_1)^2) \\
&= \sigma^2 \left[ 1 + \rho \frac{(1+\rho)^2 A^2}{1 + ((1+\rho)^2 - 1)A^2} \right].
\end{aligned}
\tag{A.15}
$$

Similarly, the second diagonal component, $\lambda_2$, is:

$$
\begin{aligned}
\lambda_2 &= \boldsymbol{v}_2^\top \boldsymbol{\Sigma} \boldsymbol{v}_2 \\
&= \sigma^2 \boldsymbol{v}_2^\top (\boldsymbol{I} + \rho \boldsymbol{\mu} \boldsymbol{\mu}^\top) \boldsymbol{v}_2 \\
&= \sigma^2 (1 + \rho(\boldsymbol{\mu}^\top \boldsymbol{v}_2)^2) \\
&= \sigma^2 \left[ 1 + \rho \left( 1 - \frac{(1+\rho)^2 A^2}{1 + ((1+\rho)^2 - 1)A^2} \right) \right].
\end{aligned}
\tag{A.16}
$$

The off-diagonal term, $\nu$, is calculated as follows:

$$
\begin{aligned}
\nu &= -\boldsymbol{v}_1^\top \boldsymbol{\Sigma} \boldsymbol{v}_2 \\
&= -\sigma^2 \boldsymbol{v}_1^\top (\boldsymbol{I} + \rho \boldsymbol{\mu} \boldsymbol{\mu}^\top) \boldsymbol{v}_2 \\
&= -\sigma^2 (\boldsymbol{v}_1^\top \boldsymbol{v}_2 + \rho(\boldsymbol{\mu}^\top \boldsymbol{v}_1)(\boldsymbol{\mu}^\top \boldsymbol{v}_2)) \\
&= -\sigma^2 \rho (\boldsymbol{\mu}^\top \boldsymbol{v}_1)(\boldsymbol{\mu}^\top \boldsymbol{v}_2) \\
&= \sigma^2 \rho \frac{(1+\rho)A\sqrt{1 - A^2}}{1 + ((1+\rho)^2 - 1)A^2}.
\end{aligned}
\tag{A.17}
$$

## B    Dimensionality Reduction to a Two-Variable System

**Assumption on the Initialization.**    Throughout our analysis, we assume that the parameters are initialized at scale $s$, where $s \ll 1$. Specifically, to ensure that the learning dynamics escape the unstable saddle point at the origin, we focus on non-degenerate initializations where the parameters satisfy the following bounds for some positive constants $\underline{c}$ and $\bar{c}$ independent of $s$:

$$
\underline{c}s \le \|\boldsymbol{a}(0)\| \le \bar{c}s, \quad \underline{c}s \le \|\boldsymbol{w}_i(0)\| \le \bar{c}s \quad (i = 1, \dots, d).
\tag{B.1}
$$

Also we assume $\boldsymbol{a}(0) \ne \boldsymbol{w}_1(0)$.

Under Gaussian initialization with standard deviation $s$, these condition holds almost surely.

**Overview of the Dimensionality Reduction Strategy.**    It is challenging to prove that the approximation error between the full system and the reduced system remains $O(s)$ at all times. This is because the early phase lasts $t = O(\log(1/s))$. Naively applying Gronwall's inequality here yields a divergent bound of

$O(s \exp(O(\log(1/s)))) = O(1)$. To avoid this exponential blow-up, we decouple the growing and decaying modes.

To this end, we introduce the variables $\boldsymbol{u}$ and $\boldsymbol{v}$, which correspond to the growing and decaying modes, respectively, of the linearized $(\boldsymbol{a}, \boldsymbol{w}_1)$ subsystem:

$$\boldsymbol{u} := \frac{\boldsymbol{a} + \boldsymbol{w}_1}{2}, \quad \boldsymbol{v} := \frac{\boldsymbol{a} - \boldsymbol{w}_1}{2}, \tag{B.2}$$

so that $\boldsymbol{a} = \boldsymbol{u} + \boldsymbol{v}$ and $\boldsymbol{w}_1 = \boldsymbol{u} - \boldsymbol{v}$. By the assumption on the initialization, there exist positive constants $c_{u0}$ and $c_{v0}$ such that $\|\boldsymbol{u}(0)\| = c_{u0}s$ and $\|\boldsymbol{v}(0)\| = c_{v0}s$.

From Equations (9) and (12), we obtain

$$\tau \frac{d}{dt}\boldsymbol{u} = \|\boldsymbol{\Sigma}_{xy}\|\boldsymbol{u} + \boldsymbol{e}_u, \tag{B.3}$$

$$\tau \frac{d}{dt}\boldsymbol{v} = -\|\boldsymbol{\Sigma}_{xy}\|\boldsymbol{v} + \boldsymbol{e}_v, \tag{B.4}$$

where

$$\boldsymbol{e}_u := -(\lambda_1 \boldsymbol{a}^\top \boldsymbol{w}_1 - \nu \boldsymbol{a}^\top \boldsymbol{w}_2)\boldsymbol{u} - \frac{1}{2}(-\nu \boldsymbol{a}^\top \boldsymbol{w}_1 + \lambda_2 \boldsymbol{a}^\top \boldsymbol{w}_2)\boldsymbol{w}_2 - \frac{1}{2}\sum_{i=3}^{d}\sigma^2(\boldsymbol{a}^\top \boldsymbol{w}_i)\boldsymbol{w}_i, \tag{B.5}$$

$$\boldsymbol{e}_v := (\lambda_1 \boldsymbol{a}^\top \boldsymbol{w}_1 - \nu \boldsymbol{a}^\top \boldsymbol{w}_2)\boldsymbol{v} - \frac{1}{2}(-\nu \boldsymbol{a}^\top \boldsymbol{w}_1 + \lambda_2 \boldsymbol{a}^\top \boldsymbol{w}_2)\boldsymbol{w}_2 - \frac{1}{2}\sum_{i=3}^{d}\sigma^2(\boldsymbol{a}^\top \boldsymbol{w}_i)\boldsymbol{w}_i. \tag{B.6}$$

Applying the variation of constants formula to Equation (B.3) and Equation (B.4), we obtain

$$\boldsymbol{u}(t) = \boldsymbol{u}(0)e^{\|\boldsymbol{\Sigma}_{xy}\|t/\tau} + \frac{1}{\tau}\int_0^t e^{\|\boldsymbol{\Sigma}_{xy}\|(t-t')/\tau}\boldsymbol{e}_u(t')\,dt', \tag{B.7}$$

$$\boldsymbol{v}(t) = \boldsymbol{v}(0)e^{-\|\boldsymbol{\Sigma}_{xy}\|t/\tau} + \frac{1}{\tau}\int_0^t e^{-\|\boldsymbol{\Sigma}_{xy}\|(t-t')/\tau}\boldsymbol{e}_v(t')\,dt'. \tag{B.8}$$

## B.1 Initial Alignment and Condensation in the Early Phase (Proof of Lemma 3.1)

In this subsection, we prove Lemma 3.1.

**Lemma B.1.** *For all $0 \le t < \frac{2\tau}{3(\|\boldsymbol{\Sigma}_{xy}\| + 2\operatorname{Tr}\boldsymbol{\Sigma})}\log\frac{1}{s}$, the maximum parameter norm defined as $X(t) := \max\{\|\boldsymbol{a}(t)\|, \|\boldsymbol{w}_1(t)\|, \|\boldsymbol{w}_2(t)\|, \ldots, \|\boldsymbol{w}_d(t)\|\}$ satisfies the following exponential bound:*

$$X(t) \le X(0)e^{(\|\boldsymbol{\Sigma}_{xy}\| + 2\operatorname{Tr}\boldsymbol{\Sigma})t/\tau}. \tag{B.9}$$

*Proof.* To analyze the growth of $X(t)$, we first establish bounds the residual error terms. To this end, we can bound terms as:

$$|\lambda_1 \boldsymbol{a}^\top \boldsymbol{w}_1 - \nu \boldsymbol{a}^\top \boldsymbol{w}_2| \le \lambda_1 \|\boldsymbol{a}\|\|\boldsymbol{w}_1\| + \nu\|\boldsymbol{a}\|\|\boldsymbol{w}_2\| \le (\lambda_1 + \nu)X(t)^2, \tag{B.10}$$

$$|-\nu \boldsymbol{a}^\top \boldsymbol{w}_1 + \lambda_2 \boldsymbol{a}^\top \boldsymbol{w}_2| \le \nu\|\boldsymbol{a}\|\|\boldsymbol{w}_1\| + \lambda_2\|\boldsymbol{a}\|\|\boldsymbol{w}_2\| \le (\nu + \lambda_2)X(t)^2, \tag{B.11}$$

$$\left|\sum_{i=3}^{d}\sigma^2(\boldsymbol{a}^\top \boldsymbol{w}_i)\right| \le \sum_{i=3}^{d}\sigma^2\|\boldsymbol{a}\|\|\boldsymbol{w}_i\| \le (d-2)\sigma^2 X(t)^2. \tag{B.12}$$

Since $\operatorname{Tr}\boldsymbol{\Sigma} = \lambda_1 + \lambda_2 + (d-2)\sigma^2$ and $\nu = |\boldsymbol{v}_1^\top \boldsymbol{\Sigma}\boldsymbol{v}_2| \le \sqrt{\lambda_1\lambda_2} \le (\lambda_1 + \lambda_2)/2$, we have $(\lambda_1 + \nu) + (\nu + \lambda_2) + (d-2)\sigma^2 \le 2\operatorname{Tr}\boldsymbol{\Sigma}$.

Applying these bounds to Equations (9) to (12) lead to the following differential inequality for the time derivative of the maximum norm:

$$\tau \frac{d}{dt}X(t) \le \|\boldsymbol{\Sigma}_{xy}\|X(t) + 2\operatorname{Tr}\boldsymbol{\Sigma}X(t)^3. \tag{B.13}$$

Now we define $T_* := \frac{2\tau}{3(\|\mathbf{\Sigma}_{xy}\|+2\operatorname{Tr}\mathbf{\Sigma})} \log \frac{1}{s}$. Suppose for contradiction that $X(t) \geq 1$ for some $t \in [0, T_*)$. Since $X(0) = \bar{c}s < 1$ and $X(t)$ is continuous, there must exist a first time $T \in (0, T_*)$ such that $X(T) = 1$ and $X(t) < 1$ for all $t \in [0, T)$. Therefore, in this interval,

$$\tau \frac{d}{dt} X(t) \leq (\|\mathbf{\Sigma}_{xy}\| + 2\operatorname{Tr}\mathbf{\Sigma})X(t). \tag{B.14}$$

Integrating this inequality, we obtain

$$X(t) \leq X(0)e^{(\|\mathbf{\Sigma}_{xy}\|+2\operatorname{Tr}\mathbf{\Sigma})t/\tau}. \tag{B.15}$$

Since $T \leq T_*$ and $X(0) = \bar{c}s$, we have

$$X(T) \leq \bar{c}se^{(\|\mathbf{\Sigma}_{xy}\|+2\operatorname{Tr}\mathbf{\Sigma})T_*/\tau} = \bar{c}se^{2\log(1/s)/3} = \bar{c}s^{1/3}. \tag{B.16}$$

For all sufficiently small $s$, we have $\bar{c}s^{1/3} < 1/2$. This implies $X(T) \leq \bar{c}s^{1/3} < 1/2$. This contradicts $X(T) = 1$. Therefore, $X(t) < 1$ for all $t \in [0, T_*)$.

Consequently, Inequality (B.14) holds on $[0, T_*)$, and hence the desired bound Inequality (B.15) also follows. $\square$

*Proof of Lemma 3.1.* By the definition of $X(t)$, we have

$$\|\mathbf{u}\| \leq \frac{1}{2}(\|\mathbf{a}\| + \|\mathbf{w}_1\|) \leq X(t), \quad \|\mathbf{v}\| \leq \frac{1}{2}(\|\mathbf{a}\| + \|\mathbf{w}_1\|) \leq X(t). \tag{B.17}$$

Therefore, we can uniformly bound the residual error terms (Equations (B.5) and (B.6)) similarly to the proof of Lemma B.1:

$$\|\mathbf{e}_u\| \leq 2\operatorname{Tr}\mathbf{\Sigma}X(t)^3, \quad \|\mathbf{e}_v\| \leq 2\operatorname{Tr}\mathbf{\Sigma}X(t)^3. \tag{B.18}$$

For $t < \frac{2\tau}{3(\|\mathbf{\Sigma}_{xy}\|+2\operatorname{Tr}\mathbf{\Sigma})} \log \frac{1}{s}$, Lemma B.1 implies

$$2\operatorname{Tr}\mathbf{\Sigma}X(t)^3 \leq C_{\mathrm{res}}s^3 e^{3(\|\mathbf{\Sigma}_{xy}\|+2\operatorname{Tr}\mathbf{\Sigma})t/\tau}, \tag{B.19}$$

where $C_{\mathrm{res}} = 2\operatorname{Tr}\mathbf{\Sigma}\bar{c}^3$. Therefore,

$$\|\mathbf{e}_u\| \leq C_{\mathrm{res}}s^3 e^{3(\|\mathbf{\Sigma}_{xy}\|+2\operatorname{Tr}\mathbf{\Sigma})t/\tau}, \quad \|\mathbf{e}_v\| \leq C_{\mathrm{res}}s^3 e^{3(\|\mathbf{\Sigma}_{xy}\|+2\operatorname{Tr}\mathbf{\Sigma})t/\tau}. \tag{B.20}$$

Substituting the bound Inequality (B.20) into the integral terms (Equations (B.7) and (B.8)), we use the upper bound on $t$, which implies $s^3 e^{3(\|\mathbf{\Sigma}_{xy}\|+2\operatorname{Tr}\mathbf{\Sigma})t/\tau} \leq s^3(s^{-2/3})^3 = s$. Therefore, for some constant $C_1 > 0$, we obtain

$$\begin{aligned}
\left\|\frac{1}{\tau}\int_0^t e^{\|\mathbf{\Sigma}_{xy}\|(t-t')/\tau}\mathbf{e}_u(t')\,dt'\right\| &\leq \frac{1}{\tau}\int_0^t e^{\|\mathbf{\Sigma}_{xy}\|(t-t')/\tau}\|\mathbf{e}_u(t')\|\,dt' \\
&\leq \frac{C_{\mathrm{res}}s^3}{\tau}e^{\|\mathbf{\Sigma}_{xy}\|t/\tau}\int_0^t e^{(3(\|\mathbf{\Sigma}_{xy}\|+2\operatorname{Tr}\mathbf{\Sigma})-\|\mathbf{\Sigma}_{xy}\|)t'/\tau}\,dt' \\
&\leq \frac{C_{\mathrm{res}}s^3}{3(\|\mathbf{\Sigma}_{xy}\|+2\operatorname{Tr}\mathbf{\Sigma})-\|\mathbf{\Sigma}_{xy}\|}e^{3(\|\mathbf{\Sigma}_{xy}\|+2\operatorname{Tr}\mathbf{\Sigma})t/\tau} \\
&\leq C_1 s,
\end{aligned} \tag{B.21}$$

$$\begin{aligned}
\left\|\frac{1}{\tau}\int_0^t e^{-\|\mathbf{\Sigma}_{xy}\|(t-t')/\tau}\mathbf{e}_v(t')\,dt'\right\| &\leq \frac{1}{\tau}\int_0^t e^{-\|\mathbf{\Sigma}_{xy}\|(t-t')/\tau}\|\mathbf{e}_v(t')\|\,dt' \\
&\leq \frac{C_{\mathrm{res}}s^3}{\tau}e^{-\|\mathbf{\Sigma}_{xy}\|t/\tau}\int_0^t e^{(3(\|\mathbf{\Sigma}_{xy}\|+2\operatorname{Tr}\mathbf{\Sigma})+\|\mathbf{\Sigma}_{xy}\|)t'/\tau}\,dt' \\
&\leq \frac{C_{\mathrm{res}}s^3}{3(\|\mathbf{\Sigma}_{xy}\|+2\operatorname{Tr}\mathbf{\Sigma})+\|\mathbf{\Sigma}_{xy}\|}e^{3(\|\mathbf{\Sigma}_{xy}\|+2\operatorname{Tr}\mathbf{\Sigma})t/\tau} \\
&\leq C_1 s.
\end{aligned} \tag{B.22}$$

By substituting the initial condition $\boldsymbol{u}(0) = \boldsymbol{r}_1$ and Inequality (B.21) into Equation (B.7), we get

$$\|\boldsymbol{u}(t) - \boldsymbol{r}_1 e^{\|\boldsymbol{\Sigma}_{xy}\| t/\tau}\| \leq C_1 s. \tag{B.23}$$

Likewise, applying Inequality (B.22) to Equation (B.8), we obtain

$$\|\boldsymbol{v}(t)\| \leq \|\boldsymbol{v}(0)\| e^{-\|\boldsymbol{\Sigma}_{xy}\| t/\tau} + C_1 s \leq C_2 s. \tag{B.24}$$

Finally, since $\boldsymbol{a} = \boldsymbol{u} + \boldsymbol{v}$ and $\boldsymbol{w}_1 = \boldsymbol{u} - \boldsymbol{v}$, applying the triangle inequality yields:

$$\|\boldsymbol{w}_1(t) - \boldsymbol{r}_1 e^{\|\boldsymbol{\Sigma}_{xy}\| t/\tau}\| \leq \|\boldsymbol{u}(t) - \boldsymbol{r}_1 e^{\|\boldsymbol{\Sigma}_{xy}\| t/\tau}\| + \|\boldsymbol{v}(t)\| \leq (C_1 + C_2)s, \tag{B.25}$$

$$\|\boldsymbol{a}(t) - \boldsymbol{r}_1 e^{\|\boldsymbol{\Sigma}_{xy}\| t/\tau}\| \leq \|\boldsymbol{u}(t) - \boldsymbol{r}_1 e^{\|\boldsymbol{\Sigma}_{xy}\| t/\tau}\| + \|\boldsymbol{v}(t)\| \leq (C_1 + C_2)s. \tag{B.26}$$

This proves the claim with $C = C_1 + C_2$.

$\square$

## B.2 Auxiliary Integral Bound

The following lemma will be used repeatedly in the next two subsections.

**Lemma B.2.** *Let $x(t) > 0$ satisfy $\tau \frac{d}{dt} x(t) \geq \gamma x(t)$ for some $\gamma > 0$. Let $M_K > 0$, $p > 0$, and $\beta \neq p\gamma$. If $K(t, t') \leq M_K e^{\beta(t-t')/\tau}$, then*

$$\frac{1}{\tau} \int_0^t K(t, t') x(t')^p \, dt' \leq M_K \begin{cases} \frac{x(t)^{\beta/\gamma} x(0)^{p-\beta/\gamma}}{\beta - \gamma p} & (\beta/\gamma > p), \\ \frac{x(t)^p}{\gamma p - \beta} & (0 < \beta/\gamma < p), \\ \frac{x(t)^p}{\gamma p} & (\beta \leq 0). \end{cases} \tag{B.27}$$

*Proof.* After rescaling $K$, it suffices to treat the case $M_K = 1$. Since $\tau \frac{d}{dt} x \geq \gamma x$, we have $x(t) \geq x(t') e^{\gamma(t-t')/\tau}$ for $0 \leq t' \leq t$. Hence, for $\beta > 0$, $K(t, t') \leq e^{\beta(t-t')/\tau} \leq \left( \frac{x(t)}{x(t')} \right)^{\beta/\gamma}$. Using also $\tau \frac{d}{dt} x \geq \gamma x$,

$$\begin{aligned} \frac{1}{\tau} \int_0^t K(t, t') x(t')^p \, dt' &\leq \frac{1}{\gamma} \int_{x(0)}^{x(t)} \left( \frac{x(t)}{x} \right)^{\beta/\gamma} x^{p-1} \, dx \\ &\leq \frac{x(t)^{\beta/\gamma}}{\gamma} \int_{x(0)}^{x(t)} x^{p-1-\beta/\gamma} \, dx \\ &\leq \begin{cases} \frac{x(t)^{\beta/\gamma} x(0)^{p-\beta/\gamma}}{\beta - \gamma p} & (\beta/\gamma > p), \\ \frac{x(t)^p}{\gamma p - \beta} & (\beta/\gamma < p) \end{cases} \end{aligned} \tag{B.28}$$

If $\beta \leq 0$, then $K(t, t') \leq e^{\beta(t-t')/\tau} \leq 1$, so

$$\frac{1}{\tau} \int_0^t K(t, t') x(t')^p \, dt' \leq \frac{1}{\gamma} \int_{x(0)}^{x(t)} x^{p-1} \, dx \leq \frac{x(t)^p}{\gamma p}. \tag{B.29}$$

$\square$

## B.3 Global Boundedness of Orthogonal Components (Proof of Lemma 3.2)

This subsection establishes that the learning dynamics remain effectively confined to a low-dimensional subspace.

### B.3.1 Preliminaries

We now establish the local bounds needed for the main argument, up to an exit time defined by the weight magnitudes.

Before analyzing the principal components, we briefly note the behavior of the weights $\boldsymbol{w}_i$ for $i \geq 3$. Taking the inner product with $\boldsymbol{w}_i$ on both sides of Equation (11) yields $\tau \frac{d}{dt} \|\boldsymbol{w}_i\|^2 \leq -2\sigma^2 (\boldsymbol{a}^\top \boldsymbol{w}_i)^2 \leq 0$. Given the initialization $\|\boldsymbol{w}_i(0)\| \leq \bar{c}s$, it immediately follows that $\|\boldsymbol{w}_i(t)\| \leq \bar{c}s$ for all $t > 0$. Therefore, we focus on the remaining components.

Before proceeding to the main local analysis up to the exit time, we examine the behavior of $\boldsymbol{u}$ and $\boldsymbol{v}$ in the earliest stage of learning. Lemma B.3 establishes that $\boldsymbol{u}$ eventually surpasses $\boldsymbol{v}$ regardless of the relative magnitudes of the initial weights.

**Lemma B.3.** *For all sufficiently small $s > 0$, if $t^* = \frac{\tau}{2\|\boldsymbol{\Sigma}_{xy}\|} \log \left( \frac{c_{v0}}{c_{u0}} + 2 \right)$, then the following inequality holds:*

$$\|\boldsymbol{v}(t^*)\| \leq \|\boldsymbol{u}(t^*)\|. \tag{B.30}$$

*Proof.* Since $t^*$ is independent of $s$, we have

$$t^* < \frac{2\tau}{3(\|\boldsymbol{\Sigma}_{xy}\| + 2 \operatorname{Tr} \boldsymbol{\Sigma})} \log \frac{1}{s} \tag{B.31}$$

for all sufficiently small $s$. Applying the bound from Lemma B.1 on the interval $[0, t^*]$, we obtain

$$X(t) \leq X(0) e^{(\|\boldsymbol{\Sigma}_{xy}\| + 2 \operatorname{Tr} \boldsymbol{\Sigma})t/\tau} \leq C_X s, \tag{B.32}$$

for some constant $C_X > 0$. Because the residuals $\boldsymbol{e}_u(t)$ and $\boldsymbol{e}_v(t)$ scale as $X(t)^3$, this guarantees that their corresponding integral terms are uniformly bounded by $O(s^3)$. Specifically, there exists a constant $C_* > 0$ such that the integral bounds satisfy:

$$\left\| \int_0^{t^*} e^{\|\boldsymbol{\Sigma}_{xy}\|(t^* - t')/\tau} \boldsymbol{e}_u(t') \, dt' \right\|, \quad \left\| \int_0^{t^*} e^{-\|\boldsymbol{\Sigma}_{xy}\|(t^* - t')/\tau} \boldsymbol{e}_v(t') \, dt' \right\| \leq C_* s^3. \tag{B.33}$$

Using Equations (B.7) and (B.8), we get

$$\|\boldsymbol{u}(t^*)\| \geq e^{\|\boldsymbol{\Sigma}_{xy}\| t^*/\tau} \|\boldsymbol{u}(0)\| - C_* s^3, \tag{B.34}$$

$$\|\boldsymbol{v}(t^*)\| \leq e^{-\|\boldsymbol{\Sigma}_{xy}\| t^*/\tau} \|\boldsymbol{v}(0)\| + C_* s^3. \tag{B.35}$$

Since

$$e^{\|\boldsymbol{\Sigma}_{xy}\| t^*/\tau} = \sqrt{\frac{c_{v0}}{c_{u0}} + 2}, \qquad e^{-\|\boldsymbol{\Sigma}_{xy}\| t^*/\tau} = \frac{1}{\sqrt{\frac{c_{v0}}{c_{u0}} + 2}}, \tag{B.36}$$

and $\|\boldsymbol{u}(0)\| = c_{u0} s$, $\|\boldsymbol{v}(0)\| = c_{v0} s$, we obtain

$$\|\boldsymbol{u}(t^*)\| - \|\boldsymbol{v}(t^*)\| \geq \left( c_{u0} \sqrt{\frac{c_{v0}}{c_{u0}} + 2} - \frac{c_{v0}}{\sqrt{\frac{c_{v0}}{c_{u0}} + 2}} \right) s - 2 C_* s^3$$

$$= \frac{2 c_{u0}}{\sqrt{\frac{c_{v0}}{c_{u0}} + 2}} s - 2 C_* s^3. \tag{B.37}$$

The coefficient of $s$ is strictly positive, so for sufficiently small $s$ the $O(s)$ term dominates the $O(s^3)$ remainder. Hence $\|\boldsymbol{u}(t^*)\| \geq \|\boldsymbol{v}(t^*)\|$. $\qquad\square$

The preceding analysis shows that the network parameters remain of order $O(s)$ during this very early phase of learning. Furthermore, Lemma B.3 ensures that even if the initial weights satisfy $\|\boldsymbol{v}(0)\| > \|\boldsymbol{u}(0)\|$, there exists a time $t^*$ at which $\|\boldsymbol{u}(t^*)\| \geq \|\boldsymbol{v}(t^*)\|$, while all weights are still bounded by $O(s)$. Therefore, we may assume $\|\boldsymbol{v}(0)\| \leq \|\boldsymbol{u}(0)\|$ without loss of generality. (Note that, since $t^*$ is independent of $s$ and all weights remain $O(s)$ on $[0, t^*]$, this contribution of dynamics up to $t_*$ can be absorbed into the constants of the subsequent bounds.)

To rigorously evaluate the dynamics before the weights grow to an $O(1)$ scale, we introduce an exit time $T_\varepsilon$ as the first time the maximum norm of the weights reaches a predefined threshold $\varepsilon$:

$$T_\varepsilon := \inf\{t > 0 : \max\{\|\boldsymbol{a}(t)\|, \|\boldsymbol{w}_1(t)\|, \|\boldsymbol{w}_2(t)\|, \ldots, \|\boldsymbol{w}_d(t)\|\} \geq \varepsilon\} \tag{B.38}$$

where $\varepsilon > 0$ is a constant independent of the initialization scale $s$. The exact value of $\varepsilon$ will be chosen to be sufficiently small to ensure that the following propositions hold throughout the interval $[0, T_\varepsilon]$.

**Lemma B.4.** *There exists a constant $c_1 > 0$ such that for all $t \in [0, T_\varepsilon]$, the following inequalities hold:*

$$\|\boldsymbol{v}(t)\| \leq \|\boldsymbol{u}(t)\|, \tag{B.39}$$

$$\tau \frac{d}{dt}\|\boldsymbol{u}(t)\| \geq c_1\|\boldsymbol{u}(t)\|. \tag{B.40}$$

*Consequently, for any times $t'$ and $t$ satisfying $0 \leq t' \leq t \leq T_\varepsilon$, we have*

$$\|\boldsymbol{u}(t')\| \leq \|\boldsymbol{u}(t)\|e^{-c_1(t-t')/\tau}. \tag{B.41}$$

*Proof.* Since $\|\boldsymbol{a}\| \leq \|\boldsymbol{u}\| + \|\boldsymbol{v}\|$, the definition of $T_\varepsilon$ (Equation (B.38)) gives, for all $t \in [0, T_\varepsilon]$,

$$|\lambda_1 \boldsymbol{a}^\top \boldsymbol{w}_1 - \nu \boldsymbol{a}^\top \boldsymbol{w}_2| \leq \lambda_1\|\boldsymbol{a}\|\|\boldsymbol{w}_1\| + \nu\|\boldsymbol{a}\|\|\boldsymbol{w}_2\| \leq (\lambda_1 + \nu)\varepsilon(\|\boldsymbol{u}\| + \|\boldsymbol{v}\|), \tag{B.42}$$

$$|-\nu \boldsymbol{a}^\top \boldsymbol{w}_1 + \lambda_2 \boldsymbol{a}^\top \boldsymbol{w}_2| \leq \nu\|\boldsymbol{a}\|\|\boldsymbol{w}_1\| + \lambda_2\|\boldsymbol{a}\|\|\boldsymbol{w}_2\| \leq (\nu + \lambda_2)\varepsilon(\|\boldsymbol{u}\| + \|\boldsymbol{v}\|), \tag{B.43}$$

$$\left|\sum_{i=3}^d \sigma^2(\boldsymbol{a}^\top \boldsymbol{w}_i)\right| \leq \sum_{i=3}^d \sigma^2\|\boldsymbol{a}\|\|\boldsymbol{w}_i\| \leq (d-2)\sigma^2\varepsilon(\|\boldsymbol{u}\| + \|\boldsymbol{v}\|). \tag{B.44}$$

Hence there exists $C_0 > 0$ such that the residual terms $\boldsymbol{e}_u$ and $\boldsymbol{e}_v$ (Equations (B.5) and (B.6)) satisfy

$$\|\boldsymbol{e}_u\|, \|\boldsymbol{e}_v\| \leq \frac{1}{2}C_0\varepsilon^2(\|\boldsymbol{u}\| + \|\boldsymbol{v}\|). \tag{B.45}$$

Applying these to Equations (B.3) and (B.4) yields

$$\tau \frac{d}{dt}\|\boldsymbol{u}\| \geq \|\boldsymbol{\Sigma}_{xy}\|\|\boldsymbol{u}\| - \frac{1}{2}C_0\varepsilon^2(\|\boldsymbol{u}\| + \|\boldsymbol{v}\|), \tag{B.46}$$

$$\tau \frac{d}{dt}\|\boldsymbol{v}\| \leq -\|\boldsymbol{\Sigma}_{xy}\|\|\boldsymbol{v}\| + \frac{1}{2}C_0\varepsilon^2(\|\boldsymbol{u}\| + \|\boldsymbol{v}\|). \tag{B.47}$$

Subtracting the second inequality from the first,

$$\tau \frac{d}{dt}(\|\boldsymbol{u}\| - \|\boldsymbol{v}\|) \geq (\|\boldsymbol{\Sigma}_{xy}\| - C_0\varepsilon^2)(\|\boldsymbol{u}\| + \|\boldsymbol{v}\|). \tag{B.48}$$

Choosing $\varepsilon$ sufficiently small so that $\|\boldsymbol{\Sigma}_{xy}\| - C_0\varepsilon^2 \geq 0$, we obtain $\tau \frac{d}{dt}(\|\boldsymbol{u}\| - \|\boldsymbol{v}\|) \geq 0$ on $[0, T_\varepsilon]$. Since $\|\boldsymbol{u}(0)\| \geq \|\boldsymbol{v}(0)\|$, it follows that $\|\boldsymbol{u}(t)\| \geq \|\boldsymbol{v}(t)\|$ for all $t \in [0, T_\varepsilon]$.

Also, from Inequality (B.46),

$$\begin{aligned} \tau \frac{d}{dt}\|\boldsymbol{u}\| &\geq \|\boldsymbol{\Sigma}_{xy}\|\|\boldsymbol{u}\| - \frac{1}{2}C_0\varepsilon^2(\|\boldsymbol{u}\| + \|\boldsymbol{v}\|) \\ &\geq (\|\boldsymbol{\Sigma}_{xy}\| - C_0\varepsilon^2)\|\boldsymbol{u}\|. \end{aligned} \tag{B.49}$$

Setting

$$c_1 := \|\mathbf{\Sigma}_{xy}\| - C_0\varepsilon^2 \tag{B.50}$$

and taking $\varepsilon$ smaller if necessary, we can ensure $c_1 > 0$, which gives Inequality (B.40):

$$\tau\frac{d}{dt}\|\boldsymbol{u}(t)\| \geq c_1\|\boldsymbol{u}(t)\|. \tag{B.51}$$

Integrating from $t'$ to $t$ yields

$$\ln\left(\frac{\|\boldsymbol{u}(t)\|}{\|\boldsymbol{u}(t')\|}\right) \geq \frac{c_1}{\tau}(t - t'). \tag{B.52}$$

Rearranging gives Inequality (B.41):

$$\|\boldsymbol{u}(t')\| \leq \|\boldsymbol{u}(t)\|e^{-c_1(t-t')/\tau}. \tag{B.53}$$

$\square$

**Corollary B.5.** *For all $M > 0$, there exists a time $t_0$, independent of $s$, such that for all sufficiently small $s > 0$, for all $t \in [t_0, T_\varepsilon]$,*

$$\|\boldsymbol{u}(t)\| - \|\boldsymbol{v}(t)\| \geq Ms. \tag{B.54}$$

*Proof.* In the proof of Lemma B.4, we showed that, on $[0, T_\varepsilon]$,

$$\tau\frac{d}{dt}(\|\boldsymbol{u}(t)\| - \|\boldsymbol{v}(t)\|) \geq c_1(\|\boldsymbol{u}(t)\| + \|\boldsymbol{v}(t)\|). \tag{B.55}$$

Moreover, Inequality (B.40) and $\|\boldsymbol{u}(0)\| = c_{u0}s$ imply $\|\boldsymbol{u}(t)\| \geq c_{u0}se^{c_1 t/\tau}$ for $t \in [0, T_\varepsilon]$. Since $\|\boldsymbol{u}(0)\| \geq \|\boldsymbol{v}(0)\|$, integrating the first inequality gives, for $t \in [0, T_\varepsilon]$,

$$\begin{aligned}
\|\boldsymbol{u}(t)\| - \|\boldsymbol{v}(t)\| &\geq \|\boldsymbol{u}(0)\| - \|\boldsymbol{v}(0)\| + \frac{c_1}{\tau}\int_0^t (\|\boldsymbol{u}(t')\| + \|\boldsymbol{v}(t')\|)\,dt' \\
&\geq \frac{c_1}{\tau}\int_0^t \|\boldsymbol{u}(t')\|\,dt' \\
&\geq c_{u0}s(e^{c_1 t/\tau} - 1).
\end{aligned} \tag{B.56}$$

Choose $t_0 := \frac{\tau}{c_1}\log\left(1 + \frac{M}{c_{u0}}\right)$. Since $t_0$ is independent of $s$, Lemma B.1 gives $X(t) < \varepsilon$ on $[0, t_0]$ for all sufficiently small $s$, and hence $t_0 < T_\varepsilon$. Applying this bound for $t \in [t_0, T_\varepsilon]$ yields Inequality (B.54). $\square$

**Lemma B.6.** *There exist constants $C_{w_21}, C_{w_22} > 0$ such that for all $t \in [0, T_\varepsilon]$, the following inequality holds:*

$$\|\boldsymbol{w}_2(t)\| \leq C_{w_21}s + C_{w_22}\|\boldsymbol{u}(t)\|^3. \tag{B.57}$$

*Proof.* Using $|\boldsymbol{a}^\top \boldsymbol{w}_1| \leq \|\boldsymbol{u}\|^2$, $\|\boldsymbol{a}\| \leq 2\|\boldsymbol{u}\|$, and $|\boldsymbol{a}^\top \boldsymbol{w}_2| \leq 2\|\boldsymbol{u}\|\|\boldsymbol{w}_2\|$ in Equation (10), we obtain

$$\tau\frac{d}{dt}\|\boldsymbol{w}_2\| \leq 4\lambda_2\|\boldsymbol{u}\|^2\|\boldsymbol{w}_2\| + 2\nu\|\boldsymbol{u}\|^3. \tag{B.58}$$

Let

$$K(t, t') := \exp\left(\frac{1}{\tau}\int_{t'}^t 4\lambda_2\|\boldsymbol{u}(r)\|^2\,dr\right). \tag{B.59}$$

The variation of constants formula gives:

$$\|\boldsymbol{w}_2(t)\| \leq K(t, 0)\|\boldsymbol{w}_2(0)\| + \frac{2\nu}{\tau}\int_0^t K(t, t')\|\boldsymbol{u}(t')\|^3\,dt'. \tag{B.60}$$

By Inequality (B.41) and $\|\boldsymbol{u}(t)\| \leq \varepsilon$, for $0 \leq t' \leq t \leq T_\varepsilon$,

$$\frac{1}{\tau} \int_{t'}^{t} 4\lambda_2 \|\boldsymbol{u}(r)\|^2 \, dr \leq \frac{4\lambda_2 \|\boldsymbol{u}(t)\|^2}{\tau} \int_{t'}^{t} e^{-2c_1(t-r)/\tau} \, dr \leq \frac{2\lambda_2}{c_1} \varepsilon^2. \tag{B.61}$$

Hence $K(t, t') \leq e^{\frac{2\lambda_2}{c_1}\varepsilon^2} =: C_K$. Applying Lemma B.2 with $x(t) = \|\boldsymbol{u}(t)\|$, $\gamma = c_1$, $\beta = 0$, and $p = 3$, we obtain

$$\frac{1}{\tau} \int_0^t K(t, t') \|\boldsymbol{u}(t')\|^3 \, dt' \leq \frac{C_K}{3c_1} \|\boldsymbol{u}(t)\|^3. \tag{B.62}$$

Substituting this into Inequality (B.60) yields:

$$\|\boldsymbol{w}_2(t)\| \leq C_K \|\boldsymbol{w}_2(0)\| + \frac{2\nu C_K}{3c_1} \|\boldsymbol{u}(t)\|^3$$
$$\leq C_{w_2 1} s + C_{w_2 2} \|\boldsymbol{u}(t)\|^3 \tag{B.63}$$

for some constants $C_{w_2 1}, C_{w_2 2} > 0$. $\qquad \square$

**Corollary B.7.** *There exists a constant $C_u > 0$ such that for all $t \in [0, T_\varepsilon]$,*

$$\|\boldsymbol{v}(t)\|, \|\boldsymbol{a}(t)\|, \|\boldsymbol{w}_i(t)\| \leq C_u \|\boldsymbol{u}(t)\|, \qquad (i = 1, \ldots, d). \tag{B.64}$$

*Proof.* By Inequalities (B.39) and (B.40), we have $\|\boldsymbol{v}(t)\| \leq \|\boldsymbol{u}(t)\|$ and $\|\boldsymbol{u}(t)\| \geq \|\boldsymbol{u}(0)\| = c_{u0} s$ on $[0, T_\varepsilon]$. Hence

$$\|\boldsymbol{a}(t)\| = \|\boldsymbol{u}(t) + \boldsymbol{v}(t)\| \leq 2\|\boldsymbol{u}(t)\|, \quad \|\boldsymbol{w}_1(t)\| = \|\boldsymbol{u}(t) - \boldsymbol{v}(t)\| \leq 2\|\boldsymbol{u}(t)\|. \tag{B.65}$$

Moreover, Lemma B.6 and $\|\boldsymbol{u}(t)\| \leq \varepsilon$ yield

$$\|\boldsymbol{w}_2(t)\| \leq C_{w_2 1} s + C_{w_2 2} \|\boldsymbol{u}(t)\|^3 \leq \left( \frac{C_{w_2 1}}{c_{u0}} + C_{w_2 2} \varepsilon^2 \right) \|\boldsymbol{u}(t)\|. \tag{B.66}$$

For $i \geq 3$, the monotonicity argument from the beginning of this subsection gives

$$\|\boldsymbol{w}_i(t)\| \leq \bar{c} s \leq \frac{\bar{c}}{c_{u0}} \|\boldsymbol{u}(t)\|. \tag{B.67}$$

The claim follows by taking $C_u$ to be the maximum of the constants above. $\qquad \square$

### B.3.2 Main Proof of Lemma 3.2

*Proof of Lemma 3.2.* We aim to show that the vector $\boldsymbol{e}_\perp := ((\boldsymbol{P}_\perp \boldsymbol{u})^\top, (\boldsymbol{P}_\perp \boldsymbol{v})^\top, (\boldsymbol{P}_\perp \boldsymbol{w}_2)^\top, \ldots, (\boldsymbol{P}_\perp \boldsymbol{w}_d)^\top)^\top \in \mathbb{R}^{m(d+1)}$, which captures all components of the system orthogonal to the condensation direction $\hat{\boldsymbol{r}}_1$, remains $O(s)$ uniformly in time. Since $\boldsymbol{a} = \boldsymbol{u} + \boldsymbol{v}$ and $\boldsymbol{w}_1 = \boldsymbol{u} - \boldsymbol{v}$, controlling $\boldsymbol{P}_\perp \boldsymbol{u}$ and $\boldsymbol{P}_\perp \boldsymbol{v}$ is equivalent to controlling the orthogonal parts of both $\boldsymbol{a}$ and $\boldsymbol{w}_1$.

We introduce $\boldsymbol{\theta}(t) := \boldsymbol{W}(t)^\top \boldsymbol{a}(t)$ and $r_i(t) := \boldsymbol{v}_i^\top \boldsymbol{\Sigma}(\boldsymbol{\beta} - \boldsymbol{\theta}(t))$. Since $\boldsymbol{\Sigma}\boldsymbol{\beta} = \boldsymbol{\Sigma}_{xy} = \|\boldsymbol{\Sigma}_{xy}\|\boldsymbol{v}_1$,

$$\boldsymbol{\Sigma}\boldsymbol{v}_1 = \lambda_1 \boldsymbol{v}_1 - \nu \boldsymbol{v}_2, \quad \boldsymbol{\Sigma}\boldsymbol{v}_2 = -\nu\boldsymbol{v}_1 + \lambda_2 \boldsymbol{v}_2, \quad \boldsymbol{\Sigma}\boldsymbol{v}_i = \sigma^2 \boldsymbol{v}_i \ (i \geq 3), \tag{B.68}$$

and $\boldsymbol{\theta}(t) = \sum_{j=1}^{d} (\boldsymbol{a}^\top \boldsymbol{w}_j) \boldsymbol{v}_j$, we obtain

$$r_1(t) = \|\boldsymbol{\Sigma}_{xy}\| - \lambda_1 \boldsymbol{a}^\top \boldsymbol{w}_1 + \nu \boldsymbol{a}^\top \boldsymbol{w}_2, \quad r_2(t) = \nu \boldsymbol{a}^\top \boldsymbol{w}_1 - \lambda_2 \boldsymbol{a}^\top \boldsymbol{w}_2, \quad r_i(t) = -\sigma^2 \boldsymbol{a}^\top \boldsymbol{w}_i \ (i = 3, \ldots, d). \tag{B.69}$$

In terms of these scalar coefficients, Equations (10), (11), (B.3) and (B.4) become

$$\tau \frac{d}{dt} \boldsymbol{u} = r_1 \boldsymbol{u} + \frac{r_2}{2} \boldsymbol{w}_2 + \frac{1}{2} \sum_{i=3}^{d} r_i \boldsymbol{w}_i, \tag{B.70}$$

$$\tau \frac{d}{dt} \boldsymbol{v} = -r_1 \boldsymbol{v} + \frac{r_2}{2} \boldsymbol{w}_2 + \frac{1}{2} \sum_{i=3}^{d} r_i \boldsymbol{w}_i, \tag{B.71}$$

$$\tau \frac{d}{dt} \boldsymbol{w}_i = r_i(\boldsymbol{u} + \boldsymbol{v}), \qquad (i = 2, \ldots, d). \tag{B.72}$$

Applying the projection operator $\boldsymbol{P}_\perp$ to these equations preserves the same scalar coefficients, so the orthogonal components satisfy

$$\tau \frac{d}{dt} \boldsymbol{P}_\perp \boldsymbol{u} = r_1 \boldsymbol{P}_\perp \boldsymbol{u} + \frac{r_2}{2} \boldsymbol{P}_\perp \boldsymbol{w}_2 + \frac{1}{2} \sum_{i=3}^{d} r_i \boldsymbol{P}_\perp \boldsymbol{w}_i, \tag{B.73}$$

$$\tau \frac{d}{dt} \boldsymbol{P}_\perp \boldsymbol{v} = -r_1 \boldsymbol{P}_\perp \boldsymbol{v} + \frac{r_2}{2} \boldsymbol{P}_\perp \boldsymbol{w}_2 + \frac{1}{2} \sum_{i=3}^{d} r_i \boldsymbol{P}_\perp \boldsymbol{w}_i, \tag{B.74}$$

$$\tau \frac{d}{dt} \boldsymbol{P}_\perp \boldsymbol{w}_i = r_i (\boldsymbol{P}_\perp \boldsymbol{u} + \boldsymbol{P}_\perp \boldsymbol{v}), \qquad (i = 2, \ldots, d). \tag{B.75}$$

Stacking the above equations yields

$$\tau \frac{d}{dt} \boldsymbol{e}_\perp = \boldsymbol{R} \boldsymbol{e}_\perp, \tag{B.76}$$

where

$$\boldsymbol{R}(t) = \begin{pmatrix} r_1(t) & 0 & \frac{r_2(t)}{2} & \frac{r_3(t)}{2} & \cdots & \frac{r_d(t)}{2} \\ 0 & -r_1(t) & \frac{r_2(t)}{2} & \frac{r_3(t)}{2} & \cdots & \frac{r_d(t)}{2} \\ r_2(t) & r_2(t) & 0 & 0 & \cdots & 0 \\ r_3(t) & r_3(t) & 0 & 0 & \cdots & 0 \\ \vdots & \vdots & \vdots & \vdots & \ddots & \vdots \\ r_d(t) & r_d(t) & 0 & 0 & \cdots & 0 \end{pmatrix} \otimes \boldsymbol{I}_m. \tag{B.77}$$

The proof proceeds in two steps: we first bound the dynamics up to the predefined exit time $T_\varepsilon$, and then bound the subsequent evolution on the interval $[T_\varepsilon, \infty)$.

**Step 1: Bounding the dynamics up to $T_\varepsilon$.** Let $\widetilde{\boldsymbol{\Lambda}} := \mathrm{diag}(\|\boldsymbol{\Sigma}_{xy}\|, -\|\boldsymbol{\Sigma}_{xy}\|, 0, \ldots, 0) \otimes \boldsymbol{I}_m$, so that

$$\tau \frac{d}{dt} \boldsymbol{e}_\perp = (\widetilde{\boldsymbol{\Lambda}} + (\boldsymbol{R} - \widetilde{\boldsymbol{\Lambda}})) \boldsymbol{e}_\perp. \tag{B.78}$$

On $[0, T_\varepsilon]$, Corollary B.7 yields $\|\boldsymbol{a}(t)\|, \|\boldsymbol{w}_i(t)\| \le C_u \|\boldsymbol{u}(t)\|$ for every $i = 1, \ldots, d$. Therefore each product appearing in the coefficients $r_1 - \|\boldsymbol{\Sigma}_{xy}\|$, $r_2$, and $r_i$ is of the form $\boldsymbol{a}^\top \boldsymbol{w}_j$ and is bounded by

$$|\boldsymbol{a}^\top \boldsymbol{w}_j| \le \|\boldsymbol{a}\| \|\boldsymbol{w}_j\| \le C_u^2 \|\boldsymbol{u}\|^2, \qquad (j = 1, \ldots, d). \tag{B.79}$$

From the definitions of Equations (B.69) and (B.77), it follows that there exists a constant $C_{R1} > 0$ such that, for all $t \in [0, T_\varepsilon]$,

$$\|\boldsymbol{R}(t) - \widetilde{\boldsymbol{\Lambda}}\| \le C_{R1} \|\boldsymbol{u}(t)\|^2. \tag{B.80}$$

Define the running supremum $\overline{e}_\perp(t) := \sup_{t' \in [0,t]} \|\boldsymbol{e}_\perp(t')\|$. Applying the variation of constants formula gives

$$\boldsymbol{e}_\perp(t) = e^{\widetilde{\boldsymbol{\Lambda}} t / \tau} \boldsymbol{e}_\perp(0) + \frac{1}{\tau} \int_0^t e^{\widetilde{\boldsymbol{\Lambda}}(t-t')/\tau} (\boldsymbol{R}(t') - \widetilde{\boldsymbol{\Lambda}}) \boldsymbol{e}_\perp(t') \, dt'. \tag{B.81}$$

For the first term, since $\boldsymbol{P}_\perp \boldsymbol{u}(0) = 0$,

$$e^{\widetilde{\boldsymbol{\Lambda}} t / \tau} \boldsymbol{e}_\perp(0) = (0, e^{-\|\boldsymbol{\Sigma}_{xy}\| t / \tau} \boldsymbol{P}_\perp \boldsymbol{v}(0), \boldsymbol{P}_\perp \boldsymbol{w}_2(0), \ldots, \boldsymbol{P}_\perp \boldsymbol{w}_d(0))^\top \tag{B.82}$$

and hence

$$\|e^{\widetilde{\boldsymbol{\Lambda}} t / \tau} \boldsymbol{e}_\perp(0)\| \le C_{e1} s. \tag{B.83}$$

For the second term, we use $\|e^{\widetilde{\boldsymbol{\Lambda}}(t-t')/\tau}\| \le e^{\|\boldsymbol{\Sigma}_{xy}\|(t-t')/\tau}$ and Inequality (B.80):

$$\left\| \frac{1}{\tau} \int_0^t e^{\widetilde{\boldsymbol{\Lambda}}(t-t')/\tau} (\boldsymbol{R}(t') - \widetilde{\boldsymbol{\Lambda}}) \boldsymbol{e}_\perp(t') \, dt' \right\| \le \frac{1}{\tau} \int_0^t \|e^{\widetilde{\boldsymbol{\Lambda}}(t-t')/\tau}\| \|\boldsymbol{R}(t') - \widetilde{\boldsymbol{\Lambda}}\| \|\boldsymbol{e}_\perp(t')\| \, dt'$$

$$\le C_{R1} \overline{e}_\perp(t) \frac{1}{\tau} \int_0^t e^{\|\boldsymbol{\Sigma}_{xy}\|(t-t')/\tau} \|\boldsymbol{u}(t')\|^2 \, dt'. \tag{B.84}$$

Apply Lemma B.2 with $x(t) = \|\boldsymbol{u}(t)\|$, $\gamma = c_1$, $\beta = \|\boldsymbol{\Sigma}_{xy}\|$, and $p = 2$. By Inequality (B.40), $\tau \frac{d}{dt} x(t) \geq c_1 x(t)$, and since $c_1 = \|\boldsymbol{\Sigma}_{xy}\| - C_0 \varepsilon^2$, choosing $\varepsilon$ sufficiently small ensures $\beta/\gamma = \|\boldsymbol{\Sigma}_{xy}\|/c_1 \in (1, 2)$. Thus the second case of Lemma B.2 yields

$$\frac{1}{\tau} \int_0^t e^{\|\boldsymbol{\Sigma}_{xy}\|(t-t')/\tau} \|\boldsymbol{u}(t')\|^2 \, dt' \leq \frac{\|\boldsymbol{u}(t)\|^2}{2c_1 - \|\boldsymbol{\Sigma}_{xy}\|} \leq \frac{\varepsilon^2}{\|\boldsymbol{\Sigma}_{xy}\| - 2C_0 \varepsilon^2}. \tag{B.85}$$

Combining the above estimates, we arrive at

$$\overline{e}_\perp(t) \leq C_{e1} s + \frac{C_{R1} \varepsilon^2}{\|\boldsymbol{\Sigma}_{xy}\| - 2C_0 \varepsilon^2} \overline{e}_\perp(t). \tag{B.86}$$

Taking $\varepsilon$ smaller if necessary so that $\frac{C_{R1} \varepsilon^2}{\|\boldsymbol{\Sigma}_{xy}\| - 2C_0 \varepsilon^2} \leq \frac{1}{2}$, we obtain

$$\sup_{t \in [0, T_\varepsilon]} \|\boldsymbol{e}_\perp(t)\| = \overline{e}_\perp(T_\varepsilon) \leq 2C_{e1} s. \tag{B.87}$$

**Step 2: Bounding the dynamics over the infinite horizon $[T_\varepsilon, \infty)$.** Consider the loss

$$\mathcal{L}(t) = (\boldsymbol{\theta}(t) - \boldsymbol{\beta})^\top \boldsymbol{\Sigma} (\boldsymbol{\theta}(t) - \boldsymbol{\beta}) = \boldsymbol{\beta}^\top \boldsymbol{\Sigma} \boldsymbol{\beta} - 2\boldsymbol{\beta}^\top \boldsymbol{\Sigma} \boldsymbol{\theta}(t) + \boldsymbol{\theta}(t)^\top \boldsymbol{\Sigma} \boldsymbol{\theta}(t). \tag{B.88}$$

Under the gradient flow,

$$\tau \frac{d\mathcal{L}}{dt} = \nabla_{\boldsymbol{a}} \mathcal{L}^\top \frac{d\boldsymbol{a}}{dt} + \sum_{i=1}^d \nabla_{\boldsymbol{w}_i} \mathcal{L}^\top \frac{d\boldsymbol{w}_i}{dt} = -\|\nabla_{\boldsymbol{a}, \boldsymbol{W}} \mathcal{L}\|^2 \leq 0. \tag{B.89}$$

Since $\boldsymbol{\theta} = \boldsymbol{W}^\top \boldsymbol{a}$, the chain rule gives $\nabla_{\boldsymbol{W}} \mathcal{L} = \boldsymbol{a} (\nabla_{\boldsymbol{\theta}} \mathcal{L})^\top$ and hence $\|\nabla_{\boldsymbol{W}} \mathcal{L}\|_F^2 = \|\boldsymbol{a}\|^2 \|\nabla_{\boldsymbol{\theta}} \mathcal{L}\|^2$. Thus,

$$\|\nabla_{\boldsymbol{a}, \boldsymbol{W}} \mathcal{L}\|^2 = \|\nabla_{\boldsymbol{a}} \mathcal{L}\|^2 + \|\nabla_{\boldsymbol{W}} \mathcal{L}\|_F^2 = \|\nabla_{\boldsymbol{a}} \mathcal{L}\|^2 + \|\boldsymbol{a}\|^2 \|\nabla_{\boldsymbol{\theta}} \mathcal{L}\|^2 \geq \|\boldsymbol{a}\|^2 \|\nabla_{\boldsymbol{\theta}} \mathcal{L}\|^2, \tag{B.90}$$

$$\|\nabla_{\boldsymbol{\theta}} \mathcal{L}\|^2 = 4\|\boldsymbol{\Sigma}(\boldsymbol{\theta} - \boldsymbol{\beta})\|^2 \geq 4\sigma^2 (\boldsymbol{\theta} - \boldsymbol{\beta})^\top \boldsymbol{\Sigma} (\boldsymbol{\theta} - \boldsymbol{\beta}) = 4\sigma^2 \mathcal{L}(t). \tag{B.91}$$

By Lemma B.6, choosing $\varepsilon$ sufficiently small and then $s$ sufficiently small gives $\|\boldsymbol{w}_2(t)\| \leq \varepsilon/2$ for $t \leq T_\varepsilon$, while $\|\boldsymbol{w}_i(t)\| \leq \overline{c} s \leq \varepsilon/2$ for $i \geq 3$. Hence the exit at time $T_\varepsilon$ must occur through $\boldsymbol{a}$ or $\boldsymbol{w}_1$, so

$$\|\boldsymbol{u}(T_\varepsilon)\| + \|\boldsymbol{v}(T_\varepsilon)\| \geq \max\{\|\boldsymbol{a}(T_\varepsilon)\|, \|\boldsymbol{w}_1(T_\varepsilon)\|\} = \varepsilon. \tag{B.92}$$

Also, using the same variation of constants argument as in the proof of Lemma B.6, we obtain

$$\|\boldsymbol{v}(T_\varepsilon)\| \leq \|\boldsymbol{v}(0)\| + \frac{C_0 \varepsilon^2}{c_1} \|\boldsymbol{u}(T_\varepsilon)\|. \tag{B.93}$$

Since $\|\boldsymbol{v}(0)\| = O(s)$, taking $\varepsilon$ small enough that $C_0 \varepsilon^2/c_1 \leq 1/3$, and then $s$ small enough, this yields $\|\boldsymbol{v}(T_\varepsilon)\| \leq \|\boldsymbol{u}(T_\varepsilon)\|/3 + O(s)$, hence $\boldsymbol{a}(T_\varepsilon)^\top \boldsymbol{w}_1(T_\varepsilon) = \|\boldsymbol{u}(T_\varepsilon)\|^2 - \|\boldsymbol{v}(T_\varepsilon)\|^2 \geq \varepsilon^2/4$.

Since all weight norms are at most $\varepsilon$ at time $T_\varepsilon$, we have $\|\boldsymbol{\theta}(T_\varepsilon)\| = O(\varepsilon^2)$ and therefore $\boldsymbol{\theta}(T_\varepsilon)^\top \boldsymbol{\Sigma} \boldsymbol{\theta}(T_\varepsilon) = O(\varepsilon^4)$. On the other hand, $\boldsymbol{\beta}^\top \boldsymbol{\Sigma} \boldsymbol{\theta}(T_\varepsilon) = \boldsymbol{\Sigma}_{xy}^\top \boldsymbol{\theta}(T_\varepsilon) = \|\boldsymbol{\Sigma}_{xy}\| \boldsymbol{a}(T_\varepsilon)^\top \boldsymbol{w}_1(T_\varepsilon) \geq \|\boldsymbol{\Sigma}_{xy}\| \varepsilon^2/4$. Hence there exists $c_{\mathcal{L}} > 0$, independent of $s$, such that

$$\mathcal{L}(T_\varepsilon) \leq \boldsymbol{\beta}^\top \boldsymbol{\Sigma} \boldsymbol{\beta} - 2 \cdot \frac{\|\boldsymbol{\Sigma}_{xy}\|}{4} \varepsilon^2 + O(\varepsilon^4) \leq \boldsymbol{\beta}^\top \boldsymbol{\Sigma} \boldsymbol{\beta} - c_{\mathcal{L}} \varepsilon^2. \tag{B.94}$$

For any $t \geq T_\varepsilon$, Equation (B.89) gives $\mathcal{L}(t) \leq \mathcal{L}(T_\varepsilon)$, while $\mathcal{L}(t) \geq \boldsymbol{\beta}^\top \boldsymbol{\Sigma} \boldsymbol{\beta} - 2\boldsymbol{\beta}^\top \boldsymbol{\Sigma} \boldsymbol{\theta}(t) \geq \boldsymbol{\beta}^\top \boldsymbol{\Sigma} \boldsymbol{\beta} - 2\|\boldsymbol{\Sigma}_{xy}\| \|\boldsymbol{\theta}(t)\|$. Therefore $\|\boldsymbol{\theta}(t)\| \geq \frac{c_{\mathcal{L}}}{2\|\boldsymbol{\Sigma}_{xy}\|} \varepsilon^2 =: C_\theta$ for all $t \geq T_\varepsilon$.

Let $\boldsymbol{M} := \boldsymbol{a} \boldsymbol{a}^\top - \boldsymbol{W} \boldsymbol{W}^\top$, which is the conserved quantity and satisfies $\|\boldsymbol{M}\| = O(s^2)$. Since $\|\boldsymbol{\theta}(t)\|^2 = \boldsymbol{a}(t)^\top \boldsymbol{W}(t) \boldsymbol{W}(t)^\top \boldsymbol{a}(t) = \|\boldsymbol{a}(t)\|^4 - \boldsymbol{a}(t)^\top \boldsymbol{M} \boldsymbol{a}(t)$, we get $\|\boldsymbol{a}(t)\|^4 - C_\theta^2 \geq -\|\boldsymbol{M}\| \|\boldsymbol{a}(t)\|^2$, hence $\|\boldsymbol{a}(t)\|^2 \geq$

$C_\theta - \|\boldsymbol{M}\|/2$. For all sufficiently small $s$, this yields $\|\boldsymbol{a}(t)\| \geq \sqrt{C_\theta/2} =: c_a$ for $t \geq T_\varepsilon$. From Equations (B.89) to (B.91), with $\mu := 4\sigma^2 c_a^2$, we obtain $\tau \frac{d\mathcal{L}}{dt} \leq -\mu \mathcal{L}(t)$ and hence

$$\mathcal{L}(t) \leq \mathcal{L}(T_\varepsilon) e^{-\frac{\mu}{\tau}(t-T_\varepsilon)}. \tag{B.95}$$

Since $\mathcal{L}(t) \geq \sigma^2 \|\boldsymbol{\theta} - \boldsymbol{\beta}\|^2$, we obtain

$$\|\boldsymbol{\theta}(t) - \boldsymbol{\beta}\| \leq \sqrt{\frac{\mathcal{L}(T_\varepsilon)}{\sigma^2}} e^{-\frac{\mu}{2\tau}(t-T_\varepsilon)}. \tag{B.96}$$

Since $|r_i(t)| = |\boldsymbol{v}_i^\top \boldsymbol{\Sigma}(\boldsymbol{\theta}(t) - \boldsymbol{\beta})| \leq \sigma^2(1+\rho)\|\boldsymbol{\theta}(t) - \boldsymbol{\beta}\|$, there exists $C_{R2} > 0$ such that

$$\|\boldsymbol{R}(t)\| \leq C_{R2}\|\boldsymbol{\theta}(t) - \boldsymbol{\beta}\|. \tag{B.97}$$

Combining this with Inequality (B.96), we obtain

$$\|\boldsymbol{R}(t)\| \leq C_{R2} \sqrt{\frac{\mathcal{L}(T_\varepsilon)}{\sigma^2}} e^{-\frac{\mu}{2\tau}(t-T_\varepsilon)} =: C_{R3} e^{-\frac{\mu}{2\tau}(t-T_\varepsilon)}. \tag{B.98}$$

Now, from Equation (B.76), the deviation vector satisfies

$$\boldsymbol{e}_\perp(t) = \boldsymbol{e}_\perp(T_\varepsilon) + \frac{1}{\tau} \int_{T_\varepsilon}^t \boldsymbol{R}(t') \boldsymbol{e}_\perp(t') \, dt', \tag{B.99}$$

and therefore

$$\|\boldsymbol{e}_\perp(t)\| \leq \|\boldsymbol{e}_\perp(T_\varepsilon)\| + \frac{1}{\tau} \int_{T_\varepsilon}^t \|\boldsymbol{R}(t')\| \|\boldsymbol{e}_\perp(t')\| \, dt'. \tag{B.100}$$

Applying Gronwall's inequality yields

$$\|\boldsymbol{e}_\perp(t)\| \leq \|\boldsymbol{e}_\perp(T_\varepsilon)\| \exp\left(\frac{1}{\tau} \int_{T_\varepsilon}^t \|\boldsymbol{R}(t')\| \, dt'\right). \tag{B.101}$$

Thus, using Inequality (B.98), we have

$$\frac{1}{\tau} \int_{T_\varepsilon}^t \|\boldsymbol{R}(t')\| \, dt' \leq \frac{C_{R3}}{\tau} \int_{T_\varepsilon}^t e^{-\frac{\mu}{2\tau}(t'-T_\varepsilon)} \, dt' = \frac{2C_{R3}}{\mu}\left(1 - e^{-\frac{\mu}{2\tau}(t-T_\varepsilon)}\right) \leq \frac{2C_{R3}}{\mu}. \tag{B.102}$$

Also, since Step 1 gives $\|\boldsymbol{e}_\perp(T_\varepsilon)\| \leq 2C_{e1}s$, the preceding two bounds imply

$$\sup_{t\in[T_\varepsilon,\infty)} \|\boldsymbol{e}_\perp(t)\| \leq 2C_{e1} e^{2C_{R3}/\mu} s =: C_{e2}s. \tag{B.103}$$

**Step 3: Combining the bounds.** Combining Steps 1 and 2, and writing $C_{\boldsymbol{e}} := \max\{2C_{e1}, C_{e2}\}$, we obtain

$$\|\boldsymbol{e}_\perp(t)\| \leq C_{\boldsymbol{e}}s. \tag{B.104}$$

Since $\boldsymbol{a} = \boldsymbol{u} + \boldsymbol{v}$ and $\boldsymbol{w}_1 = \boldsymbol{u} - \boldsymbol{v}$, it follows that

$$\|\boldsymbol{P}_\perp \boldsymbol{a}(t)\| + \|\boldsymbol{P}_\perp \boldsymbol{w}_1(t)\| + \|\boldsymbol{P}_\perp \boldsymbol{w}_2(t)\| \leq 2\|\boldsymbol{P}_\perp \boldsymbol{u}(t)\| + 2\|\boldsymbol{P}_\perp \boldsymbol{v}(t)\| + \|\boldsymbol{P}_\perp \boldsymbol{w}_2(t)\| \leq 3\|\boldsymbol{e}_\perp(t)\|. \tag{B.105}$$

Therefore, with $C_\perp := 3C_{\boldsymbol{e}}$, we obtain

$$\|\boldsymbol{P}_\perp \boldsymbol{a}(t)\| + \|\boldsymbol{P}_\perp \boldsymbol{w}_1(t)\| + \|\boldsymbol{P}_\perp \boldsymbol{w}_2(t)\| \leq C_\perp s \tag{B.106}$$

for all $t \geq 0$. $\qquad\square$

### B.4 Derivation of the Two-Variable System (Proof of Theorem 3.3)

Building upon the global boundedness of the orthogonal components established in the previous subsection, we now provide the complete proof of Theorem 3.3. We demonstrate that the projected trajectory of the full system can be tightly approximated by the reduced two-variable dynamics, explicitly bounding the tracking error across all learning phases.

*Proof of Theorem 3.3.* Define

$$\tilde{a}(t) := \hat{\boldsymbol{r}}_1^\top \boldsymbol{a}(t), \quad \tilde{w}_1(t) := \hat{\boldsymbol{r}}_1^\top \boldsymbol{w}_1(t), \quad \tilde{w}_2(t) := \hat{\boldsymbol{r}}_1^\top \boldsymbol{w}_2(t). \tag{B.107}$$

By Lemma 3.2, the components orthogonal to $\hat{\boldsymbol{r}}_1$, as well as the modes $\boldsymbol{w}_i$ with $i \geq 3$, remain uniformly $O(s)$ for all $t \geq 0$. Thus, up to an $O(s)$ error, the full trajectory is captured by its projection onto $(\tilde{a}, \tilde{w}_1, \tilde{w}_2)$.

**Effective Two-Variable Description of the Projected Exact Trajectory.** By leveraging the bound $\left\| \boldsymbol{a}\boldsymbol{a}^\top - \sum_{i=1}^d \boldsymbol{w}_i\boldsymbol{w}_i^\top \right\| = O(s^2)$ together with $|\tilde{w}_i(t)| \leq \|\boldsymbol{w}_i(t)\| \leq \bar{c}s$ for all $i \geq 3$ and projecting the conservation law onto $\hat{\boldsymbol{r}}_1$, there exists a constant $C_q > 0$, independent of $s$, such that

$$\left| \tilde{a}(t)^2 - (\tilde{w}_1(t)^2 + \tilde{w}_2(t)^2) \right| \leq \sum_{i=3}^d |\tilde{w}_i(t)|^2 + \left\| \boldsymbol{a}\boldsymbol{a}^\top - \sum_{i=1}^d \boldsymbol{w}_i\boldsymbol{w}_i^\top \right\| \leq C_q^2 s^2. \tag{B.108}$$

Since $\||x| - |y|\|^2 \leq |x^2 - y^2|$, Inequality (B.108) gives

$$\left| |\tilde{a}(t)| - \sqrt{\tilde{w}_1(t)^2 + \tilde{w}_2(t)^2} \right| \leq C_q s. \tag{B.109}$$

We now show that $|\tilde{a}(t)|$ can be replaced by $\tilde{a}(t)$ up to an $O(s)$ error. To this end, we first treat the initial transient. Choose a fixed constant $M > 0$ sufficiently large, and let $t_0 = O(1)$ be the time given by Corollary B.5. On $[0, t_0]$, all weights remain $O(s)$, and therefore $\left| \tilde{a}(t) - \sqrt{\tilde{w}_1(t)^2 + \tilde{w}_2(t)^2} \right| \leq C_{\text{in}}s$.

Next consider $t \in [t_0, T_\varepsilon]$, where $T_\varepsilon$ is the exit time defined in Equation (B.38). From Lemma 3.2, there exists a constant $C_{\perp u} > 0$ such that $\|\boldsymbol{P}_\perp \boldsymbol{u}(t)\| \leq C_{\perp u}s$. Taking $M > C_{\perp u}$ in Corollary B.5, and using the fact that the parallel component $\hat{\boldsymbol{r}}_1^\top \boldsymbol{u}(t)$ is positive at $t = t_0$ and cannot vanish on $[t_0, T_\varepsilon]$, we obtain $\hat{\boldsymbol{r}}_1^\top \boldsymbol{u}(t) \geq \|\boldsymbol{u}(t)\| - C_{\perp u}s$. Hence

$$\tilde{a}(t) = \hat{\boldsymbol{r}}_1^\top \boldsymbol{u}(t) + \hat{\boldsymbol{r}}_1^\top \boldsymbol{v}(t) \geq \|\boldsymbol{u}(t)\| - \|\boldsymbol{v}(t)\| - C_{\perp u}s \geq (M - C_{\perp u})s > 0. \tag{B.110}$$

on $[t_0, T_\varepsilon]$.

Finally, for $t \geq T_\varepsilon$, Step 2 in the proof of Lemma 3.2 gives a lower bound $\|\boldsymbol{a}(t)\| \geq c_a > 0$, independent of $s$. Combining this with the global orthogonal-component bound $\|\boldsymbol{P}_\perp \boldsymbol{a}(t)\| \leq C_\perp s$, we have $|\tilde{a}(t)| \geq c_a/2$ for all sufficiently small $s$. Since $\tilde{a}(T_\varepsilon) > 0$ and $\tilde{a}(t)$ is continuous, $\tilde{a}(t)$ cannot cross zero after $T_\varepsilon$. Therefore $\tilde{a}(t) > 0$ for all $t \geq T_\varepsilon$.

Combining the three intervals, there exists a constant $C_{\text{tr}} > 0$, independent of $s$, such that

$$\left| \tilde{a}(t) - \sqrt{\tilde{w}_1(t)^2 + \tilde{w}_2(t)^2} \right| \leq C_{\text{tr}}s \tag{B.111}$$

for all $t \geq 0$. Thus, up to an $O(s)$ error, the projected exact state is effectively described by the two variables

$$\tilde{\boldsymbol{w}}(t) := (\tilde{w}_1(t), \tilde{w}_2(t))^\top. \tag{B.112}$$

Therefore, in the remainder of the proof we take $\tilde{\boldsymbol{w}}$ as the exact trajectory to be compared with the ideal reduced system, while $\tilde{a}$ is treated as the dependent quantity recovered from $\tilde{\boldsymbol{w}}$ through Inequality (B.111).

**Ideal Reduced Trajectory.** We define the subspace parametrized by three scalar functions $(a^\star(t), w_1^\star(t), w_2^\star(t))$ as follows:

$$\mathcal{M} = \{(\boldsymbol{a}, \boldsymbol{w}_1, \ldots, \boldsymbol{w}_d) : \boldsymbol{a} = a^\star \hat{\boldsymbol{r}}_1, \boldsymbol{w}_1 = w_1^\star \hat{\boldsymbol{r}}_1, \boldsymbol{w}_2 = w_2^\star \hat{\boldsymbol{r}}_1, \boldsymbol{w}_i = 0 \ \ (i \geq 3)\}. \tag{B.113}$$

From Equations (9) to (12), if $(\boldsymbol{a}(0), \boldsymbol{w}_1(0), \ldots, \boldsymbol{w}_d(0)) \in \mathcal{M}$, then $(\boldsymbol{a}(t), \boldsymbol{w}_1(t), \ldots, \boldsymbol{w}_d(t)) \in \mathcal{M}$ for all $t \geq 0$. In addition, the conservation law implies $\frac{d}{dt}((a^\star)^2 - (w_1^\star)^2 - (w_2^\star)^2) = 0$.

Within this invariant set $\mathcal{M}$, we construct the ideal trajectory by defining the specific scalar components $(a(t), w_1(t), w_2(t))$. We set their initial conditions to precisely match the growing mode of the projected exact initialization:

$$a(0) = u_0 + \frac{q_0^2}{4u_0}, \quad w_1(0) = u_0 - \frac{q_0^2}{4u_0}, \quad w_2(0) = q_0 \tag{B.114}$$

where

$$u_0 := \hat{\boldsymbol{r}}_1^\top \frac{\boldsymbol{a}(0) + \boldsymbol{w}_1(0)}{2}, \quad q_0 := \hat{\boldsymbol{r}}_1^\top \boldsymbol{w}_2(0). \tag{B.115}$$

This initialization ensures that $a(0)^2 - w_1(0)^2 - w_2(0)^2 = 0$, which implies that $a(t)^2 = w_1(t)^2 + w_2(t)^2$ holds for all $t \geq 0$.

Since $a(0) > 0$, the sign of $a(t)$ could change only if $a(t_0) = 0$ for some $t_0$. In that case, necessarily $w_1(t_0) = w_2(t_0) = 0$, so the trajectory reaches the origin, which is an equilibrium of the system. Uniqueness of solutions for this rank-one system implies that the trajectory would have to remain at the origin for all later times, contradicting the nontrivial initialization. Hence $a(t) = \sqrt{w_1(t)^2 + w_2(t)^2}$ for all $t \geq 0$.

Consequently, the dynamics on $\mathcal{M}$ with the initial conditions given by Equation (B.114) are fully described by the two-dimensional vector

$$\boldsymbol{w}(t) := (w_1(t), w_2(t))^\top. \tag{B.116}$$

This exactly follows the dynamics of the reduced system (Equations (16) and (17)) with the initial condition in Equation (18).

**Time Intervals and Proof Strategy.** We compare the projected exact trajectory $\tilde{\boldsymbol{w}}$ and the ideal reduced trajectory $\boldsymbol{w}$ over three time intervals. Fix the small threshold $\varepsilon$ from Equation (B.38) and set $\varepsilon_0 := \varepsilon/3$. Define

$$T_{\varepsilon_0} := \inf \{t \geq 0 : \max\{\|\tilde{\boldsymbol{w}}(t)\|, \|\boldsymbol{w}(t)\|\} \geq \varepsilon_0\}. \tag{B.117}$$

By construction, both trajectories remain within a small neighborhood of the origin on the interval $[0, T_{\varepsilon_0}]$.

As shown later in Lemma C.4, the reduced two-variable system (Equations (16) and (17)) has a unique nonzero equilibrium, and this equilibrium is locally exponentially stable. In addition, under the initialization in Equation (18), Lemmas C.5 and C.6 imply that the ideal reduced trajectory $\boldsymbol{w}(t)$ converges to this equilibrium. We denote this limit by $\boldsymbol{w}_\infty$. For any fixed $\varepsilon_\infty > 0$, we define the time at which the ideal reduced trajectory enters an $\varepsilon_\infty$-neighborhood of this equilibrium as:

$$T_{\varepsilon_\infty} := \inf\{t \geq 0 : \|\boldsymbol{w}(t) - \boldsymbol{w}_\infty\| \leq \varepsilon_\infty\}. \tag{B.118}$$

For the fixed sufficiently small initialization scale $s > 0$, Corollary C.7 gives $T_{\varepsilon_\infty} < \infty$.

The proof is then divided into three steps. In Step 1, we analyze the near-origin dynamics on $[0, T_{\varepsilon_0}]$ by transforming the trajectories into lifted coordinates to decouple the growing and decaying modes. In Step 2, we return to the two-variable coordinates to characterize the tracking error over the intermediate interval $[T_{\varepsilon_0}, T_{\varepsilon_\infty}]$. In Step 3, we use the asymptotic stability of $\boldsymbol{w}_\infty$ to control the error on the final interval $[T_{\varepsilon_\infty}, \infty)$.

**Step 1: Bounding the dynamics on the interval** $[0, T_{\varepsilon_0}]$**.** On the near-origin interval $[0, T_{\varepsilon_0}]$, we compare the two trajectories $\tilde{\boldsymbol{w}}$ and $\boldsymbol{w}$ after rewriting them in lifted coordinates that separate the growing and decaying modes.

For the projected exact trajectory $\tilde{\boldsymbol{w}}$, we define

$$\tilde{u}(t) := \frac{\tilde{a}(t) + \tilde{w}_1(t)}{2}, \quad \tilde{v}(t) := \frac{\tilde{a}(t) - \tilde{w}_1(t)}{2}, \quad \tilde{\boldsymbol{z}}(t) := (\tilde{u}(t), \tilde{v}(t), \tilde{w}_2(t))^\top. \tag{B.119}$$

Accordingly, projecting Equations (9), (10) and (12) onto $\hat{\boldsymbol{r}}_1$ and rewriting the result in the variables $\tilde{\boldsymbol{z}}$ yields

$$\tau \frac{d}{dt}\tilde{\boldsymbol{z}} = \boldsymbol{\Lambda}\tilde{\boldsymbol{z}} + \boldsymbol{h}(\tilde{\boldsymbol{z}}) + \boldsymbol{\xi}, \tag{B.120}$$

where $\boldsymbol{\Lambda} = \mathrm{diag}(\|\boldsymbol{\Sigma}_{xy}\|, -\|\boldsymbol{\Sigma}_{xy}\|, 0)$, and $\boldsymbol{h} : \mathbb{R}^3 \to \mathbb{R}^3$ denotes the explicit cubic polynomial vector field appearing in the lifted dynamics of the variables $(\tilde{u}, \tilde{v}, \tilde{w}_2)$. The vector $\boldsymbol{\xi} = (\xi_u, \xi_v, \xi_2)^\top$ collects all terms involving the orthogonal components and the remaining projected weights $\tilde{w}_i$ for $i \geq 3$. The linear terms project exactly onto the reduced variables, so $\boldsymbol{\xi}$ comes only from the cubic terms. After decomposing each vector into its $\hat{\boldsymbol{r}}_1$ component and its orthogonal component, and then projecting onto $\hat{\boldsymbol{r}}_1$, all contributions with exactly one orthogonal factor vanish by orthogonality. Thus every residual term contains at least two orthogonal or remaining factors . The remaining factor is either one of $(\tilde{u}, \tilde{v}, \tilde{w}_2)$ or another orthogonal or remainder component. For instance, projecting the $\boldsymbol{w}_2$ equation onto $\hat{\boldsymbol{r}}_1$ gives $\tau \frac{d}{dt}\tilde{w}_2 = (\nu \boldsymbol{a}^\top \boldsymbol{w}_1 - \lambda_2 \boldsymbol{a}^\top \boldsymbol{w}_2)\tilde{a}$. Since $\boldsymbol{a}^\top \boldsymbol{w}_j = \tilde{a}\tilde{w}_j + (\boldsymbol{P}_\perp \boldsymbol{a})^\top (\boldsymbol{P}_\perp \boldsymbol{w}_j)$, the unperturbed term in $\boldsymbol{h}(\tilde{\boldsymbol{z}})$ is $(\nu \tilde{a}\tilde{w}_1 - \lambda_2 \tilde{a}\tilde{w}_2)\tilde{a}$, while the corresponding residual is $(\nu(\boldsymbol{P}_\perp \boldsymbol{a})^\top (\boldsymbol{P}_\perp \boldsymbol{w}_1) - \lambda_2 (\boldsymbol{P}_\perp \boldsymbol{a})^\top (\boldsymbol{P}_\perp \boldsymbol{w}_2))\tilde{a}$. By Lemma 3.2, this term is bounded by $Cs^2|\tilde{a}| \leq Cs^2(|\tilde{u}| + |\tilde{v}|)$. The other residual terms have the same structure. For the terms involving the remaining components $i \geq 3$, we additionally use $\|\boldsymbol{w}_i(t)\| \leq \bar{c}s$. Hence all orthogonal and remaining components are uniformly $O(s)$, and the residual terms are bounded by $O(s^2|\tilde{u}|)$, $O(s^2|\tilde{v}|)$, $O(s^2|\tilde{w}_2|)$, and $O(s^3)$. Therefore, for all $t \geq 0$,

$$\|\boldsymbol{\xi}(t)\| \leq C_\xi s^2(|\tilde{u}| + |\tilde{v}| + |\tilde{w}_2| + s). \tag{B.121}$$

For $t \in [0, T_{\varepsilon_0}]$, by definition we have $\|\tilde{\boldsymbol{w}}(t)\| \leq \varepsilon_0$, hence $|\tilde{w}_1(t)|, |\tilde{w}_2(t)| \leq \varepsilon_0$. In addition, Inequality (B.111) gives $|\tilde{a}(t)| \leq \sqrt{\tilde{w}_1(t)^2 + \tilde{w}_2(t)^2} + C_{\mathrm{tr}}s \leq 2\varepsilon_0$ for all sufficiently small $s$. Using Lemma 3.2, we obtain

$$\|\boldsymbol{a}(t)\|, \|\boldsymbol{w}_1(t)\|, \|\boldsymbol{w}_2(t)\| \leq 2\varepsilon_0 + C_\perp s \leq \varepsilon \tag{B.122}$$

for all sufficiently small $s$, because $\varepsilon_0 = \varepsilon/3$. Therefore $[0, T_{\varepsilon_0}] \subset [0, T_\varepsilon]$, so Corollary B.7 applies throughout $[0, T_{\varepsilon_0}]$. Therefore,

$$|\tilde{u}(t)| \leq \|\boldsymbol{u}(t)\|, \quad |\tilde{v}(t)| \leq \|\boldsymbol{v}(t)\| \leq C_u \|\boldsymbol{u}(t)\|, \quad |\tilde{w}_2(t)| \leq \|\boldsymbol{w}_2(t)\| \leq C_u \|\boldsymbol{u}(t)\|. \tag{B.123}$$

Thus, using Inequality (B.123), we have

$$\|\tilde{\boldsymbol{z}}(t)\| \leq C_{\tilde{z}} \|\boldsymbol{u}(t)\|. \tag{B.124}$$

Also, substituting these bounds into Inequality (B.121), for $t \in [0, T_{\varepsilon_0}]$, we obtain

$$\|\boldsymbol{\xi}(t)\| \leq C_{\xi 1} s^2 \|\boldsymbol{u}(t)\|. \tag{B.125}$$

In addition, $\|\boldsymbol{u}(t)\| \leq (\|\boldsymbol{a}(t)\| + \|\boldsymbol{w}_1(t)\|)/2 \leq \varepsilon = 3\varepsilon_0$.

For the ideal reduced trajectory $\boldsymbol{w}$, we set

$$u(t) := \frac{\sqrt{w_1(t)^2 + w_2(t)^2} + w_1(t)}{2}, \quad v(t) := \frac{\sqrt{w_1(t)^2 + w_2(t)^2} - w_1(t)}{2}, \tag{B.126}$$

$$\boldsymbol{z}(t) := (u(t), v(t), w_2(t))^\top. \tag{B.127}$$

Since the ideal reduced trajectory $\boldsymbol{z}$ is a special case of the general dynamics where all orthogonal components and the remaining weights $\boldsymbol{w}_i$ for $i \geq 3$ are identically zero, its lifted dynamics are governed by the exact same differential equations as Equation (B.120) but without the residual error term $\boldsymbol{\xi}$. Therefore,

$$\tau \frac{d}{dt}\boldsymbol{z} = \boldsymbol{\Lambda}\boldsymbol{z} + \boldsymbol{h}(\boldsymbol{z}). \tag{B.128}$$

The arguments of Lemmas B.4 and B.6 and the proof of Corollary B.7 also apply directly to $(u, v, w_2)$ on $[0, T_{\varepsilon_0}]$, with the same constant $c_1 = \|\boldsymbol{\Sigma}_{xy}\| - C_0\varepsilon^2$. In particular, the following inequalities hold:

$$0 \leq v(t) \leq u(t), \quad \tau \frac{d}{dt}u(t) \geq c_1 u(t), \tag{B.129}$$

$$\|\boldsymbol{z}(t)\| \leq C_z u(t), \tag{B.130}$$

and $u(t) \leq \varepsilon_0$.

Let $\boldsymbol{e}_z := \tilde{\boldsymbol{z}} - \boldsymbol{z}$ and $\overline{e}_z(t) := \sup_{t' \in [0,t]} \|\boldsymbol{e}_z(t')\|$. Subtracting Equation (B.128) from Equation (B.120), we obtain

$$\tau \frac{d}{dt} \boldsymbol{e}_z = \boldsymbol{\Lambda} \boldsymbol{e}_z + (\boldsymbol{h}(\tilde{\boldsymbol{z}}) - \boldsymbol{h}(\boldsymbol{z})) + \boldsymbol{\xi}. \tag{B.131}$$

The variation of constants formula gives

$$\boldsymbol{e}_z(t) = e^{\boldsymbol{\Lambda} t / \tau} \boldsymbol{e}_z(0) + \frac{1}{\tau} \int_0^t e^{\boldsymbol{\Lambda}(t-t')/\tau} (\boldsymbol{h}(\tilde{\boldsymbol{z}}(t')) - \boldsymbol{h}(\boldsymbol{z}(t')) + \boldsymbol{\xi}(t')) \, dt'. \tag{B.132}$$

For the first term, we have

$$e^{\boldsymbol{\Lambda} t / \tau} \boldsymbol{e}_z(0) = (0, (\tilde{v}(0) - v(0)) e^{-\|\boldsymbol{\Sigma}_{xy}\| t / \tau}, 0)^\top, \tag{B.133}$$

and hence

$$\|e^{\boldsymbol{\Lambda} t / \tau} \boldsymbol{e}_z(0)\| \leq C_{z1} s. \tag{B.134}$$

Since $\boldsymbol{h}$ is cubic, the integral form of the mean-value theorem together with Inequalities (B.124) and (B.130) yields

$$\|\boldsymbol{h}(\tilde{\boldsymbol{z}}) - \boldsymbol{h}(\boldsymbol{z})\| \leq C_h(\|\boldsymbol{u}\|^2 + u^2)\|\boldsymbol{e}_z\|. \tag{B.135}$$

Taking norms the second term in Equation (B.132) and using Inequalities (B.125) and (B.135), we obtain

$$\left\| \frac{1}{\tau} \int_0^t e^{\boldsymbol{\Lambda}(t-t')/\tau} (\boldsymbol{h}(\tilde{\boldsymbol{z}}(t')) - \boldsymbol{h}(\boldsymbol{z}(t')) + \boldsymbol{\xi}(t')) \, dt' \right\|$$
$$\leq C_h \overline{e}_z(t) \frac{1}{\tau} \int_0^t e^{\|\boldsymbol{\Sigma}_{xy}\|(t-t')/\tau} (\|\boldsymbol{u}(t')\|^2 + (u(t'))^2) \, dt' + C_{\xi 1} s^2 \frac{1}{\tau} \int_0^t e^{\|\boldsymbol{\Sigma}_{xy}\|(t-t')/\tau} \|\boldsymbol{u}(t')\| \, dt'. \tag{B.136}$$

Let $\alpha := \|\boldsymbol{\Sigma}_{xy}\|/c_1 = \|\boldsymbol{\Sigma}_{xy}\|/(\|\boldsymbol{\Sigma}_{xy}\| - C_0\varepsilon^2) \in (1,2)$, where the inclusion follows by choosing $\varepsilon$ sufficiently small. Since $T_{\varepsilon_0} \leq T_\varepsilon$, Inequality (B.40) gives $\tau \frac{d}{dt} \|\boldsymbol{u}\| \geq c_1 \|\boldsymbol{u}\|$, and Inequality (B.129) gives the same inequality for $u$. Applying Lemma B.2 with $(x, \gamma, \beta, p) = (\|\boldsymbol{u}\|, c_1, \|\boldsymbol{\Sigma}_{xy}\|, 2)$, $(u, c_1, \|\boldsymbol{\Sigma}_{xy}\|, 2)$, and $(\|\boldsymbol{u}\|, c_1, \|\boldsymbol{\Sigma}_{xy}\|, 1)$, respectively, we obtain

$$\frac{1}{\tau} \int_0^t e^{\|\boldsymbol{\Sigma}_{xy}\|(t-t')/\tau} \|\boldsymbol{u}(t')\|^2 \, dt' \leq \frac{\|\boldsymbol{u}(t)\|^2}{2c_1 - \|\boldsymbol{\Sigma}_{xy}\|} \leq C_{u2} \varepsilon_0^2, \tag{B.137}$$

$$\frac{1}{\tau} \int_0^t e^{\|\boldsymbol{\Sigma}_{xy}\|(t-t')/\tau} u(t')^2 \, dt' \leq \frac{u(t)^2}{2c_1 - \|\boldsymbol{\Sigma}_{xy}\|} \leq C_{u3} \varepsilon_0^2, \tag{B.138}$$

$$s^2 \frac{1}{\tau} \int_0^t e^{\|\boldsymbol{\Sigma}_{xy}\|(t-t')/\tau} \|\boldsymbol{u}(t')\| \, dt' \leq C_{\xi 1}'' s^2 \|\boldsymbol{u}(t)\|^\alpha \|\boldsymbol{u}(0)\|^{1-\alpha} \leq C_{\xi 1}' \varepsilon_0^\alpha s^{3-\alpha}. \tag{B.139}$$

Here the first two lines use the second case of Lemma B.2, together with $\|\boldsymbol{u}(t)\|^2 \leq 9\varepsilon_0^2$, while the third line uses the first case with $\|\boldsymbol{u}(t)\| \leq 3\varepsilon_0$ and $\|\boldsymbol{u}(0)\| = c_{u0} s$.

Using Equation (B.132), Inequality (B.134), and Inequalities (B.137) to (B.139) in Inequality (B.136), we obtain

$$\overline{e}_z(t) \leq C_{z1} s + C_{h1} \varepsilon_0^2 \overline{e}_z(t) + C_{\xi 1}' \varepsilon_0^\alpha s^{3-\alpha}. \tag{B.140}$$

Choosing $\varepsilon_0$ sufficiently small so that $C_{h1} \varepsilon_0^2 \leq 1/2$, and using $s^{3-\alpha} = O(s)$ since $3 - \alpha > 1$, we obtain, for all $t \in [0, T_{\varepsilon_0}]$,

$$\overline{e}_z(t) \leq C_z s. \tag{B.141}$$

Since $\tilde{w}_1 - w_1 = (\tilde{u} - u) - (\tilde{v} - v)$ and $\tilde{w}_2 - w_2 = \tilde{w}_2 - w_2$, Inequality (B.141) immediately implies, for all $t \in [0, T_{\varepsilon_0}]$,

$$\|\tilde{\boldsymbol{w}}(t) - \boldsymbol{w}(t)\| \leq C_{w1} s. \tag{B.142}$$

**Step 2: Bounding the dynamics over the intermediate interval $[T_{\varepsilon_0}, T_{\varepsilon_\infty}]$.** We now return to the two-variable coordinates. Define $\boldsymbol{F} : \mathbb{R}^2 \to \mathbb{R}^2$ by

$$\boldsymbol{F}(x_1, x_2) := \begin{pmatrix} \|\boldsymbol{\Sigma}_{xy}\| \sqrt{x_1^2 + x_2^2} - \lambda_1 x_1 (x_1^2 + x_2^2) + \nu x_2 (x_1^2 + x_2^2) \\ \nu x_1 (x_1^2 + x_2^2) - \lambda_2 x_2 (x_1^2 + x_2^2) \end{pmatrix}. \tag{B.143}$$

Then the reduced two-variable dynamics and the projected full-system dynamics take the form

$$\tau \frac{d}{dt} \boldsymbol{w} = \boldsymbol{F}(\boldsymbol{w}), \tag{B.144}$$

$$\tau \frac{d}{dt} \tilde{\boldsymbol{w}} = \boldsymbol{F}(\tilde{\boldsymbol{w}}) + \boldsymbol{\zeta}(t), \tag{B.145}$$

where the total error term is $\boldsymbol{\zeta}(t) := \boldsymbol{E}_a(t) + \hat{\boldsymbol{\xi}}(t)$ with

$$\boldsymbol{E}_a(t) := \begin{pmatrix} \|\boldsymbol{\Sigma}_{xy}\|(\tilde{a} - \sqrt{\tilde{w}_1^2 + \tilde{w}_2^2}) - \lambda_1 \tilde{w}_1 (\tilde{a}^2 - (\tilde{w}_1^2 + \tilde{w}_2^2)) + \nu \tilde{w}_2 (\tilde{a}^2 - (\tilde{w}_1^2 + \tilde{w}_2^2)) \\ \nu \tilde{w}_1 (\tilde{a}^2 - (\tilde{w}_1^2 + \tilde{w}_2^2)) - \lambda_2 \tilde{w}_2 (\tilde{a}^2 - (\tilde{w}_1^2 + \tilde{w}_2^2)) \end{pmatrix} \tag{B.146}$$

and $\hat{\boldsymbol{\xi}}(t) := (\xi_u(t) - \xi_v(t), \xi_2(t))^\top$.

Let $\boldsymbol{e}(t) := \tilde{\boldsymbol{w}}(t) - \boldsymbol{w}(t)$. Subtracting Equation (B.144) from Equation (B.145), we obtain

$$\tau \frac{d}{dt} \boldsymbol{e} = \boldsymbol{F}(\tilde{\boldsymbol{w}}) - \boldsymbol{F}(\boldsymbol{w}) + \boldsymbol{\zeta}(t). \tag{B.147}$$

By the definition of $T_{\varepsilon_0}$ and continuity, $\max\{\|\tilde{\boldsymbol{w}}(T_{\varepsilon_0})\|, \|\boldsymbol{w}(T_{\varepsilon_0})\|\} = \varepsilon_0$. Combining this with Inequality (B.142) gives $\varepsilon_0 \leq \|\boldsymbol{w}(T_{\varepsilon_0})\| + \|\tilde{\boldsymbol{w}}(T_{\varepsilon_0}) - \boldsymbol{w}(T_{\varepsilon_0})\| \leq \|\boldsymbol{w}(T_{\varepsilon_0})\| + C_{w1}s$, and hence $\|\boldsymbol{w}(T_{\varepsilon_0})\| \geq \varepsilon_0/2$ for all sufficiently small $s$. Applying Corollary C.8, established later in Appendix C.2.1, with $R = \varepsilon_0/2$ and $T_\star = T_{\varepsilon_0}$ shows that the interval length $T_{\varepsilon_\infty} - T_{\varepsilon_0}$ is bounded independently of $s$. On this interval, $\boldsymbol{w}(t)$ is bounded by Lemma C.6, while $\tilde{\boldsymbol{w}}(t)$ is uniformly bounded because the loss monotonicity in Equation (B.89) bounds $\|\boldsymbol{\theta}(t)\|$ and the conserved relation $\boldsymbol{a}\boldsymbol{a}^\top - \boldsymbol{W}\boldsymbol{W}^\top = O(s^2)$ then bounds $\|\boldsymbol{a}(t)\|$ and $\|\boldsymbol{W}(t)\|_F$. Hence both trajectories lie in a fixed bounded set. Since the only non-polynomial term in $\boldsymbol{F}$ is $(x_1, x_2) \mapsto \sqrt{x_1^2 + x_2^2}$, which is globally 1-Lipschitz, and the remaining terms of $\boldsymbol{F}$ are polynomial, there exists $L_{\varepsilon_0, \varepsilon_\infty} > 0$ such that $\|\boldsymbol{F}(\tilde{\boldsymbol{w}}) - \boldsymbol{F}(\boldsymbol{w})\| \leq L_{\varepsilon_0, \varepsilon_\infty} \|\tilde{\boldsymbol{w}} - \boldsymbol{w}\|$ on $[T_{\varepsilon_0}, T_{\varepsilon_\infty}]$. Since $\tilde{\boldsymbol{w}}$ remains bounded, Inequalities (B.108) and (B.111) gives $\|\boldsymbol{E}_a(t)\| \leq C'_{E_a}(|\tilde{a}(t) - \sqrt{\tilde{w}_1(t)^2 + \tilde{w}_2(t)^2}| + |\tilde{a}(t)^2 - (\tilde{w}_1(t)^2 + \tilde{w}_2(t)^2)|) \leq C_{E_a}s$, while Inequality (B.121) and the fact that the lifted variables $(\tilde{u}(t), \tilde{v}(t), \tilde{w}_2(t))$ remain uniformly bounded on $[T_{\varepsilon_0}, T_{\varepsilon_\infty}]$ imply $\|\hat{\boldsymbol{\xi}}(t)\| \leq C_{\xi 2} s^2$. Hence

$$\|\boldsymbol{\zeta}(t)\| \leq \|\boldsymbol{E}_a(t)\| + \|\hat{\boldsymbol{\xi}}(t)\| \leq C_{\zeta 2} s \tag{B.148}$$

for some constant $C_{\zeta 2} > 0$.

Applying Gronwall's inequality on $[T_{\varepsilon_0}, t]$ yields

$$\|\boldsymbol{e}(t)\| \leq \|\boldsymbol{e}(T_{\varepsilon_0})\| e^{L_{\varepsilon_0, \varepsilon_\infty}(t - T_{\varepsilon_0})/\tau} + \frac{C_{\zeta 2} s}{L_{\varepsilon_0, \varepsilon_\infty}} \left( e^{L_{\varepsilon_0, \varepsilon_\infty}(t - T_{\varepsilon_0})/\tau} - 1 \right). \tag{B.149}$$

From the conclusion of Step 1, we have the initial error bound $\|\boldsymbol{e}(T_{\varepsilon_0})\| \leq C_{\varepsilon_0} s$ for some $C_{\varepsilon_0} > 0$. Since the transit time is bounded, the exponential factor is uniformly $O(1)$ on $[T_{\varepsilon_0}, T_{\varepsilon_\infty}]$. Therefore, the error remains bounded by a constant multiple of $s$ throughout the entire intermediate interval:

$$\|\boldsymbol{e}(t)\| \leq C_{w2} s \tag{B.150}$$

for some constant $C_{w2} > 0$.

**Step 3: Bounding the dynamics over the infinite horizon $[T_{\varepsilon_\infty}, \infty)$.** By step 2, Equation (B.145) may be viewed as a perturbation of Equation (B.144), with forcing term satisfying $\|\boldsymbol{\zeta}(t)\| \leq C_{\zeta 3} s$ uniformly for all $t \geq T_{\varepsilon_\infty}$.

Since $\boldsymbol{w}_\infty$ is an exponentially stable equilibrium of the reduced two-variable system, standard perturbation results for exponentially stable systems (Khalil (2002), Theorem 4.14 and Lemma 9.2) imply that, for all $t \geq T_{\varepsilon_\infty}$, the tracking error satisfies

$$\|\tilde{\boldsymbol{w}}(t) - \boldsymbol{w}(t)\| \leq C_{w3} s. \tag{B.151}$$

**Step 4: Combining the bounds.** Combining the bounds over the three intervals from Steps 1 to 3, there exists a constant $C_\| > 0$ such that, for all $t \geq 0$,

$$\|\tilde{\boldsymbol{w}}(t) - \boldsymbol{w}(t)\| \leq C_\| s. \tag{B.152}$$

The global relation Inequality (B.111) then gives

$$\left| \tilde{a}(t) - \sqrt{w_1(t)^2 + w_2(t)^2} \right| \leq \left| \tilde{a}(t) - \sqrt{\tilde{w}_1(t)^2 + \tilde{w}_2(t)^2} \right| + \|\tilde{\boldsymbol{w}}(t) - \boldsymbol{w}(t)\| \leq (C_{\text{tr}} + C_\|)s \tag{B.153}$$

for all $t \geq 0$. Recall that the actual trajectories in the full network can be decomposed into the projected components and the orthogonal components, i.e., $\boldsymbol{w}_i(t) = \tilde{w}_i(t)\hat{\boldsymbol{r}}_1 + \boldsymbol{P}_\perp \boldsymbol{w}_i(t)$ and $\boldsymbol{a}(t) = \tilde{a}(t)\hat{\boldsymbol{r}}_1 + \boldsymbol{P}_\perp \boldsymbol{a}(t)$. The orthogonal components are globally bounded by $\|\boldsymbol{P}_\perp \boldsymbol{a}(t)\|, \|\boldsymbol{P}_\perp \boldsymbol{w}_i(t)\| \leq C_\perp s$ from Lemma 3.2. Applying the triangle inequality to these parallel and orthogonal bounds directly yields the final statement of the theorem:

$$\|\boldsymbol{w}_1(t) - \hat{\boldsymbol{r}}_1 w_1(t)\| \leq Cs, \quad \|\boldsymbol{w}_2(t) - \hat{\boldsymbol{r}}_1 w_2(t)\| \leq Cs, \quad \left\| \boldsymbol{a}(t) - \hat{\boldsymbol{r}}_1 \sqrt{w_1(t)^2 + w_2(t)^2} \right\| \leq Cs. \tag{B.154}$$

$\square$

# C  Analysis of the Reduced Dynamics

In this section, we provide a detailed analysis of the reduced dynamics.

## C.1  Early Phase

In this subsection, to rigorously bound the trajectories and ensure that the linear driving force strictly dominates the interaction terms, we introduce a constant $R$ satisfying $\|\boldsymbol{\Sigma}_{xy}\| - \lambda_1 R^2 > 0$ and then define an exit time $T_R = \inf\{t \geq 0 : w_1(t)^2 + w_2(t)^2 \geq R^2\}$.

Furthermore, we define the region $\Omega_{B_1, B_2} = \{(w_1, w_2) : w_1 \geq -B_1 s, w_2 \geq -B_2 s\}$. Here, $B_1$ and $B_2$ are positive constants chosen so that $(w_1(0), w_2(0)) \in \Omega_{B_1, B_2}$ and satisfy $-\nu B_1 + \lambda_2 B_2 > 0$. This region is used only as an auxiliary device in the proof below.

The subsequent statements in this subsection restrict the analysis to the interval $t \in [0, T_R)$. We show that our analytical approximations for the early phase dynamics hold as long as the weights remain within this boundary.

### C.1.1  Bounding the Component $w_2$ (Proof of Proposition 3.4)

**Lemma C.1.** *Let $B_1$ and $B_2$ be positive constants such that $(w_1(0), w_2(0)) \in \Omega_{B_1, B_2}$ and $-\nu B_1 + \lambda_2 B_2 > 0$. Then for all sufficiently small $s > 0$, and all $t \geq 0$, we have $(w_1(t), w_2(t)) \in \Omega_{B_1, B_2}$.*

*Proof.* Suppose there exists a first exit time $t^* = \inf\{t \geq 0 : (w_1(t), w_2(t)) \notin \Omega_{B_1, B_2}\}$. Then, by continuity of $(w_1(t), w_2(t))$, we have $(w_1(t^*), w_2(t^*)) \in \partial\Omega_{B_1, B_2}$. At this time, the trajectory must leave $\Omega_{B_1, B_2}$ through either the left boundary $w_1 = -B_1 s$ or the lower boundary $w_2 = -B_2 s$.

First, consider the case where $w_1(t^*) \geq -B_1 s$ and $w_2(t^*) = -B_2 s$. Evaluating Equation (17) at $t = t^*$ and using $-\nu B_1 + \lambda_2 B_2 > 0$, we obtain

$$\begin{aligned}
\tau \frac{d}{dt} w_2(t^*) &= (w_1(t^*)^2 + w_2(t^*)^2)(\nu w_1(t^*) - \lambda_2 w_2(t^*)) \\
&\geq (w_1(t^*)^2 + w_2(t^*)^2)(-\nu B_1 + \lambda_2 B_2)s \\
&> 0.
\end{aligned} \tag{C.1}$$

Hence the trajectory cannot cross the boundary $w_2 = -B_2 s$ from inside to outside at time $t^*$, which is a contradiction.

Next, consider the case where $w_1(t^*) = -B_1 s$ and $w_2(t^*) \geq -B_2 s$. If $w_2(t^*) \geq 0$, then

$$\tau \frac{d}{dt} w_1(t^*) \geq \|\boldsymbol{\Sigma}_{xy}\| \sqrt{w_1(t^*)^2 + w_2(t^*)^2} \geq \|\boldsymbol{\Sigma}_{xy}\| B_1 s > 0. \tag{C.2}$$

If $-B_2 s \leq w_2(t^*) < 0$, then $w_1(t^*)^2 + w_2(t^*)^2 \leq (B_1^2 + B_2^2)s^2$, and hence

$$\tau \frac{d}{dt} w_1(t^*) = \|\boldsymbol{\Sigma}_{xy}\| \sqrt{w_1(t^*)^2 + w_2(t^*)^2} + (\lambda_1 B_1 s + \nu w_2(t^*))(w_1(t^*)^2 + w_2(t^*)^2)$$
$$\geq \|\boldsymbol{\Sigma}_{xy}\| B_1 s - \nu B_2 (B_1^2 + B_2^2)s^3. \tag{C.3}$$

Therefore, for all sufficiently small $s > 0$, we have $dw_1(t^*)/dt > 0$. Thus the trajectory cannot cross the boundary $w_1 = -B_1 s$ from inside to outside at time $t^*$, again a contradiction.

Since neither boundary can be crossed from inside to outside, the assumed exit time $t^*$ cannot exist. Therefore,

$$(w_1(t), w_2(t)) \in \Omega_{B_1, B_2} \tag{C.4}$$

for all $t \geq 0$. $\qquad\square$

**Lemma C.2.** *Let $B_1$ and $B_2$ be as in Lemma C.1. Then, for all sufficiently small $s > 0$, and all $t \in [0, T_R)$, we have*

$$\tau \frac{d}{dt} w_1 \geq \frac{1}{2}(\|\boldsymbol{\Sigma}_{xy}\| - \lambda_1 R^2)\sqrt{w_1^2 + w_2^2} > 0. \tag{C.5}$$

*Proof.* From Equation (16), using $(w_1, w_2) \in \Omega_{B_1, B_2}$ and $\sqrt{w_1^2 + w_2^2} \leq R$, for a sufficiently small $s > 0$, we have

$$\tau \frac{d}{dt} w_1 = \left( \|\boldsymbol{\Sigma}_{xy}\| - \lambda_1 w_1 \sqrt{w_1^2 + w_2^2} + \nu w_2 \sqrt{w_1^2 + w_2^2} \right) \sqrt{w_1^2 + w_2^2}$$
$$\geq (\|\boldsymbol{\Sigma}_{xy}\| - \lambda_1 R^2 - \nu B_2 R s)\sqrt{w_1^2 + w_2^2}$$
$$\geq \frac{1}{2}(\|\boldsymbol{\Sigma}_{xy}\| - \lambda_1 R^2)\sqrt{w_1^2 + w_2^2}$$
$$> 0. \tag{C.6}$$

Note that the strict inequality holds because the origin $(w_1, w_2) = (0, 0)$ is a fixed point of the system. By the uniqueness of solutions to ODEs, a trajectory starting from a non-zero initialization cannot reach the origin in finite time, ensuring $\sqrt{w_1^2 + w_2^2} > 0$. $\qquad\square$

**Lemma C.3.** *Let $B_1$ and $B_2$ be as in Lemma C.1. Then, there exists a constant $K > 0$ such that, for all sufficiently small $s > 0$ and all $t \in [0, T_R)$,*

$$w_2(t) \leq \frac{2\nu\sqrt{1 + (\nu/\lambda_2)^2}}{3(\|\boldsymbol{\Sigma}_{xy}\| - \lambda_1 R^2)} w_1(t)^3 + Ks. \tag{C.7}$$

*Proof.* First, observe that if $\nu w_1/\lambda_2 \leq w_2$, we trivially have $\tau dw_2 \leq 0$. Since $\tau dw_1/dt > 0$ from Lemma C.2, this implies $dw_2/dw_1 \leq 0$. Thus, for the purpose of upper-bounding $dw_2/dw_1$, we only need to consider the region where $\tau dw_2/dt > 0$, which strictly requires $w_2 < \nu w_1/\lambda_2$.

Assuming $w_2 < \nu w_1/\lambda_2$, since $(w_1, w_2) \in \Omega_{B_1, B_2}$, we also have $-w_2 \leq B_2 s$. Therefore,

$$\tau \frac{dw_2}{dt} = (w_1^2 + w_2^2)(\nu w_1 - \lambda_2 w_2) \leq (w_1^2 + w_2^2)(\nu w_1 + \lambda_2 B_2 s). \tag{C.8}$$

We now bound this expression by dividing into two cases based on the sign of $w_1$.

First, we consider the case where $-B_1 s \leq w_1 < 0$. Since $w_1 < 0$, we have $\nu w_1 + \lambda_2 B_2 s < \lambda_2 B_2 s$. Combining this with $\sqrt{w_1^2 + w_2^2} \leq R$, we obtain

$$\tau \frac{dw_2}{dt} \leq \lambda_2 B_2 s(w_1^2 + w_2^2) \leq \lambda_2 B_2 R s \sqrt{w_1^2 + w_2^2}. \tag{C.9}$$

Next, we consider the case where $0 \leq w_1$. In this region, $w_2$ is bounded by $-B_2 s \leq w_2 < \nu w_1/\lambda_2$. Therefore,

$$\sqrt{w_1^2 + w_2^2} \leq \sqrt{w_1^2 + \max\left(B_2 s, \frac{\nu}{\lambda_2} w_1\right)^2} \leq w_1 \sqrt{1 + \left(\frac{\nu}{\lambda_2}\right)^2} + B_2 s. \tag{C.10}$$

Substituting this into Equation (C.8), we have

$$\begin{aligned}
\tau \frac{dw_2}{dt} &\leq \sqrt{w_1^2 + w_2^2} \left(w_1 \sqrt{1 + \left(\frac{\nu}{\lambda_2}\right)^2} + B_2 s\right)(\nu w_1 + \lambda_2 B_2 s) \\
&\leq \left(\nu \sqrt{1 + \left(\frac{\nu}{\lambda_2}\right)^2} w_1^2 + B_3 s\right) \sqrt{w_1^2 + w_2^2}
\end{aligned} \tag{C.11}$$

where $B_3 \geq \lambda_2 B_2 R$ is a constant. Here, the cross terms $O(sw_1)$ are absorbed into the $O(s)$ term since $w_1 \leq R$. Note that this bound also encompasses the first case.

Therefore, dividing by Equation (C.5), we have

$$\frac{dw_2}{dw_1} \leq \frac{2\nu \sqrt{1 + (\nu/\lambda_2)^2}}{\|\boldsymbol{\Sigma}_{xy}\| - \lambda_1 R^2} w_1^2 + B_3 s. \tag{C.12}$$

for all $t \in [0, T_R)$. Integrating this differential inequality with respect to $w_1$ from $w_1(0)$ to $w_1(t)$, we obtain

$$w_2(t) - w_2(0) \leq \frac{2\nu \sqrt{1 + (\nu/\lambda_2)^2}}{3(\|\boldsymbol{\Sigma}_{xy}\| - \lambda_1 R^2)}(w_1(t)^3 - w_1(0)^3) + B_3 s(w_1(t) - w_1(0)) \tag{C.13}$$

Since $w_1(t) < R$ for all $t \in [0, T_R)$ and the initialization satisfies $w_1(0) = O(s)$ and $w_2(0) = O(s)$, the remaining terms on the right-hand side are all of order $O(s)$. Therefore, they can be absorbed into a single term $Ks$ for some constant $K > 0$, yielding Equation (C.7). $\square$

*Proof of Proposition 3.4.* From the definition of $\Omega_{B_1, B_2}$, we have the lower bound $w_2(t) \geq -B_2 s$. Combining this with the upper bound provided by Lemma C.3 and defining $C = \max\{K, B_2\}$, we immediately obtain

$$|w_2(t)| \leq \frac{2\nu \sqrt{1 + (\nu/\lambda_2)^2}}{3(\|\boldsymbol{\Sigma}_{xy}\| - \lambda_1 R^2)} |w_1(t)|^3 + Cs. \tag{C.14}$$

for all $t \in [0, T_R)$. $\square$

### C.1.2 Decoupled Dynamics of the Component $w_1$ (Proof of Proposition 3.5)

Building on the suppression of $w_2$ established above, we demonstrate that the evolution of $w_1$ can be effectively decoupled and approximated by a one-dimensional differential equation.

*Proof of Proposition 3.5.* We rewrite Equation (16) as:

$$\tau \frac{d}{dt} w_1(t) = \|\boldsymbol{\Sigma}_{xy}\| w_1(t) - \lambda_1 w_1(t)^3 + \varepsilon_R(t) \tag{C.15}$$

where

$$\varepsilon_R(t) = \|\boldsymbol{\Sigma}_{xy}\| \left(\sqrt{w_1^2 + w_2^2} - w_1\right) - \lambda_1 w_1 w_2^2 + \nu w_2(w_1^2 + w_2^2). \tag{C.16}$$

Therefore,

$$|\varepsilon_R(t)| \leq \|\boldsymbol{\Sigma}_{xy}\| \left|\sqrt{w_1^2 + w_2^2} - w_1\right| + \lambda_1 |w_1||w_2|^2 + \nu |w_2|(|w_1|^2 + |w_2|^2). \tag{C.17}$$

By Proposition 3.4, there exist constants $K_1, K_2 > 0$ such that $|w_2| \leq K_1|w_1|^3 + K_2 s$ for all $t \in [0, T_R)$. By the definition of $\Omega_{B_1,B_2}$, we have $w_1 \geq -B_1 s$. If $|w_1| \leq B_1 s$, we have

$$\left| \sqrt{w_1^2 + w_2^2} - w_1 \right| \leq \sqrt{w_1^2 + w_2^2} + |w_1| \leq 2|w_1| + |w_2| \leq 2B_1 s + K_1 B_1^3 s^3 + K_2 s \leq M_1 s. \tag{C.18}$$

If $w_1 > B_1 s$, then $w_1 > 0$ and hence

$$\left| \sqrt{w_1^2 + w_2^2} - w_1 \right| = \sqrt{w_1^2 + w_2^2} - w_1 = \frac{w_2^2}{\sqrt{w_1^2 + w_2^2} + w_1} \leq \frac{w_2^2}{2w_1} \leq M_2 w_1^5 + M_3 s. \tag{C.19}$$

Moreover, the remaining two terms are bounded by $M_4|w_1|^5 + M_5 s$, together with the early-phase bound $w_1^2 + w_2^2 < R^2$. Combining these bounds yields

$$|\varepsilon_R(t)| \leq C_1|w_1|^5 + C_2 s \tag{C.20}$$

for some constants $C_1, C_2 > 0$. $\qquad\square$

### C.1.3 Derivation of the Early Phase Timescale

Using the decoupled dynamics, we derive an explicit analytical approximation for the growth of $w_1$ and quantify the characteristic timescale of its sigmoidal transition.

From Proposition 3.5, if we neglect the remainder term $\varepsilon_R(t)$, we can define an idealized approximation of the trajectory, denoted by $w_1^{(0)}(t)$, which strictly follows the truncated dynamics:

$$\tau \frac{d}{dt} w_1^{(0)}(t) = \|\mathbf{\Sigma}_{xy}\| w_1^{(0)}(t) - \lambda_1 w_1^{(0)}(t)^3. \tag{C.21}$$

Solving this Bernoulli differential equation with the initial condition $w_1^{(0)}(0) = w_1(0)$ gives the exact closed-form solution:

$$w_1^{(0)}(t) = \sqrt{\frac{\|\mathbf{\Sigma}_{xy}\|}{\lambda_1} \left( 1 + \left( \frac{\|\mathbf{\Sigma}_{xy}\|/\lambda_1}{w_1(0)^2} - 1 \right) e^{-\frac{2\|\mathbf{\Sigma}_{xy}\|}{\tau} t} \right)^{-1}}. \tag{C.22}$$

As long as the residual $\varepsilon_R(t)$ remains small, this idealized solution closely tracks the true magnitude $w_1(t)$.

Let

$$w_{1\infty}^{(0)} := \sqrt{\frac{\|\mathbf{\Sigma}_{xy}\|}{\lambda_1}} \tag{C.23}$$

denote the asymptotic saturation value. To quantify the sigmoidal transition, fix any fraction $0 < p < 1$ and define $t_p$ as the time at which the approximation reaches this fraction:

$$w_1^{(0)}(t_p) = p w_{1\infty}^{(0)}. \tag{C.24}$$

Substituting this into Equation (C.22) gives

$$p^2 = \left( 1 + \left( \frac{(w_{1\infty}^{(0)})^2}{w_1(0)^2} - 1 \right) e^{-\frac{2\|\mathbf{\Sigma}_{xy}\|}{\tau} t_p} \right)^{-1}, \tag{C.25}$$

hence

$$t_p = \frac{\tau}{2\|\mathbf{\Sigma}_{xy}\|} \log \left[ \left( \frac{(w_{1\infty}^{(0)})^2}{w_1(0)^2} - 1 \right) \frac{p^2}{1 - p^2} \right]. \tag{C.26}$$

Using $(w_{1\infty}^{(0)})^2 = \|\mathbf{\Sigma}_{xy}\|/\lambda_1$, this can be rewritten as

$$t_p = \frac{\tau}{2\|\mathbf{\Sigma}_{xy}\|} \log \left[ \left( \frac{\|\mathbf{\Sigma}_{xy}\|}{\lambda_1 w_1(0)^2} - 1 \right) \frac{p^2}{1 - p^2} \right]. \tag{C.27}$$

When the initialization is small, with $w_1(0) = \Theta(s)$ for $s \ll 1$, the argument of the logarithm is dominated by the term $\|\mathbf{\Sigma}_{xy}\|/(\lambda_1 w_1(0)^2) \gg 1$. Thus, for any fixed fraction $p \in (0,1)$, the time $t_p$ can be asymptotically expanded as:

$$t_p = \frac{\tau}{2\|\mathbf{\Sigma}_{xy}\|} \log\left(\frac{\|\mathbf{\Sigma}_{xy}\|}{\lambda_1 w_1(0)^2}\right) + O(1). \tag{C.28}$$

Expressing this in terms of the initialization scale $s$, the leading term is:

$$\frac{\tau}{2\|\mathbf{\Sigma}_{xy}\|} \log\left(\frac{\|\mathbf{\Sigma}_{xy}\|}{\lambda_1 s^2}\right). \tag{C.29}$$

Crucially, the choice of the specific fraction $p$ affects only the additive $O(1)$ constant, leaving the leading-order logarithmic scaling invariant. This justifies the characteristic timescale for the early phase presented in the main text.

## C.2 Later Phase

In this subsection, we provide the derivation of the upper bound on the timescale for the later phase given in the main text.

### C.2.1 Equilibrium, Stability, and Convergence of the Reduced Two-Variable System

Before deriving the later-phase timescale, we first establish the equilibrium and convergence properties of the reduced two-variable system (Equations (16) and (17)). Let

$$\mathbf{H} := \begin{pmatrix} \lambda_1 & -\nu \\ -\nu & \lambda_2 \end{pmatrix}. \tag{C.30}$$

Since $\mathbf{H}$ is the restriction of the covariance matrix $\mathbf{\Sigma}$ to the span of $\{\mathbf{v}_1, \mathbf{v}_2\}$, it is positive definite. In the present spiked covariance setting, $\mathbf{H}$ has eigenvalues $\sigma^2$ and $\sigma^2(1+\rho)$. In particular, $\lambda_{\min}(\mathbf{H}) = \sigma^2$ and $\lambda_{\max}(\mathbf{H}) = \sigma^2(1+\rho)$. Write the target vector as $\boldsymbol{\beta} = c_1\mathbf{v}_1 + c_2\mathbf{v}_2$ and set $\mathbf{c} := (c_1, c_2)^\top$. Then

$$\mathbf{H}\mathbf{c} = (\|\mathbf{\Sigma}_{xy}\|, 0)^\top, \quad \|\mathbf{c}\| = 1. \tag{C.31}$$

With $a(t) := \sqrt{w_1(t)^2 + w_2(t)^2}$ and $\mathbf{w}(t) := (w_1(t), w_2(t))^\top$, the reduced two-variable system can be written as

$$\tau \frac{d}{dt}\mathbf{w} = a\mathbf{H}(\mathbf{c} - a\mathbf{w}). \tag{C.32}$$

**Lemma C.4.** *The reduced system defined by Equation* (C.32) *has a unique nonzero equilibrium* $\mathbf{c}$*, which is locally exponentially stable.*

*Proof.* If $\mathbf{w} \neq 0$ is an equilibrium, then $a > 0$ and Equation (C.32) gives $\mathbf{c} = a\mathbf{w}$. Taking norms and using $\|\mathbf{c}\| = 1$ and $a = \|\mathbf{w}\|$, we obtain $1 = a\|\mathbf{w}\| = a^2$. Hence $a = 1$ and $\mathbf{w} = \mathbf{c}$. Thus $\mathbf{c}$ is the unique nonzero equilibrium.

To check local stability, let $\mathbf{F}(\mathbf{w}) := a\mathbf{H}(\mathbf{c} - a\mathbf{w})$ denote the vector field in Equation (C.32). At $\mathbf{w} = \mathbf{c}$, we have $a = 1$ and $\nabla_{\mathbf{w}} a|_{\mathbf{w}=\mathbf{c}} = \mathbf{c}$, so the Jacobian is

$$J_{\mathbf{F}}(\mathbf{c}) = -\mathbf{H}(\mathbf{I} + \mathbf{c}\mathbf{c}^\top). \tag{C.33}$$

Since both $\mathbf{H}$ and $\mathbf{I} + \mathbf{c}\mathbf{c}^\top$ are symmetric positive definite, their product $\mathbf{H}(\mathbf{I} + \mathbf{c}\mathbf{c}^\top)$ has positive eigenvalues. Therefore all eigenvalues of $J_{\mathbf{F}}(\mathbf{c})$ have strictly negative real parts, which proves local exponential stability. $\square$

In the following, $\mathbf{w}(t)$ denotes the solution of Equation (C.32) with the initial condition in Equation (18), namely the reduced trajectory defined in Theorem 3.3. For this trajectory, define

$$\boldsymbol{\theta}(t) := a(t)\mathbf{w}(t), \quad \mathcal{L}_2(t) := (\mathbf{c} - \boldsymbol{\theta}(t))^\top \mathbf{H}(\mathbf{c} - \boldsymbol{\theta}(t)), \quad \mathcal{L}_{2,0} := \mathbf{c}^\top \mathbf{H}\mathbf{c}. \tag{C.34}$$

**Lemma C.5.** *Let $R > 0$ satisfy $\frac{2\nu\sqrt{1+(\nu/\lambda_2)^2}}{3(\|\mathbf{\Sigma}_{xy}\|-\lambda_1 R^2)}R^2 < \frac{1}{4}$ and $\eta_R := \sqrt{3}\|\mathbf{\Sigma}_{xy}\|R^2 - \lambda_{\max}(\mathbf{H})R^4 > 0$. Then, for all sufficiently small $s > 0$, $\mathcal{L}_2(T_R) \leq \mathcal{L}_{2,0} - \eta_R$.*

*Proof.* Set $K_R := \frac{2\nu\sqrt{1+(\nu/\lambda_2)^2}}{3(\|\mathbf{\Sigma}_{xy}\|-\lambda_1 R^2)}$. By Proposition 3.4, there exists a constant $C > 0$ such that $|w_2(t)| \leq K_R|w_1(t)|^3 + Cs \leq K_R R^3 + Cs$ for all $t \in [0, T_R]$. Since $K_R R^2 < 1/4$, taking $s > 0$ sufficiently small gives $|w_2(t)| \leq R/2$ on $[0, T_R]$. Using continuity, we also have $|w_2(T_R)| \leq R/2$. Because $\|\mathbf{w}(T_R)\| = R$, this implies $|w_1(T_R)| \geq \sqrt{3}R/2$. Lemma C.1 gives $w_1(T_R) \geq -B_1 s$. For all sufficiently small $s > 0$, the negative alternative is impossible, and hence $w_1(T_R) \geq \sqrt{3}R/2$.

Using $\boldsymbol{\theta}(T_R) = R\mathbf{w}(T_R)$, $\|\boldsymbol{\theta}(T_R)\| = R^2$, and Equation (C.31), we obtain

$$
\begin{aligned}
\mathcal{L}_{2,0} - \mathcal{L}_2(T_R) &= 2\mathbf{c}^\top \mathbf{H}\boldsymbol{\theta}(T_R) - \boldsymbol{\theta}(T_R)^\top \mathbf{H}\boldsymbol{\theta}(T_R) \\
&= 2\|\mathbf{\Sigma}_{xy}\|Rw_1(T_R) - \boldsymbol{\theta}(T_R)^\top \mathbf{H}\boldsymbol{\theta}(T_R) \\
&\geq \sqrt{3}\|\mathbf{\Sigma}_{xy}\|R^2 - \lambda_{\max}(\mathbf{H})R^4 \\
&= \eta_R.
\end{aligned}
\tag{C.35}
$$

$\square$

**Lemma C.6.** *For all sufficiently small $s > 0$, the solution $\mathbf{w}(t)$ satisfies $\mathbf{w}(t) \to \mathbf{c}$ as $t \to \infty$.*

*Proof.* Since $a = \|\mathbf{w}\|$, we have $\|\boldsymbol{\theta}\| = a^2$. The dynamics in Equation (C.32) imply

$$
\tau \frac{d}{dt}a = \mathbf{w}^\top \mathbf{H}(\mathbf{c} - a\mathbf{w}).
\tag{C.36}
$$

Therefore,

$$
\tau \frac{d}{dt}\mathcal{L}_2(t) = -2\left(\mathbf{w}^\top \mathbf{H}(\mathbf{c} - \boldsymbol{\theta})\right)^2 - 2a^2\|\mathbf{H}(\mathbf{c} - \boldsymbol{\theta})\|^2 \leq 0.
\tag{C.37}
$$

Thus $\mathcal{L}_2(t)$ is nonincreasing.

Let $R$, $\eta_R$, and $T_R$ be as in Lemma C.5. By the monotonicity of $\mathcal{L}_2(t)$ and the conclusion of Lemma C.5, for all $t \geq T_R$,

$$
\eta_R \leq \mathcal{L}_{2,0} - \mathcal{L}_2(t) = 2\mathbf{c}^\top \mathbf{H}\boldsymbol{\theta} - \boldsymbol{\theta}^\top \mathbf{H}\boldsymbol{\theta} \leq 2\|\mathbf{H}\mathbf{c}\|\|\boldsymbol{\theta}\|.
\tag{C.38}
$$

Using $\delta_R := \eta_R/(2\|\mathbf{H}\mathbf{c}\|) > 0$, we have $a(t)^2 = \|\boldsymbol{\theta}(t)\| \geq \delta_R$ for all $t \geq T_R$.

Since $\|\mathbf{H}(\mathbf{c} - \boldsymbol{\theta})\|^2 \geq \lambda_{\min}(\mathbf{H})\mathcal{L}_2(t)$, Inequality (C.37) gives, for all $t \geq T_R$,

$$
\tau \frac{d}{dt}\mathcal{L}_2(t) \leq -2\delta_R \lambda_{\min}(\mathbf{H})\mathcal{L}_2(t).
\tag{C.39}
$$

This differential inequality implies $\mathcal{L}_2(t) \to 0$, so $\boldsymbol{\theta}(t) \to \mathbf{c}$. Since $\boldsymbol{\theta} = a\mathbf{w}$, $a = \|\mathbf{w}\|$, and $\|\mathbf{c}\| = 1$, we also have $a(t)^2 = \|\boldsymbol{\theta}(t)\| \to 1$, and hence $\mathbf{w}(t) = \boldsymbol{\theta}(t)/a(t) \to \mathbf{c}$. $\square$

This proves the existence of the limit $\mathbf{w}_\infty := \lim_{t\to\infty} \mathbf{w}(t)$. By Lemma C.6, we have $\mathbf{w}_\infty = \mathbf{c}$.

**Corollary C.7.** *For all $\varepsilon_\infty > 0$ and all sufficiently small $s > 0$, $T_{\varepsilon_\infty} = \inf\{t \geq 0 : \|\mathbf{w}(t) - \mathbf{w}_\infty\| \leq \varepsilon_\infty\}$ is finite.*

*Proof.* Fix $\varepsilon_\infty > 0$. For all sufficiently small $s > 0$, Lemma C.6 gives $\mathbf{w}(t) \to \mathbf{w}_\infty$. Hence there exists a finite time $T$ such that $\|\mathbf{w}(t) - \mathbf{w}_\infty\| \leq \varepsilon_\infty$ for all $t \geq T$. Hence $T_{\varepsilon_\infty}$ is finite. $\square$

**Corollary C.8.** *For all sufficiently small $R > 0$ and all $\varepsilon_\infty > 0$, there exists $M_{R,\varepsilon_\infty} < \infty$ such that, for all sufficiently small $s > 0$ and all $T_\star \geq 0$ satisfying $\|\mathbf{w}(T_\star)\| \geq R$, $T_{\varepsilon_\infty} - T_\star \leq M_{R,\varepsilon_\infty}$.*

*Proof.* Fix a sufficiently small $R > 0$ and $\varepsilon_\infty > 0$. Let $T_R$ and $\eta_R$ be as in Lemma C.5, and set $\alpha_R := \frac{\eta_R \lambda_{\min}(\boldsymbol{H})}{\|\boldsymbol{H}\boldsymbol{c}\|\tau} > 0$. Define

$$\delta_{\varepsilon_\infty} := \min\left\{\frac{1}{2}, \frac{\varepsilon_\infty}{2\sqrt{2}}\right\}, \quad \gamma_{\varepsilon_\infty} := \lambda_{\min}(\boldsymbol{H})\delta_{\varepsilon_\infty}^2. \tag{C.40}$$

Integrating Inequality (C.39) from $T_R$ to $t$ and using $\mathcal{L}_2(T_R) \leq \mathcal{L}_{2,0}$, we obtain, for all $t \geq T_R$,

$$\mathcal{L}_2(t) \leq \mathcal{L}_{2,0}\exp\left(-\alpha_R(t - T_R)\right). \tag{C.41}$$

Therefore, with

$$M_{R,\varepsilon_\infty} := \frac{1}{\alpha_R}\max\left\{0, \log\frac{\mathcal{L}_{2,0}}{\gamma_{\varepsilon_\infty}}\right\}, \tag{C.42}$$

we have $\mathcal{L}_2(t) \leq \gamma_{\varepsilon_\infty}$ for all $t \geq T_R + M_{R,\varepsilon_\infty}$.

For all sufficiently small $s > 0$, $\|\boldsymbol{w}(0)\| < R$. Hence, if $\|\boldsymbol{w}(T_\star)\| \geq R$, then continuity gives $T_R \leq T_\star$. Hence every $t \geq T_\star + M_{R,\varepsilon_\infty}$ also satisfies $t \geq T_R + M_{R,\varepsilon_\infty}$, and therefore $\mathcal{L}_2(t) \leq \gamma_{\varepsilon_\infty}$. By the definition of $\mathcal{L}_2$ and the positive definiteness of $\boldsymbol{H}$, we obtain

$$\lambda_{\min}(\boldsymbol{H})\|\boldsymbol{\theta}(t) - \boldsymbol{c}\|^2 \leq \mathcal{L}_2(t) \leq \gamma_{\varepsilon_\infty} = \lambda_{\min}(\boldsymbol{H})\delta_{\varepsilon_\infty}^2. \tag{C.43}$$

Hence $\|\boldsymbol{\theta}(t) - \boldsymbol{c}\| \leq \delta_{\varepsilon_\infty}$. Since $\|\boldsymbol{c}\| = 1$ and $\delta_{\varepsilon_\infty} \leq 1/2$, this implies $\|\boldsymbol{\theta}(t)\| \geq 1/2$. Using $\boldsymbol{\theta} = a\boldsymbol{w}$ and $a = \|\boldsymbol{w}\| = \sqrt{\|\boldsymbol{\theta}\|}$, we get

$$
\begin{aligned}
\|\boldsymbol{w}(t) - \boldsymbol{c}\| &= \left\|\frac{\boldsymbol{\theta}(t)}{\sqrt{\|\boldsymbol{\theta}(t)\|}} - \boldsymbol{c}\right\| \\
&= \left\|\frac{\boldsymbol{\theta}(t) - \boldsymbol{c}}{\sqrt{\|\boldsymbol{\theta}(t)\|}} + \left(\frac{1}{\sqrt{\|\boldsymbol{\theta}(t)\|}} - 1\right)\boldsymbol{c}\right\| \\
&\leq \frac{\|\boldsymbol{\theta}(t) - \boldsymbol{c}\|}{\sqrt{\|\boldsymbol{\theta}(t)\|}} + \frac{|1 - \|\boldsymbol{\theta}(t)\||}{\sqrt{\|\boldsymbol{\theta}(t)\|}\left(1 + \sqrt{\|\boldsymbol{\theta}(t)\|}\right)} \\
&\leq 2\sqrt{2}\,\|\boldsymbol{\theta}(t) - \boldsymbol{c}\| \\
&\leq \varepsilon_\infty.
\end{aligned}
\tag{C.44}
$$

Here we used $|1 - \|\boldsymbol{\theta}(t)\|| \leq \|\boldsymbol{\theta}(t) - \boldsymbol{c}\|$ and $\|\boldsymbol{\theta}(t)\| \geq 1/2$. The last inequality follows from $\|\boldsymbol{\theta}(t) - \boldsymbol{c}\| \leq \delta_{\varepsilon_\infty}$ and $\delta_{\varepsilon_\infty} \leq \varepsilon_\infty/(2\sqrt{2})$. Since $\boldsymbol{w}_\infty = \boldsymbol{c}$, this gives $\|\boldsymbol{w}(t) - \boldsymbol{w}_\infty\| \leq \varepsilon_\infty$ for all $t \geq T_\star + M_{R,\varepsilon_\infty}$, and thus $T_{\varepsilon_\infty} \leq T_\star + M_{R,\varepsilon_\infty}$. This proves the claimed upper bound. $\square$

### C.2.2 Derivation of the Later Phase Timescale (Proof of Proposition 3.6)

*Proof of Proposition 3.6.* Under the assumptions, we have $w_1(t) \geq w_{1\infty}$ and $w_2(t) < w_{2\infty}$. Using these inequalities, we can establish a lower bound on the derivative of $w_2$:

$$\tau\frac{dw_2}{dt} = (w_1^2 + w_2^2)(\nu w_1 - \lambda_2 w_2) \geq (w_{1\infty}^2 + w_2^2)(\nu w_{1\infty} - \lambda_2 w_2). \tag{C.45}$$

By Lemmas C.4 and C.6, the nonzero equilibrium of the reduced system is $\boldsymbol{w}_\infty = \boldsymbol{c}$. In particular, $\|\boldsymbol{w}_\infty\| = \|\boldsymbol{c}\| = 1$. Moreover, the second component of Equation (C.31), or equivalently Equation (17) at equilibrium, gives $\nu w_{1\infty} - \lambda_2 w_{2\infty} = 0$. Hence $w_{2\infty} = \nu w_{1\infty}/\lambda_2$, and we obtain

$$w_{1\infty}^2\left(1 + \left(\frac{\nu}{\lambda_2}\right)^2\right) = 1. \tag{C.46}$$

Together with the assumption $w_2(t) < w_{2\infty}$, this gives

$$\nu w_{1\infty} - \lambda_2 w_2(t) = \lambda_2(w_{2\infty} - w_2(t)) > 0. \tag{C.47}$$

Hence $dw_2/dt > 0$ on $[t_1, t_2]$, so $w_2$ can be used as an integration variable. Therefore,

$$
\begin{aligned}
t - t_1 &= \tau \int_{w_2(t_1)}^{w_2(t)} \frac{dw_2}{(w_1^2 + w_2^2)(\nu w_1 - \lambda_2 w_2)} \\
&\leq \tau \int_{w_2(t_1)}^{w_2(t)} \frac{dw_2}{(w_{1\infty}^2 + w_2^2)(\nu w_{1\infty} - \lambda_2 w_2)} \\
&= \tau \frac{1}{\lambda_2 w_{1\infty}^2 \left(1 + \left(\frac{\nu}{\lambda_2}\right)^2\right)} \int_{w_2(t_1)}^{w_2(t)} \left(\frac{w_2 + \frac{\nu}{\lambda_2} w_{1\infty}}{w_2^2 + w_{1\infty}^2} - \frac{1}{w_2 - \frac{\nu}{\lambda_2} w_{1\infty}}\right) dw_2 \\
&= \frac{\tau}{\lambda_2} \left[\frac{\nu}{\lambda_2} \arctan\left(\frac{w_2}{w_{1\infty}}\right) + \frac{1}{2} \log(w_2^2 + w_{1\infty}^2) - \log\left|w_2 - \frac{\nu}{\lambda_2} w_{1\infty}\right|\right]_{w_2(t_1)}^{w_2(t)}.
\end{aligned}
\tag{C.48}
$$

Let us evaluate the integral using the antiderivative $G(w_2)$:

$$
G(w_2) := \frac{\nu}{\lambda_2} \arctan\left(\frac{w_2}{w_{1\infty}}\right) + \frac{1}{2} \log(w_2^2 + w_{1\infty}^2) - \log(w_{2\infty} - w_2).
\tag{C.49}
$$

Since the integrand is strictly positive for $w_2 \in [0, w_{2\infty})$, $G(w_2)$ is a strictly increasing function. Therefore, for any fixed $\varepsilon > 0$, we have $G(\varepsilon) > G(0)$. Thus,

$$
t_2 - t_1 \leq \frac{\tau}{\lambda_2}(G(w_2(t_2)) - G(\varepsilon)) < \tau \frac{1}{\lambda_2}(G(w_2(t_2)) - G(0)).
\tag{C.50}
$$

Evaluating $G(w_2)$ at the upper bound $w_2(t_2) = (1 - e^{-\delta}) w_{2\infty}$ yields:

$$
G(w_2(t_2)) = \frac{\nu}{\lambda_2} \arctan\left(\frac{(1 - e^{-\delta}) w_{2\infty}}{w_{1\infty}}\right) + \frac{1}{2} \log((1 - e^{-\delta})^2 w_{2\infty}^2 + w_{1\infty}^2) - \log(w_{2\infty} - (1 - e^{-\delta}) w_{2\infty}).
\tag{C.51}
$$

Evaluating $G(w_2)$ at 0, we get

$$
G(0) = \frac{1}{2} \log(w_{1\infty}^2) - \log(w_{2\infty}).
\tag{C.52}
$$

Subtracting $G(0)$ from $G(w_2(t_2))$, we obtain

$$
G(w_2(t_2)) - G(0) = \frac{\nu}{\lambda_2} \arctan\left(\frac{\nu}{\lambda_2}(1 - e^{-\delta})\right) + \frac{1}{2} \log\left(\frac{w_{1\infty}^2 + w_{2\infty}^2(1 - e^{-\delta})^2}{w_{1\infty}^2}\right) + \delta.
\tag{C.53}
$$

Using the fixed-point relation $w_{2\infty}/w_{1\infty} = \nu/\lambda_2$, we can obtain the bound in Inequality (23):

$$
t_2 - t_1 \leq \frac{\tau}{\lambda_2}\left[\frac{\nu}{\lambda_2} \arctan\left(\frac{\nu}{\lambda_2}(1 - e^{-\delta})\right) + \frac{1}{2} \log\left(1 + \left(\frac{\nu}{\lambda_2}(1 - e^{-\delta})\right)^2\right) + \delta\right].
\tag{C.54}
$$

$\square$

# D  Derivation of the Generalization Error

*Proof of Theorem 4.1.* For $\boldsymbol{\Sigma} = \sigma^2(\boldsymbol{I} + \rho \boldsymbol{\mu}\boldsymbol{\mu}^\top)$, the eigenvalues are

$$
\lambda_1 = \sigma^2(1 + \rho), \quad \lambda_i = \sigma^2 \ (i = 2, \ldots, d).
\tag{D.1}
$$

Substituting this spectrum into Equation (28) gives

$$
\begin{aligned}
1 &= \frac{\lambda}{\kappa(\lambda)} + \frac{1}{n} \sum_{i=1}^{d} \frac{\lambda_i}{\kappa(\lambda) + \lambda_i} \\
&= \frac{\lambda}{\kappa(\lambda)} + \frac{d}{n} \frac{1}{d} \sum_{i=1}^{d} \frac{\lambda_i}{\kappa(\lambda) + \lambda_i} \\
&= \frac{\lambda}{\kappa(\lambda)} + \frac{d}{n} \left(\frac{d - 1}{d} \frac{\sigma^2}{\kappa(\lambda) + \sigma^2} + \frac{1}{d} \frac{\sigma^2(1 + \rho)}{\kappa(\lambda) + \sigma^2(1 + \rho)}\right)
\end{aligned}
\tag{D.2}
$$

In the high-dimensional limit $n, d \to \infty$ with $d/n \to \gamma$, the spike term is weighted by $1/d$ and therefore vanishes. Hence,

$$
\begin{aligned}
1 &= \frac{\lambda}{\kappa(\lambda)} + \gamma \frac{\sigma^2}{\kappa(\lambda) + \sigma^2} \\
&= \frac{(\kappa(\lambda) + \sigma^2)\lambda + \gamma \sigma^2 \kappa(\lambda)}{\kappa(\lambda)(\kappa(\lambda) + \sigma^2)}.
\end{aligned}
\tag{D.3}
$$

Rearranging the terms, we obtain the quadratic equation:

$$
\kappa(\lambda)^2 - (\lambda + (\gamma - 1)\sigma^2)\kappa(\lambda) - \sigma^2 \lambda = 0.
\tag{D.4}
$$

Define $\kappa_0 := \lim_{\lambda \to 0^+} \kappa(\lambda)$. Taking $\lambda \to 0^+$ in Equation (D.4) gives

$$
\kappa_0^2 - (\gamma - 1)\sigma^2 \kappa_0 = 0.
\tag{D.5}
$$

Since $\gamma > 1$ and $\kappa_0 > 0$, we obtain:

$$
\kappa_0 = \sigma^2(\gamma - 1).
\tag{D.6}
$$

Next, differentiate Equation (D.4) with respect to $\lambda$:

$$
2\kappa(\lambda)\frac{d\kappa(\lambda)}{d\lambda} - \kappa(\lambda) - (\gamma - 1)\sigma^2 \frac{d\kappa(\lambda)}{d\lambda} - \sigma^2 = 0.
\tag{D.7}
$$

Therefore,

$$
\frac{d\kappa(\lambda)}{d\lambda} = \frac{\kappa(\lambda) + \sigma^2}{2\kappa(\lambda) - \sigma^2(\gamma - 1)}.
\tag{D.8}
$$

Taking the limit $\lambda \to 0^+$, we obtain

$$
\lim_{\lambda \to 0^+} \frac{d\kappa(\lambda)}{d\lambda} = \frac{\kappa_0 + \sigma^2}{2\kappa_0 - \sigma^2(\gamma - 1)} = \frac{\gamma}{\gamma - 1}.
\tag{D.9}
$$

Finally, since $A = \boldsymbol{\mu}^\top \boldsymbol{\beta}$, we decompose $\boldsymbol{\beta}$ as follows.

If $0 \le A < 1$, then

$$
\boldsymbol{\beta} = A\boldsymbol{\mu} + \sqrt{1 - A^2}\boldsymbol{u}_2,
\tag{D.10}
$$

where

$$
\boldsymbol{u}_2 = \frac{(\boldsymbol{I} - \boldsymbol{\mu}\boldsymbol{\mu}^\top)\boldsymbol{\beta}}{\|(\boldsymbol{I} - \boldsymbol{\mu}\boldsymbol{\mu}^\top)\boldsymbol{\beta}\|}
\tag{D.11}
$$

is a unit vector orthogonal to $\boldsymbol{\mu}$.

If $A = 1$, then $\boldsymbol{\beta} = \boldsymbol{\mu}$. In this case, let $\boldsymbol{u}_2$ be any unit vector orthogonal to $\boldsymbol{\mu}$.

Therefore, with $\boldsymbol{u}_1 = \boldsymbol{\mu}$,

$$
(\boldsymbol{\beta}^\top \boldsymbol{u}_1)^2 = A^2, \quad (\boldsymbol{\beta}^\top \boldsymbol{u}_2)^2 = 1 - A^2, \quad \lambda_1 = \sigma^2(1 + \rho), \quad \lambda_2 = \sigma^2.
\tag{D.12}
$$

Taking $\lambda \to 0^+$ in Equation (27) and using Equations (D.6), (D.9) and (D.12), we obtain

$$
\begin{aligned}
\hat{R}_0 &= \lim_{\lambda \to 0^+} \hat{R}_\lambda \\
&= \frac{\gamma}{\gamma - 1} \cdot (\sigma^2(\gamma - 1))^2 \cdot \left( \frac{\sigma^2(1 + \rho)}{(\sigma^2(\gamma - 1) + \sigma^2(1 + \rho))^2} A^2 + \frac{\sigma^2}{(\sigma^2(\gamma - 1) + \sigma^2)^2}(1 - A^2) \right) \\
&= \sigma^2 \gamma(\gamma - 1)\left( \frac{1 + \rho}{(\gamma + \rho)^2}A^2 + \frac{1}{\gamma^2}(1 - A^2) \right) \\
&= \sigma^2\left(1 - \frac{1}{\gamma}\right)\left( \frac{1 + \rho}{(1 + \rho/\gamma)^2}A^2 + (1 - A^2) \right) \\
&= \sigma^2\left(1 - \frac{1}{\gamma}\right)\left[ 1 - A^2\left(1 - \frac{1 + \rho}{(1 + \rho/\gamma)^2}\right) \right].
\end{aligned}
\tag{D.13}
$$

$\square$

