# OpenReview forum: "The Impact of Anisotropic Covariance Structure on the Training Dynamics and Generalization Error of Linear Networks"
_TMLR — Rejected by TMLR_

### Review · Reviewer_6W1n · 2026-02-05

**Summary Of Contributions:**

The submitted work analyzes the learning dynamics and generalization error of a two-layer linear network on a linear regression task with anisotropic inputs. The inputs $x$ are defined using a covariance matrix $\Sigma = \sigma^2 (I + \rho \mu^T \mu)$, which has one eigenvalue $\sigma^2 (1+ \rho)$ and remaining eigenvalues $\sigma^2$. The outputs are given by $\beta x$. The authors first study the learning dynamics starting from small initializations and show that the training exhibits a two-phase dynamics. In the early training phase, the weights grow along the input-output correlation direction, whereas in the late training phase, which is driven by the anisotropy of the data. The authors also analyze the generalization error, revealing that generalization is improved if anisotropy is aligned with the direction of the target vector $\beta$ of the task.

**Additional Comments:**

Minor comments:
* It would be nice if the authors could discuss what they mean by anisotropy in the introduction itself
* Figures 4 and 5 can be moved up

**Audience:**

Yes

**Audience Explanation:**

Yes. The work would be interesting to the learning dynamics community, as the case of anisotropy is not studied in prior works.

**Claims And Evidence:**

Yes

**Claims Explanation:**

Yes. The paper claims to analyze the learning dynamics and generalization error of a two-layer linear model trained on a linear regression task with anisotropic data. The paper thoroughly analyzes the dynamics by demonstrating a two-phase learning dynamics with the latter phase driven by the anisotropy. The authors acknowledge limitations, which include restrictions to a particular type of anisotropy and small initializations.

**Requested Changes:**

None

---

> ### Author Response · Authors · 2026-04-08
> **Response to Reviewer 6W1n**
>
> We sincerely thank you for your highly positive review and for your clear, accurate summary of our contributions.
>
> While you did not request any major changes, we would like to direct your attention to our General Response to All Reviewers. Based on feedback from the other reviewers, we have substantially restructured the theoretical sections to clearly separate formal mathematical statements (Theorems, Propositions, Lemmas) from informal intuition, and we have expanded all proofs in the Appendix. We believe these structural improvements make the paper even stronger and easier to follow.
>
> Regarding your helpful minor comments, we have fully addressed both of them in the revised manuscript:
>
> - **Anisotropy in the Introduction:** We have added a clearer discussion explaining what we mean by data anisotropy directly in the Introduction to better orient the reader from the start.
> - **Placement of Figures 4 and 5:** We have adjusted the formatting and successfully moved Figures 4 and 5 further up in the text so that they appear closer to the relevant theoretical discussions, improving the overall readability.

---

### Review · Reviewer_kV9Q · 2026-02-12

**Summary Of Contributions:**

This paper investigates how input anisotropy, modeled by a spiked covariance $\Sigma = \sigma^2(I+\rho \mu\mu^\top) = \sigma^2(1+\rho) \mu\mu^\top + \sigma^2 \sum_{\nu \text{ basis of } span(\mu)^{\perp} } \nu\nu^\top$, affects (i) the training dynamics of a two-layer linear network trained by gradient descent on linear regression, and (ii) the generalization error in a high-dimensional regime. The data model is linear $y =\beta^\top x, x \sim \mathcal{N}(0,\Sigma)$; with anisotropy controlled by the spike magnitude $\rho$ and a spike–task alignment parameter $A=\mu^\top \beta$, and with $\Sigma_{xy}=\mathbb{E}[yx]=\Sigma\beta$ as the input–output correlation vector.

On the dynamics side, the paper argues for a two-phase training behavior: an early phase dominated by alignment with the label-correlated direction, followed by a later phase whose timescale depends on variance in other principal directions. On the generalization side, the paper works in the regime $n,d\to\infty$ with $\gamma=d/n>1$ and leverages standard ridgeless / random-matrix-theory arguments to derive a closed-form expression for the test error under the spiked model.

## Strengths

* The topic is relevant: understanding how covariance geometry (isotropic vs anisotropic data) impacts training and generalization is important both theoretically and practically.
* The spiked model provides a clean, interpretable parameterization of anisotropy via $\rho$ and alignment $A$.
* The paper provides an explicit generalization expression under the spiked model and supports it with numerical experiments.

## Weaknesses

* The paper’s presentation is poor for a theory-oriented submission: it is often difficult to distinguish what is formal from what is informal, and it is not always easy to locate where a given derivation is actually proved.
* Many derivations are overly compressed, which makes verification unnecessarily time-consuming.
* The dynamics analysis appears to use a regime like $n\to\infty$ with finite $d$, while the generalization analysis uses $n,d\to\infty$ with $\gamma>1$. The connection between these regimes is not clearly established.
* The “two-phase” perspective in two-layer linear networks and the role of anisotropic data overlap with prior work (notably Atanasov et al., 2022). Similarly, the generalization part is standard ridgeless formulas with anisotropic covariance “plugged in.” The paper should clarify precisely what is nontrivial in its contribution.

**Additional Comments:**

The paper would benefit enormously from a rewrite in a formal mathematical style. In its current form, the main obstacle is not “technical correctness” but a lack of clarity and discoverability of proof.

**Audience:**

Yes

**Audience Explanation:**

This work will interest parts of the TMLR audience working on:

* optimization and learning dynamics in (deep) linear networks,
* implicit bias and alignment phenomena under gradient descent,
* the role of whitening/preconditioning/covariance geometry in learning.

Even in a stylized setting, a clean analysis of anisotropy can provide useful conceptual guidance.

**Broader Impact Concerns:**

No major broader impact concerns are apparent. The work is theoretical and does not propose direct deployment-facing systems.

**Claims And Evidence:**

Yes

**Claims Explanation:**

The main claims are (i) a two-phase training dynamics picture under anisotropic covariance, and (ii) a closed-form generalization prediction under the spiked covariance model.


* For the dynamics claims, the story is plausible and consistent with known “silent alignment / two-phase” analyses in deep linear networks under small initialization. However, the paper’s presentation makes (the derivations of) these arguments a bit harder to audit than necessary.

* For the generalization claims, the approach is standard but credible: in the high-dimensional regime $\gamma=d/n>1$, one can characterize the ridgeless solution and its test error via known random-matrix-theory expressions, then specialize to the spiked model. The numerical results appear consistent with these predictions.

Overall, I checked all the derivations/claims and believe they are correct, although the evidence is not presented in a sufficiently transparent “theory paper” style.

**Requested Changes:**

# 1) Critical (required for acceptance)

### (i) Rewrite the theory exposition in a standard structure (assumptions $\rightarrow$ results $\rightarrow$ proofs)

This is the main issue. For a paper positioning itself as theoretical, the current presentation makes it difficult to separate:
* formal statements vs informal intuition,
* proved claims vs heuristic derivations,
* and where specific steps are justified.

I strongly recommend restructuring the paper as:
* Assumptions (model, initialization regime, scaling limits, definitions),
* Lemma/Proposition/Theorem statements (clearly labeled),
* Proofs in the appendix, with explicit pointers in the main text (not only “see Appendix B”, but “Proof of Proposition 2 is in Appendix B.3”).

Informal discussion should be clearly marked as commentary, not blended with formal derivation.

### (ii) Expand derivations (do not skip intermediate steps)

Many derivations appear correct but are written in an unnecessarily condensed way. Since TMLR does not impose strict appendix length limits, the paper should include intermediate algebraic steps and define all quantities when introduced. This will greatly improve verifiability.

### (iii) Clarify (or reconcile) the mismatch between asymptotic regimes

The paper uses:
* a “dynamics” regime resembling $n\to\infty$ with finite $d$,
* and a “generalization” regime with $n,d\to\infty$ and $\gamma>1$.

The paper should clarify whether these analyses are intended to describe the same phenomenon, and if so, how they are consistent. If they are meant to answer different questions, this should be stated explicitly and cleanly.

### (iv) Strengthen novelty/positioning relative to prior work

The paper should be more explicit about what is new relative to prior work, such as Atanasov et al. (2022), which already discuss two-phase training in two-layer linear networks and include a discussion of unwhitened data. On the generalization error side, many studies examine how the data covariance structure affects generalization error and its evolution with training (see, for example, "Multi-scale Feature Learning Dynamics: Insights for Double Descent",  Pezeshki et al. (2021)). The authors should clarify precisely what is nontrivial in their contribution.

# 2) Recommended (would strengthen the paper)

Include at least one experiment beyond the exact synthetic spiked model (even semi-synthetic), to probe whether the qualitative conclusions persist.

---

> ### Author Response · Authors · 2026-04-08
> **Response to reviewer kV9Q**
>
> Thank you for acknowledging the relevance of our work and for the detailed suggestions to improve its presentation. We refer the reviewer to our General Response to All Reviewers for an overview of how we have addressed your "Critical" requests regarding the formal presentation of our theory (Concern 1-i) and the expanded derivations and intermediate algebraic steps (Concern 1-ii). Below, we address your remaining points in detail:
>
> **Response to 1-iii: Clarifying the mismatch between asymptotic regimes**
>
> We appreciate this insightful observation regarding the mismatch between the regimes. As noted, the "dynamics" and "generalization" analyses operate under different regimes to address distinct questions. Reconciling these two distinct analyses into an unified theoretical framework is a significant challenge and remains an important direction for future work. We have added explicit text to the Discussion and Conclusion section to clarify this distinction and acknowledge this limitation.
>
> **Response to 1-iv: Strengthening novelty/positioning relative to prior work**
>
> We appreciate the suggestion to further clarify our novelty. We have updated the Related Work section to better distinguish our contributions:
>
> - **Dynamics (Atanasov et al., 2022):**
>
> We acknowledge that Atanasov et al. (2022) discuss two-phase training dymamics and the effects of data anisotropy. Their analysis of anisotropy relies on a linear approximation near the origin and this effectively demonstrates the initial alignment direction $\Sigma_{xy}=\Sigma \beta$ of the weights and allows them to deduce that a subsequent realignment to $\beta$ is necessary to reach the final solution. However, the intermediate non-linear trajectory is not resolved. Specifically, once the weights grow and the linear approximation breaks down, their framework cannot theoretically characterize the nonlinear trajectory nor the explicit timescales of the learning process from saturation to final convergence.
>
> Our work extends this theoretical understanding. First, we use a spiked covariance model to reduce the high-dimensional dynamics into a tractable two-variable system. This reduction allows us to explicitly characterize the sigmoidal growth and saturation in the early phase. Furthermore, we capture the subsequent transition and convergence in the later phase, deriving explicit timescales.
>
> - **Generalization (Pezeshki et al., 2022, etc.):**
>
> We appreciate you pointing us to this relevant literature. While works like Pezeshki et al. (2022) examine generalization error dynamics for anisotropic data, such studies typically assume that only one layer is optimized while the other remains fixed. In contrast, our work focuses on the final generalization error for two-layer linear networks where both layers are updated jointly. Deriving the generalization dynamics for jointly trained multi-layer networks is highly complex and remains an open problem. Therefore, our contribution lies in providing a precise analytical expression for the spiked covariance model that explicitly quantifies the final error's dependence on the spike's magnitude and alignment.
>
> **Response to 2: Recommended experiment beyond the exact synthetic spiked model**
>
> We appreciate this suggestion to probe whether our qualitative conclusions persist beyond the synthetic model. We would be happy to run this additional experiment. If you have any specific semi-synthetic datasets or appropriate real-world data models in mind that would be well-suited for this stylized theoretical setting, we would be glad to include these additional results in our revision.

---

> > ### Comment · Reviewer_kV9Q · 2026-05-19
> > **Follow-up comments after revision and detailed proof check**
> >
> > Thank you for the detailed rebuttal and the substantial revision effort. I carefully re-read the updated manuscript together with the expanded appendices/proofs. Overall, the paper is now significantly clearer and easier to verify than the previous version.
> >
> > In particular:
> > * the reduction to the two-variable system is much better explained,
> > * the early-phase analysis is now considerably more transparent,
> > * and the novelty positioning relative to Atanasov et al. (2022) is clearer.
> >
> > I also checked most of the derivations in Sections B, C, and D in detail. Overall, the mathematical story seems coherent and I did not find a major flaw in the main derivations.
> >
> > That said, I still think a few proof sections remain somewhat compressed, especially in Appendix B.4.
> >
> > In particular, Equation (B.105) only gives control of $\left|\tilde a(t)^2-(\tilde w_1(t)^2+\tilde w_2(t)^2)\right|$, which directly implies control of $\left||\tilde a(t)|-\sqrt{\tilde w_1(t)^2+\tilde w_2(t)^2}\right|$ but not automatically Equation (B.106), $ \left|\tilde a(t)-\sqrt{\tilde w_1(t)^2+\tilde w_2(t)^2}\right|\le C_{\mathrm{tr}}s$; unless a sign argument (e.g. $\tilde a(t)\ge0$) is explicitly invoked.
> >
> > Similarly, the proof introduces
> > $$
> > T_{\varepsilon_\infty}:=\inf\{t\ge0:\|w(t)-w_\infty\|\le\varepsilon_\infty\},
> > $$
> >
> > which implicitly assumes:
> >
> > * existence of a stable equilibrium $w_\infty$,
> > * convergence of the reduced trajectory toward it,
> > * and finiteness of $T_{\varepsilon_\infty}$.
> >
> > These facts are plausible, but they are not established locally in Appendix B.4 and instead seem to rely on later arguments from Section C. A slightly more explicit reference/connection there would improve readability.
> >
> > I also found the perturbative reduction argument around Equation (B.116) somewhat compressed. The residual estimate $ \|\xi(t)\|\le C_\xi s^2(|\tilde u|+|\tilde v|+|\tilde w_2|+s)$ is structurally plausible, since the residual terms appear to contain at least two orthogonal/remainder factors and therefore contribute at order $O(s^2)$. However, because this estimate plays an important role in the tracking argument, expanding at least one representative residual term explicitly would make the reduction easier to audit.
> >
> > Likewise, the Gronwall/comparison step seems to rely implicitly on combining the unstable kernel $e^{\|\Sigma_{xy}\|(t-t')/\tau}$ with the growth control from Lemma B.4. The overall argument is plausible, but making this interaction slightly more explicit would help clarify why the forcing term generated by $\|\xi(t)\|\le C_{\xi1}s^2\|u(t)\|$ does not amplify uncontrollably during the early phase.
> >
> > I also think Proposition 3.6 should be interpreted carefully as a conditional comparison result rather than a fully self-contained theorem on the global dynamics. The proof assumes $w_1(t)\ge w_{1\infty}$ and $w_2(t)\le w_{2\infty}$, together with the existence of the limits $(w_{1\infty},w_{2\infty})$, monotonic growth of $w_2$, and positivity properties such as $0\le w_2(t)<w_{2\infty}$. The derivation itself is correct under these assumptions, but these assumptions are not themselves established in Proposition 3.6.
> >
> > More generally, several later-phase arguments seem to implicitly rely on qualitative dynamical-systems properties (stability, monotonicity, absence of additional critical points, continuation inside compact regions, etc.) which are plausible but only partially explicit in the current presentation.
> >
> > These are mostly rigor/presentation issues rather than fundamental problems.
> >
> > One additional conceptual question I had while reading Section 3.2.2: if a single spike leads to a two-phase learning dynamics, do the authors expect a covariance with $k$ separated spikes to generically induce something like $k+1$ sequential learning phases? I would be interested in hearing the authors’ intuition on this point.

---

> > > ### Author Response · Authors · 2026-05-27
> > > **Response to Reviewer kV9Q’s Post-Revision Comments**
> > >
> > > We thank the reviewer for the careful reading and the constructive suggestions regarding the proof details in Appendix B.4 and Proposition 3.6. In response, we have further revised the manuscript to expand several compressed proof steps and to clarify the conditional nature of Proposition 3.6. The corresponding revisions are highlighted in blue in the updated manuscript.
> > >
> > > - **Sign issue in the bound for $\tilde a(t)$:**
> > >
> > > We thank the reviewer for pointing out this issue. In the revised Appendix B.4, we have made more explicit the passage from the estimate on $\left||\tilde a(t)|-\sqrt{\tilde w_1(t)^2+\tilde w_2(t)^2}\right|$ to the desired estimate on $\left|\tilde a(t)-\sqrt{\tilde w_1(t)^2+\tilde w_2(t)^2}\right|$. To justify this step, we added Corollary B.5, which shows that any possible negative values of $\tilde a(t)$ are confined to the $O(s)$ regime. This allows us to replace $|\tilde a(t)|$ by $\tilde a(t)$ up to the required $O(s)$ error.
> > >
> > > - **Implicit dynamical systems assumptions behind $T_{\varepsilon_\infty}$ and later-phase arguments:**
> > >
> > > We appreciate the reviewer’s suggestion to clarify the dynamical systems assumptions used in this part of the proof. To make these assumptions explicit, we have added new lemmas and corollaries in Appendix C.2.1.
> > >
> > > These results establish the existence and uniqueness of the nonzero equilibrium $w_\infty$, its local exponential stability (Lemma C.4), the convergence of the reduced trajectory toward $w_\infty$ (Lemma C.6), and the finiteness of $T_{\varepsilon_\infty}$ (Corollary C.7). We now use these results around Equation (B.118), where $T_{\varepsilon_\infty}$ is defined.
> > >
> > > Furthermore, between Equations (B.147) and (B.148), we have slightly expanded the explanation of the continuation argument inside compact regions.
> > >
> > > - **Residual estimate in the perturbative reduction:**
> > >
> > > We thank the reviewer for this helpful suggestion. We have added an explicit expansion of a representative residual term arising from the projected $w_2$ equation near Inequality (B.121). This additional calculation makes explicit the structure behind the estimate and clarifies why the corresponding residual contribution is of order $O(s^2)$.
> > >
> > > - **Gronwall/comparison argument:**
> > >
> > > We have made the comparison step more explicit. In particular, we have added intermediate derivations leading to Inequality (B.139).
> > >
> > > - **Interpretation of Proposition 3.6:**
> > >
> > > We are grateful to the reviewer for highlighting this point. We have revised the text preceding Proposition 3.6, as well as the proposition statement itself, to make explicit that the result is conditional on the stated comparison assumptions.
> > >
> > > For completeness, we also note that an unconditional estimate of the later-phase transition time can in principle be obtained from a loss-based argument, such as the one leading to Equation (B.95), or from the argument used in Corollary C.8. However, such arguments control convergence through quantities involving the time-varying weight norm along the trajectory. To obtain a bound that does not explicitly track this norm, one would need to control the norm uniformly during the later phase, for example by replacing it with a coarse lower bound. This would lead to a less refined timescale estimate. By contrast, under the comparison assumptions stated in Proposition 3.6, we can control the later phase directly without tracking the time-varying weight norm throughout the argument.
> > >
> > > **If a single spike leads to a two-phase learning dynamics, do the authors expect a covariance with $k$ separated spikes to generically induce something like sequential $k+1$ learning phases?**
> > >
> > > We thank the reviewer for this interesting question. We are currently considering an extension of our theoretical framework to the case of $k$ separated spikes. Our intuition is that sequential growth of multiple components associated with the spikes may occur in such a setting. However, it is not yet clear whether this component-wise sequential growth would necessarily appear as $k+1$ clearly distinguishable phases in the training loss.

---

### Review · Reviewer_oJvE · 2026-02-21

**Summary Of Contributions:**

## Summary
The authors study impact of anisotropy of data distribution on learning dynamics and generalization. In this article, anisotropy is modeled via spiked covariance structure in the marginal input distribution. The class of predictors under consideration is the hypothesis class containing two-layer linear networks and the learning task is regression.

The authors observe experimentally that there are two distinct phase. An early phase where the updates to the parameters are informed by the input-output correlation and then a phase transition to the next phase where updates are informed by the input covariance. They are able to "trace the dynamics" by fitting it with synthetic differential equations. Note: This observation and the observed equivalence of the trajectory is both on a 2d-phase plane defined by a random vector and a particular orthogonal vector $\mu_{\perp}$.

The article concludes with a brief discussion on the impact of the alignment of the target linear predictor weight $\beta$ with the spike vector $\mu$ on generalization.

## Concerns
1. The analysis is in the limit of infinite training samples, and the limit of infinitesimally small initialization scale.
2. Equation 15 requires more formal justification. It is unclear to me how the bound on the timescale for Equation 15 is derived. Appendix B assumes one is solving a simplified dynamical equation that ignores the $O(\sigma^3)$ terms in Equation 14.
3. Equations 16, 17 observe a "symmetry" in the dynamics of $w_1$ and $a$. As a consequence, the authors suggest that a single common dynamical equation tracks the evolution of both parameters. Is that rigorous? If so, what is the formal mechanism for checking it? If I'm understanding correctly, the authors appear to suggest that their theoretical simplifications are justified due to the experimental evidence in Figure 2, but this in itself is not a formal theoretical argument..
4. The formal inference from Appendix C is unclear. In particular, can the authors state a rigorous mathematical theorem complete with assumptions and rigorous asymptotic statements? The discussion in Later Phase (3.2.2) reads as an informal imprecise report on mathematical calculations. A reader would strongly benefit from formal arguments.

## Conclusion
This article investigates an interesting question on the geometry of the data distribution impacts the dynamics of learning algorithms. However, as a reader, I found it hard to follow their arguments, and even harder to verify their statements.  At various points, the authors track terms as $O(s)$ or $O(s^3)$ where $s$ is the standard-deviation of the parameters at initialization. Ideally, this discussion should be made formal by way of explicit formal theorems stating explicit assumptions. Even analysis for asymptotic choices of hyperparameters (e.g. sample size, scale $s$, gradient step, regularization $\lambda$ etc) should ideally be rewritten in terms of $\lim\sup$ or other appropriate metrics. Instead, the reader finds new assumptions and informal approximations layered on top of each other and this makes the overall narrative hard to verify.

I want to emphasize that, this form of analysis, might be reasonable and perhaps even valuable when the underlying dynamics are intractable. However, as a reviewer I am unable to verify its correctness. At present, the experimental phase plots appear to serve as the primary evidence for the relevance of the largely approximate simplified theoretical dynamics.


-----
# Post-Revision


I thank the authors for their signficant effort in addressing the reviewers feedback.


## Response to my prior concerns

For each point below, I'm recalling concerns I raised earlier along with the authors pointed effort to address it.

1.  _The analysis is in the limit of infinite training samples, and the limit of infinitesimally small initialization scale._
The revised discussion explicitly acknowledges that the dynamics analysis is at finite $d$ with $n \to \infty$ and that the generalization analysis is at $d, n \to \infty$ with $d/n \to \gamma$

2. _Formal justification for the Equation 15 timescale_
Proposition 3.5 (and Appendix C.1.2) states the decoupled Bernoulli differential equation, Appendix C.1.3 derives leading-order timescale through a more explicit asymptotic computation.

3. _Symmetric/Equivalent dynamics for  $a$ and $w_1$._
Theorem 3.3 states a uniform-in-time bound $\|a(t) - \hat{r}_1\sqrt{w_1^2 + w_2^2}\| \le Cs$ for all $t \ge 0$.
The proof in Appendix B.4 decomposes the dynamics into growing three distinct time intervals for comparing the projected exact trajectory and the _ideal_ reduced trajectory

4. _Informal arguments in Section 3.2.2. on later phase dynamics_
The original Section 3.2.2 is now replaced by a more explicit narrative via Propositions 3.4–3.6.

Some additional remarks:
1. Proposition 3.6 assumes monotonicity: $w_1(t) \ge w_{1,\infty}$ and $w_2(t) \le w_{2,\infty}$, it is unclear to me if this assumption is restrictive.
2. Several theorems state that there exists constants gauranteeing a bound. While this is acceptable an explicit dependence on relevant parameters such as $(\rho, A, \sigma^2, d, \mu, \beta)$ In summary, I think the revised manuscript has sufficiently addressed my concerns. I recommend acceptance.

The scale of changes in the revised manuscript is unusual. I've read the main draft, the proof sketch and they appear _sound_ however I have only skimmed the many additional proof arguments in appendices B,C and D.
 be valuable. The arguments in the appendix would indicate that this is possible for atleast a subset of the theroetical results.

**Additional Comments:**

The scale of changes in the revised manuscript is unusual. I've read the main draft, the proof sketch and they appear _sound_ however I have only skimmed the many additional proof arguments in appendices B,C and D.

**Audience:**

Yes

**Audience Explanation:**

The topic of consideration is the impact of geometry of data distributions on the dynamics of learning algorithms. This is an important and relevant question to all researchers in machine learning and optimization.

**Broader Impact Concerns:**

At this time, I do not foresee any concern on the broader impact of this article.

**Claims And Evidence:**

Yes

**Claims Explanation:**

The revised manuscript has sufficiently addressed my concerns. I recommend acceptance.

**Requested Changes:**

To secure a recommendation for acceptance, I require that the authors re-frame their discussions on both the early and later phases into formal theoretical statements with explicit assumptions, and error terms.

---

> ### Author Response · Authors · 2026-04-08
> **Response to Reviewer oJvE**
>
> We are deeply grateful for your constructive feedback and the detailed suggestions to improve presentation of our manuscript. We provide an overview of these structural improvements and the formalization of our theoretical claims in the General Response to All Reviewers. Below, we address your specific concerns in detail:
>
> **Response to Concern 1: Infinite samples and small initialization scale**
>
> Thank you for your insightful comment. We understand the concern regarding the assumptions. However, to our knowledge, analyzing linear networks in the limit of infinite training samples together with an infinitesimally small initialization scale is a standard and well-established theoretical approach. This setting makes the learning dynamics analytically tractable. In fact, within this framework, we are able to capture essential features of the learning dynamics that would be difficult to uncover otherwise. For these reasons, we believe that these assumptions are appropriate and well justified for the purpose of the present theoretical analysis.
>
> **Response to Concern 2: Formal justification for former Equation 15**
>
> We have addressed this by introducing a formal theoretical statement. The timescale bounds and the formal justification for the initial alignment have now been rigorously established in Lemma 3.1.
>
> **Response to Concern 3: Symmetry of $w_1$ and $a$**
>
> We thank the reviewer for pointing out the need for a formal theoretical justification for the symmetry of $w_1$ and $a$. The formal mechanism justifying this symmetry is now explicitly stated through Lemma 3.1 and Proposition 3.4.
>
> In the very early stage of learning, the dynamics can be linearly approximated near the origin. If we consider the difference $v = (a - w_1) / 2$, this linear approximation shows that the impact of the initial values on this difference decays exponentially, as detailed in our derivations (e.g., Equation B.25).
>
> Beyond initialization scale, when the weights grow and the linear approximation breaks down, we utilize a conservation quantity. The symmetry between $a$ and $w_1$ could only be broken by the significant growth of $w_2$. However, Proposition 3.4 rigorously bounds $w_2$ by $O(w_1^3)$. Consequently, as long as $w_1^3$ does not become too large, the approximation $a \approx w_1$ holds. This logic rigorously justifies Proposition 3.5, where the error term $\varepsilon_R(t)$ derived from $w_2$ is formally bounded by $O(w_1^5)$ and the initialization error $O(s)$.
>
> **Response to Concern 4: Formal inference and the Later Phase**
>
> We appreciate this constructive suggestion to formalize our arguments. We have rigorously formalized these arguments into precise mathematical statements complete with explicit assumptions and asymptotic bounds:
>
> - **Formalizing the Global Dynamics (Lemma 3.2 and Theorem 3.3):**
> To rigorously establish the overall tracking of the dynamics, we thoroughly re-examined our previous proof path from former Appendix C and completely reconstructed it as Lemma 3.2 which bounds the collapse of the components orthogonal to $\hat{r}_1$. Building upon this, we established the new Theorem 3.3, which proves that the projection of the exact high-dimensional gradient flow onto the $\hat{r}_1$ direction is globally tracked by our reduced two-variable system. It explicitly bounds the uniform approximation error by $O(s)$ for a small initialization scale $s\to0$ throughout the entire process.
>
> - **Formalizing the Later Phase Timescale (Proposition 3.6):**
> The previously informal calculations regarding the timescale of the Later Phase have been formalized into Proposition 3.6.
>
> The complete, newly structured rigorous proofs for all these statements are now detailed in Appendices B and C.

---

> ### Author Response · Authors · 2026-05-27
> **Response to Reviewer oJvE’s Post-Revision Comments**
>
> We sincerely thank the reviewer for carefully reading the revised manuscript, and for providing constructive additional remarks.
>
> **Monotonicity assumptions in Proposition 3.6.**
>
> We thank the reviewer for pointing this out. We have revised the text to clarify that Proposition 3.6 is a conditional comparison result whose bound applies under the stated monotonicity assumptions.
>
> Without these monotonicity assumptions, one would need to estimate the later-phase transition time through a loss-based argument, such as the one leading to Equation (B.95), or from the argument used in Corollary C.8. However, such arguments control convergence through quantities involving the time-varying weight norm along the trajectory. To obtain a bound that does not explicitly track this norm, one would need to control the norm uniformly during the later phase, for example by replacing it with a coarse lower bound. This would lead to a less refined timescale estimate.
>
> By contrast, the monotonicity assumptions in Proposition 3.6, namely $w_1(t)\geq w_{1\infty}$ and $w_2(t)<w_{2\infty}$ on the relevant time interval, allow us to obtain a clean explicit upper bound on the later-phase timescale without tracking the time-varying weight norm throughout the argument.

---

### Author Response · Authors · 2026-04-08
**General Response to All Reviewers**

Dear Reviewers,

We apologize for the delay in resubmission. We sincerely appreciate your extension of the deadline for the decision. To address the reviewer’s comments, we have substantially revised the manuscript to address them, which took longer than we had initially expected. Notably, as a result of this revision, we were able to obtain results that are significantly stronger than those in the original manuscript, as stated in Theorem 3.3 and rigorously proven in the Appendix. We are deeply grateful to the reviewers for their insightful comments, which made this improvement possible.

In response to the collective concerns regarding the structure, verifiability, and completeness of our paper, we have substantially reframed the theoretical analysis sections (Sections 3 and 4) to ensure a standard, formal mathematical presentation.

Specifically, we have made the following structural improvements:

- **Separation of Formal Statements and Intuition:**
We have restructured Section 3 to introduce the dimensionality reduction first, followed by the formal analysis of the two-variable system. All formal mathematical statements (complete with explicit assumptions and error bounds) have been isolated into clearly labeled Lemmas, Propositions, and Theorems. Section 4 has been similarly restructured, with the main analytical expression now explicitly formulated as Theorem 4.1. All informal derivations have been rigorously justified.

- **Expansion of Derivations and Proofs:**
To greatly improve verifiability, we now provide the full, rigorous proofs in Appendices. Furthermore, we have expanded these derivations to include the intermediate algebraic steps omitted previously.

In addition to these structural revisions, we have made minor textual improvements and incorporated your specific feedback to refine the manuscript further. All changes made in the revised manuscript are highlighted in blue. Below, we address the specific points raised by each reviewer individually.


We thank you again for your valuable feedback, which helped improve the clarity, correctness, and completeness of the manuscript.

---

### Decision · Action_Editor_D43n · 2026-06-03

**Recommendation:** Reject

**Audience:**

Yes

**Audience Explanation:**

The topic is interesting to the learning theory community likely.

**Claims And Evidence:**

No

**Claims Explanation:**

The paper introduces a theoretical framework analyzing how data anisotropy, parameterized by a spiked model, impacts the learning dynamics and generalization error of two-layer linear networks in regression tasks. The work claims that training proceeds through two distinct phases (governed initially by input-output correlation and subsequently by input covariance). The work leverages standard random matrix theory to derive a closed-form generalization expression demonstrating performance gains when the data spike aligns with the target predictor. The topic on its own seems valid, but the submission was among the hard ones I have handled as an Action Editor so far. The original submission lacked mathematical rigor as raised by reviewers; the submitted revised version was a significant step up, but it was very delayed and as a result, the reviewers had a harder time verifying the claims. The revised submission is much better, but I believe it requires a new round of review from scratch given the extensive range of changes.  Another area I recommend the authors to make a more explicit statement is the difference from the established literature.